# How well are hazards associated with derechos reproduced in regional climate simulations?

Tristan Shepherd[1], Frederick Letson[1], Rebecca J. Barthelmie[2], and Sara C. Pryor[1]

[1]Department of Earth and Atmospheric Sciences, Cornell University, Ithaca, NY 14850, USA
[2]Sibley School of Mechanical and Aerospace Engineering, Cornell University, Ithaca, NY 14850, USA

*Correspondence to*: Tristan Shepherd (tshepnzl@gmail.com) and Sara C. Pryor (sp2279@cornell.edu)

**Abstract**

A 15-member ensemble of convection-permitting regional simulations of the fast-moving and destructive derecho of June 29 – 30, 2012 that impacted the northeastern urban corridor of the US is presented. This event generated 1100 reports of damaging winds,
significant wind gusts over an extensive area of up to 500,000 $km^2$, caused several fatalities and resulted in widespread loss of electrical power. Extreme events such as this are increasingly being used within pseudo-global warming experiments to examine the sensitivity of historical, societally-important events to global climate non-stationarity and how they may evolve as a result of changing thermodynamic and dynamic context. As such it is important to examine the fidelity with which such events are described in hindcast experiments. The regional simulations presented herein are performed using the Weather Research and Forecasting
(WRF) model. The resulting ensemble is used to explore simulation fidelity relative to observations for wind gust magnitudes, spatial scales of convection (as manifest in high composite reflectivity, cREF), and both rainfall and hail production as a function of model configuration (microphysics parameterization, lateral boundary conditions (LBC), start date, use of nudging, compiler choice, damping and number of vertical levels). We also examine the degree to which each ensemble member differs with respect to key mesoscale drivers of convective systems (e.g. convective available potential energy and vertical wind shear) and critical
manifestations of deep convection; e.g. vertical velocities, cold pool generation, and how those properties relate to correct characterization of the associated atmospheric hazards (wind gusts and hail). Use of a double-moment, 7-class scheme with number concentrations for all species (including hail and graupel) results in the greatest fidelity of model simulated wind gusts and convective structure against the observations of this event. All ensemble members, however, fail to capture the intensity of the event in terms of the spatial extent of convection and the production of high near-surface wind gusts. We further show very high
sensitivity to the LBC employed and specifically that simulation fidelity is higher for simulations nested within ERA-Interim than ERA5. Excess CAPE availability in all ensemble members after the Derecho passage leads to excess production of convective cells, wind gusts, cREF > 40dBZ and precipitation during a frontal passage on the subsequent day. This event proved very challenging to forecast in real-time and to reproduce in the 15-member hindcast simulation ensemble presented here. Future work could examine if simulations with other initial and lateral boundary conditions can achieve greater fidelity.

 **1 Introduction**

**1.1 Convective storms as a natural hazard**

Mesoscale convective systems (MCSs) are dynamically and thermodynamically complex (Houze, 2004; Chen et al., 2015; Weisman and Rotunno, 2004) and are thus challenging to simulate accurately. Deep convection significantly contributes to atmospheric hazards (e.g. heavy and/or persistent precipitation and associated flooding (Hu et al., 2020), lightning (Yair, 2018), extreme winds (Schoen and Ashley, 2011; Bedard et al., 1977)), and uncertainty in climate-related risks under global climate non-stationarity (Trapp, 2021; Allen, 2018). This has led to an increased demand for use of convection-permitting model frameworks (Lucas-Picher et al., 2021; Prein et al., 2015) and simulations to examine whether the threats posed by MCS are likely to be amplified under climate change.

Over the contiguous USA, hazards associated with deep convective systems, including derechos, are associated with substantial numbers of fatalities, injuries and infrastructure damage (Taszarek et al., 2020). Over both the USA and Europe the highest single daily counts of severe wind reports were both associated with derechos; wide-spread, long-lived windstorms (Taszarek et al., 2020; Corfidi et al., 2016). While derechos are most common in the southern Great Plains and Midwest, they have been observed in virtually all states east of the Rocky Mountains (Ashley and Mote, 2005). One meta-analysis suggested derechos represent an almost equal hazard over the USA to tornadoes and hurricanes. They found that during 1986-2003 there were an average of 21 derecho events per year that, on average, caused 9 deaths and 145 injuries (Ashley and Mote, 2005). Indeed, in a region extending east from Wisconsin into Pennsylvania, New York, and northern West Virginia derechos appear to be the dominant source non-tornadic convective wind fatalities (Schoen and Ashley, 2011). Derechos also cause disruptions to socio-economic systems (e.g. energy provision and transportation (Bedard et al., 1977)). An analysis of electric power delivery in the USA between 2003-2017 found that 50% of disruptions were associated with weather events. Thunderstorms were responsible for 47% of those disruptions and three types of events caused more than 60% of a utilities' customers power outage; a derecho, an ice storm and a hurricane (Shield, 2021). Further, a single derecho event during July 2009 resulted in the blow-down of 25 million trees in the US state of Minnesota and the Canadian provinces of Ontario and Quebec (Schumacher and Rasmussen, 2020). Derechos are also a major cause of economic losses in Europe (Gatzen et al., 2020). For example, a major derecho event tracked over Belarus, Lithuania, Latvia, Estonia, and Finland during August 8, 2010. It was associated with near-surface wind gusts of 36.5 ms$^{-1}$ and caused damage of over 2 million Euros in Estonia alone (Toll et al., 2015). Given the societal implications from deeply convective events – including derechos – there is interest in advanced understanding of simulation fidelity as a function of model configuration from both the short-term forecasting and climate science communities (Tian et al., 2017; Mathias et al., 2019; Schumacher and Rasmussen, 2020).

## 1.2 Derecho June 29 – 30, 2012

Work presented herein focuses on a severe thunderstorm complex that became one of the most destructive and fast-moving derechos in US history. An area of organized deep convection formed south of Chicago, Illinois on the morning of 29 June 2012 and subsequently intensified and propagated rapidly across northern Indiana, Ohio, over the Appalachian Mountains and into the Atlantic coast (Halverson, 2014). It passed over Washington DC, before moving out over the Atlantic Ocean. This event caused relatively little precipitation over the mid-Atlantic states but generated significant wind gusts over an extensive area of up to

500,000 km$^2$. The National Weather Service received over 1100 reports of wind damage (Halverson, 2014) (Figure 1) and wind gusts of 31.7 ms$^{-1}$ were reported at Dulles Airport during passage of the derecho (Figure 2). This event thus fits the early definition of a derecho as being a widespread, convectively induced straight-line windstorm associated with down-burst clusters produced by an extratropical meso-scale convective system (Johns and Hirt, 1987) and more recent detailed definitions such as; 1) convectively induced wind damage and/or gusts of > 25.7 ms$^{-1}$ over an area with a major axis of 400 km, 2) reports must be

geographically consistent, and 3) within the areas affected there must be 3 or more reports of convective gusts equal to (or greater than) 33.4 ms$^{-1}$ (Corfidi et al., 2016).

Over 20 deaths were reported during the 29-30 June 2012 derecho event. There was also widespread property damage and extensive power outages (Halverson, 2014). According to one report power outages impacted over half of all homes within West Virginia and "approximately 600,000 citizens were still without power a week later" (Kearns et al., 2014). Many homes in West

Virginia also lost access to clean water supply due to power failures at water treatment facilities (Kearns et al., 2014). During the evening of 29 June over 1.4 million people in the Washington DC metro area lost power, some of them for almost a week during a period of relative high heat stress (Short, 2016). Virginia, Ohio, Virginia and West Virginia had the largest number of customers without power (Halverson, 2014), and an analysis in 2016 found this event was the single largest cause of power outages in the state of Maryland (Short, 2016). Analyses of data contained in the NOAA Storm Reports publication for 29 and 30 June 2012

indicates wind related damage within the simulation domain used herein (Figure 1a) of over $44 million (2012$) and hail damage of over $200,000 (2012$), over 1300 high wind reports and approximately 130 reports of hail.

Prior research has suggested that Derecho events in the eastern USA are often preceded by large scale troughing over western North America (Cordeira et al., 2017). This was also evident in the June 2012 event, where associated ridging over the eastern US caused extreme near-surface air temperatures and humidity leading to issuance of heatwave advisories (Cattiaux and Yiou, 2013).

Rossby wave breaking lead to development of an intense elevated mixed layer (EML, 700-500 hPa) over the central US that subsequently propagated eastwards (Shourd and Kaplan, 2021). The upper-level flow early on June 29 was dominated by ridging over the southeastern US and a near-zonal jetstream extending from the middle of Wisconsin across the Great Lakes and into New York state, with an embedded jet streak over the northern Great Lakes (Shourd and Kaplan, 2021). Near-surface conditions were

dominated by a complex frontal boundary extending approximately west-east across Iowa into Pennsylvania, with very high humidity and high near-surface temperatures just to the south (Figure 3). It is noteworthy that the 12-hour forecast from the NAM model (grid-spacing of 12 km) valid at 8pm (local time) on 29 June 2012 indicated an extensive area of surface-based Convective Available Potential Energy (CAPE) in excess of 4000 Jkg$^{-1}$ over the Appalachian Mountains (covering almost all of the state of west Virginia) associated with the eastward propagation of the EML but projected very little precipitation, which contributed to uncertainty in forecasting the location and intensity of the derecho (Noaa, 2013).

Most forecast models operating in 2012 did not predict either extensive deep convection or a significant severe weather event (Schumacher and Rasmussen, 2020; Guastini and Bosart, 2016), although once it had initiated the Storm Prediction Center (SPC) commenced issuance of severe weather warnings (Halverson, 2014). A Service Assessment Team from the National Weather Service (NWS) evaluated performance during this event and found that "Unlike many major tornado outbreaks in the recent past, this event was not forecast well in advance." (Noaa, 2013). In part due to the multi-scale forcing of warm-season derechos, this, like other (weaker) derechos proved difficult to forecast > 12-24 hours ahead, and operational models including the North American Mesoscale (NAM) and Global Forecast System (GFS), provided "little assistance in forecasting this event more than 24 hours ahead of time". The day-3 and day-2 convective outlooks valid for 29 June showed only a 5% probability of severe thunderstorms anywhere over the eastern US, and even the Storm Prediction Center 1-day ahead convective outlook indicated only a 15% probability over most of the region that was impacted by the Derecho (Noaa, 2013). During the morning of June 29, some high-resolution, convection-permitting simulations with the High-Resolution Rapid Refresh model indicated the potential for development of intense thunderstorms and only in the afternoon of June 29 was the potential for tracking into the Mid-Atlantic coast identified (Noaa, 2013). This event has subsequently been the subject of extensive research in terms of characterization of the environmental context (Bentley and Logsdon, 2016; Guastini and Bosart, 2016; Shourd and Kaplan, 2021). It has also formed the basis of several modelling studies designed, for example, to examine whether model fidelity is enhanced by data assimilation (Fierro et al., 2014) and to evaluate representation of multiple storm characteristics in regional and global climate models at cloud system resolving scales (Liu et al., 2023). Our research is not focused on methods to improve forecasts of such events but rather to evaluate the inherent ability of the Weather Research and Forecasting (WRF) model to reproduce key aspects of this event in the contemporary climate as a function of model configuration to lay the foundation for examining how such events may evolve in the future.

**1.3 Synthesis of insights and outcomes from previous simulations of deep convection and derecho events**

Past research has illustrated that use of nested domains with convection-permitting resolutions (i.e., dx < 4 km), where the convective parameterization is deactivated and convective processes are partially resolved by explicit model physics, typically enhances simulation fidelity of deep convection (Prein et al., 2015). Emerging research has shown that using scale-aware

convective parameterizations (i.e. those schemes where numerical descriptions include a parameter that modulates convective processes as a function of horizontal resolution) throughout the model gray zone resolution helps to smooth the transition from the parameterized to resolved convective scale, leading to smaller errors in the timing and intensity of precipitation (Jeworrek et al., 2019; Mahoney et al., 2016; Wagner et al., 2018). However, model fidelity as a function of model configuration, remains an ongoing open research question. As described below, model fidelity is a strong function of the precise cloud microphysics scheme applied, model grid spacing, lateral boundary conditions, compiler choice, and the degree/manner in which the model parameterizations interact [for example, feedback between the cumulus parameterizations/cloud microphysics and the radiation scheme] (Warner, 2010; Wang and Seaman, 1997).

Compute times for simulations with WRF and other atmospheric models exhibit a relatively high dependence on cloud microphysics schemes (Barrett et al., 2019). Single-moment schemes do not predict particle size distribution for each species, which is instead derived from fixed parameters. They are thus more computationally efficient. Double-moment schemes, add a prediction equation for number concentration per species (cloud, water, ice, snow, hail, graupel). The trade-off between increased compute time – from more advanced microphysics – and meaningful forecast improvement is significant, such that the additional compute expense may not always be warranted (Jeworrek et al., 2019). Nevertheless, as the model resolution transitions through the gray zone to kilometer-scale resolution, the microphysics begins to directly influence convective and cloud scale motions through latent heating/cooling and the weight of condensate, thus a double moment scheme should be used at such scales (Morrison et al., 2020). Spectral bin schemes may offer an additional fidelity enhancement but are even more computationally demanding (Shpund et al., 2019). One analysis of hail prediction for an event that impacted Oklahoma City employed a horizontal grid spacing of 500 m and compared three different bulk microphysics (MP) schemes: the Milbrandt–Yau double-moment scheme (MY2), the Milbrandt–Yau triple-moment scheme (MY3), and the NSSL variable density-rimed ice double-moment scheme (NSSL). The authors found all three schemes generated skillful predictions for the surface areal coverage of severe surface hail (hail diameter (D) $\geq$ 25 mm) but particularly the NSSL scheme exhibited less skill for significant severe hail (D $\geq$ 50 mm) (Labriola et al., 2019a). Microphysics parameterizations are not only critical to production of solid precipitation (hail and graupel) but also to simulation of cold pool development and production of downbursts and outflow boundaries (Adams-Selin et al., 2013). Squall lines are well suited to microphysics sensitivity studies because mature squall lines contain a range of ice hydrometeor types (Xue et al., 2017). Much of the prior research examining squall line sensitivity to microphysics has been conducted with bulk schemes due to the added computational demand of bin schemes (Morrison et al., 2015; Fovell and Ogura, 1989; Mccumber et al., 1991; Fan et al., 2015; Fan et al., 2017). These studies have shown considerable spread to different microphysics, and this has been linked to varying representation of cold pool dynamics (Morrison et al., 2015; Morrison et al., 2012).

No optimal grid spacing has been found for simulation of MCSs including derechos. A previous analysis of 14 simulated MCSs found finer grid spacing was associated with better reproduction of the cold pool (grid spacing of 1 km showed enhanced skill over 3 km) but that forward propagation speeds of the MCS better matched observation for the simulations at 3 km (Squitieri and Gallus Jr, 2020). Further simulations of a derecho that impacted northern France, Belgium, the Netherlands and northwestern Germany on 3 January 2014 also found more realistic representation of the derecho intensity in simulations at a grid spacing of 1.1 km relative to simulations at 2.8 km (Mathias et al., 2019).

Other studies have examined the sensitivity to model initial and lateral boundary conditions (IC and LBC) (Hohenegger et al., 2006; Johnson, 2014). Modelling of the major derecho event tracked over Belarus, Lithuania, Latvia, Estonia and Finland during August 8, 2010 with the HARMONE model applied at a 2.5 km grid spacing found a strong dependence on IC and LBC and a time delay (of approximately 1 hr) in derecho passage approximately 15 hours into the simulation (Toll et al., 2015). Nested simulations of a European derecho event using the COSMO regional model found significant improvement in the simulation fidelity with use of ERA5 for the LBC over simulations using ERA-Interim (Mathias et al., 2019). Earlier work performed a 24-hr hindcast of the June 2012 North American derecho in regional and global climate models at cloud system resolving scales, and showed that both model types produce a delay of approximately 2-hr in feature location and associated gust front timing with a negative bias compared to RADAR composite reflectivity (Liu et al., 2023).

Simulation reproducibility differences can also arise from round-off error in floating-point operations, which are handled differently by different compilers and machines (e.g. optimized math, double precision, rounding modes). Resulting error growth in atmospheric properties from the convective- to meso-scale has been demonstrated (Zhang et al., 2007). While initial errors may be small, convective-scale errors can grow quickly, in scale, magnitude, and spatial extent while contaminating the mesoscale (Judt, 2018). Compiler selection also influences simulation compute time and past research has found WRF performance is often best when using the Intel fortran compiler (Moreno et al., 2020; Powers et al., 2021) although prior to 2021 Intel compilers were not freely available, which restricted their use to those platforms that held an Intel license. Earlier work has shown compiling WRF with Intel resulted in a performance gain of up to 26% compared to GCC (Langkamp and Böhner, 2011), and other work has shown simulations with WRF compiled using Intel outperformed GNU on a cloud computing architecture, leading to significant annual cost savings in computation costs (Siuta et al., 2016).

## 1.4 Objectives

It is important to emphasize that research presented herein is cast within the framework of use of short simulations with a Convection Permitting Regional Climate Model (CPRCM) to reproduce specific extreme events where a CPRCM is nested with LBC from a reanalysis product (Lucas-Picher et al., 2021). By simulating only few days, this case study (or storyline) approach can permit many simulations to be performed and evaluated and model dependencies can be fully investigated (Lucas-Picher et

al., 2021; Mathias et al., 2019). Accordingly, the objectives of this work are to build and evaluate an ensemble of WRF simulations performed in a hindcast mode (i.e. with reanalysis-derived LBC) that differ in terms of the microphysics schemes applied, the

180 LBC, start date, use of nudging, compiler choice, number of vertical levels, and use of damping and to use that ensemble to evaluate how simulation fidelity for an historically important high-wind mesoscale convective event:

1) Varies across microphysics schemes. The five microphysics schemes applied range in sophistication, from cloud-scale single-moment [Goddard (Tao et al., 1989)] to double-moment [Thompson (Thompson et al., 2008), Morrison , Milbrandt-Yau (Milbrandt and Yau, 2005b)] to double-moment with particle shape and density prediction [NSSL

(Mansell et al., 2010a)].

2) Varies for different LBC, start times and with and without nudging. The two reanalysis products used to provide the initial and lateral boundary conditions are ERA-Interim (Dee et al., 2011) and ERA5 (Hersbach et al., 2020).

3) Varies for different compiler selection. While the majority of simulations presented herein were performed with WRF compiled using GNU Fortran, two additional simulations that use WRF compiled with the Intel compiler are also

presented.

4) Varies for different model vertical resolutions (41 v. 65 vertical levels) and with/without upper-level Rayleigh damping.

For objectives 1-4 we evaluate fidelity with respect to; peak RADAR reflectivity and spatial extent of reflectivity at the time of maximum deep convection, cumulative precipitation, presence/absence of hail, and peak wind gusts. We also provide context for the fidelity assessment during the derecho with conditions during a subsequent frontal passage. We also seek to address a fifth

objective:

5) Evaluate the degree to which the processes involved in generation of gust fronts from derechos are represented in the WRF ensemble simulations. In this part of the analysis, we are seeking to assess the differential fidelity of the ensemble members in terms of a range of diagnostic properties, the vertical structure of deep convection, the vertical velocities, and metrics of cold pool production.

This research is being performed as part of a project designed to examine how historically important extreme events may be modified in an evolving climate. Thus, while there is evidence that data assimilation can substantially enhance forecast and hindcast skill (Bachmann et al., 2020; Johnson et al., 2015; Johnson and Wang, 2016; Federico et al., 2019), no data assimilation is performed here.

## 2 Data and Methods

### 2.1 WRF simulations

All the simulations presented herein were performed with WRF model version 3.8.1. The simulations follow the standard steps used for WRF simulation setup and execution – steps previously described by a simulation flowchart in other WRF studies (e.g. (Kumar et al., 2024). The optimal domain size, number of nests and parent-grid ratio to be used in convection-permitting simulations are open questions (Prein et al., 2015), but there is evidence of *bulk convergence* (i.e. diminishing change of domain-wide properties as a function of grid spacing) at approximately 1 km (Panosetti et al., 2019), and past research has indicated improved simulation fidelity with finer horizontal grid spacing (Liu et al., 2023). Accordingly, all simulations performed herein use a grid spacing of 1.33 km in the innermost domain (d03, see Figure 1a for the simulation domains) that covers a domain of almost 400 by 400 km (i.e. above the recommended target of 300 by 300 km (Lucas-Picher et al., 2021)). The same domain configuration (i.e. 12, 4, 1.33 km) is used in all members of the ensemble. Prior research has generally found sensitivities related to cloud microphysical parameterizations are larger than those associated with mesh refinement at kilometer scales (Roh and Satoh, 2014). Model configuration settings that are consistent across all simulations are shown in Table 1 while the settings for which the 15 ensemble members differ (e.g. microphysics scheme, simulations that test sensitivity to initial conditions, use of nudging, model start time, compiler, vertical levels, and upper-level damping) are shown in Table 2. Here we use a fixed outer WRF simulation domain grid spacing of 12 km with lateral boundary conditions (LBC) from ERA5 (dx ~ 30 km) and, for the simulations testing sensitivity to initial conditions, ERA-Interim (dx ~ 80 km), consistent with recommendations that the maximum step in resolution at the domain boundary (Lucas-Picher et al., 2021). Note that the Kain-Fitsch scheme used in the 12-km domain runs shallow convection by default. Within the inner domain no cumulus scheme is applied consistent with previous research. Because the goal of this research is to establish whether WRF can generate a derecho of the given intensity when provided only the large-scale environmental context, in most simulations no nudging is applied, and a relatively large simulation domain is selected. Two initialization dates are included in the ensemble; most simulations are initialized at 0000 UTC on 26 June approximately 4 days before the peak of the event. These are type equivalent to true 'climate mode' simulations (i.e. those initialized well ahead of the event genesis), another two are initialized at 0000 UTC on 28 June, approximately 2 days before the peak of the event but much closer to the event genesis and thus are closer to a 'weather-wise mode' where the model initialization is a few hours before the event commences.

Additional WRF *output diagnostics* options are employed. The '*output_diagnostics=1*' setting is used to output climate diagnostics to a separate history file (*wrfxtrm*) every hour for domain 1, and every 10 minutes for domain 2 and 3. Advanced settings for NSSL are not used here. The '*hail_opt*' switch for Morrison is used to run this scheme with hail. A Morrison simulation without hail is also run for comparison. The Goddard scheme does not include hail by default, but in this simulation

'*gsfcgce_hail=1*' is used to run the Goddard scheme with hail. The '*do_radar_ref=1*' namelist setting is used to compute radar reflectivity using microphysics-scheme-specific parameters in the Goddard, Thompson, and Morrison ensemble simulations. This option is not available for the NSSL and Milbrandt-Yau schemes, but radar reflectivity is still calculated by the model for those schemes without using the microphysics parameters. Two radar reflectivity estimates are provided by WRF; *REFL_10CM* (i.e. radar reflectivity in each vertical grid cell at a wavelength of 10 cm) and *REFD_MAX* (maximum derived radar reflectivity). Composite reflectivity (cREF) is used here for comparison with RADAR estimates and is the maximum value for each WRF column and time step.

## 2.2 Model evaluation

The ensemble of WRF simulations is evaluated against observations from National Weather Service (NWS) dual-polarization RADARs (Seo et al., 2015; Crum et al., 1998) and the NWS Automated Surface Observation System (ASOS) (Schmitt and Chester, 2009). There are four RADAR stations within the innermost WRF simulation domain (d03) and nine in the second domain (d02). There are 34 ASOS stations in domain d03 and 149 in domain d02 (Figure 1a).

### 2.2.1 ASOS data

The following parameters from the 5-minute ASOS data set are used in the model evaluation and diagnostic interpretation:

- Gust wind speeds ($U_{gust}$, ms$^{-1}$): Sustained and gust wind speeds within the ASOS network are measured using Vaisala 2-D sonic anemometers deployed at 10 m a.g.l.. The data are sampled at 1 Hz and digitally output as 3-second moving average wind speed. The gust wind speeds reported here represent the maximum 3-second wind speed measured in each 5–minute period when gust criteria are met. Gusts are reported in knots and are rounded *up* to the nearest whole knot. Wind gusts are reported when (Nadolski, 1998; Noaa, 2004):

    1. $U_{gust}$ is at least 3 knots (1.54 ms$^{-1}$) above the current, running 2-minute mean wind speed,

    2. $U_{gust}$ exceeds the minimum 3-second average in the last 10 minutes by at least 10 knots (5.14 ms$^{-1}$) and,

    3. The current 2-minute average wind speed is at least 3 knots (1.54 ms$^{-1}$).

- Air temperature (T, °C): measured at 2 m a.g.l. using a platinum-wire resistance thermometer.

- Sea-level Pressure (SLP, hPa): derived from station pressure measured using a digital, capacitive pressure sensor plus station altitude and ambient temperature.

- Accumulated precipitation (PPT, mm): Hourly precipitation is measured by a heated, tipping-bucket rain gauge. The data are reported in hundredths of an inch and converted to metric units herein.

A light emitting diode weather identifier instrument is used to differentiate rain and snow at ASOS stations (Wade, 2003), but hydrometeors such as hail are only reported at ASOS stations with human observers. Thus for ~ 400 fully automated ASOS stations

across the US there are no hail reporting functions. Hence, hail occurrence reported by the ASOS network (including the portion within the current domain of interest) is likely to be negatively biased. ASOS facilities with a surface-based observer also augment the reports with flags to indicate the presence of thunderstorms. These data are presented herein to supplement evidence of high reflectivity from RADAR. We also employ data from all 28 rawinsondes within the simulation domain in the fidelity assessment of the initial conditions from each reanalysis product and start time. In these analyses the conditions on two geopotential surfaces (700 hPa and 500 hPa) as derived using WRF real from the ERA5 and ERA-Interim reanalysis products are interpolated to these pressure levels using the wrf_interp program (available at: https://github.com/pick2510/wrf_interp) and the rawinsonde observations for the closest release time.

### 2.2.2 RADAR

Dual polarization Doppler S-band WSR-88D RADAR form the basis of the NWS network (Crum et al., 1998; Seo et al., 2015). Scans are performed at between nine and fourteen elevation angles (0.5° to 19.5°) depending on precipitation conditions. Data are collected with a standard azimuthal resolution of 1° and range resolution of 0.25 km (Noaa, 2016a, 2017). Data used herein are restricted to within 200 km of each RADAR station.

Five key RADAR-derived properties sampled at 10-minute intervals are used in the WRF model evaluation:

- Composite reflectivity (cREF, dBZ) which is the maximum reflectivity in each vertical column.
- Precipitation rate (mmhr$^{-1}$) derived from reflectivity using Z-R relationships (Noaa, 2016a).
- Hail reports and MESH: Hail presence in cloud is derived from reflectivity, aspect ratio of hydrometeors, vertically-integrated liquid, and altitude of the melting layer (Noaa, 2016b; Witt et al., 1998). Hail reports include the geographic position and the 75th percentile hailstone diameter (or maximum estimated size of hail, MESH) (Johnson et al., 1998; Wallace et al., 2019). In the current work, a distinction is drawn between hail reports with MESH > 25 mm and those without. This is a diameter threshold that has been previously used for identifying 'severe hail' (Labriola et al., 2019b).
- The NCEP/EMC 4KM Gridded Data Stage IV precipitation product (Du, 2011) which is a blend of RADAR-derived precipitation and in situ measurements is also used in the model fidelity assessment. The spatial fields of accumulated precipitation from the RADAR and the Stage IV product are very similar but the total domain-wide amounts during the Derecho and Frontal periods differ.
- Radial wind speeds (ms$^{-1}$) are presented herein (Figure 2) from the 0.5° elevation angle and are computed from the Doppler shift (Alpert and Kumar, 2007).

All RADAR measurements are sampled at a 10-minute interval to match the WRF output and are re-gridded onto the WRF grid used for domain d03 prior to their use in the model evaluation. Where two RADAR cover the same area the data are averaged

using inverse-distance weighting. RADAR coverage of domain d03 is almost complete. RADAR data are available for 86436 total grid cells in d03 which is 99.4% of the total number of WRF grid cells.

## 2.3 Assessing and attributing model fidelity

The WRF simulation period encompasses both the derecho that forms the focus of this research and a subsequent frontal passage. These two periods are each associated with high cREF in RADAR output and WRF ensemble members and are separated by a short period of lower reflectivity (zero d03 cells exceeding 40 dBZ) (Figure 4a). The number of ASOS stations in d03 reporting thunderstorms also show a clear distinction between these two events (Figure 4a), and is used to delineate:

    1. Derecho period:  29-Jun-2012 21:30:00 to 30-Jun-2012 13:30:00 UTC
2. Front period:      30-Jun-2012 15:20:00 to 01-Jul-2012 14:50:00 UTC

All 15 members of the WRF ensemble exhibit a time delay in simulating the derecho intensification and passage as represented by the period of the spatial extent of cREF > 40 dBZ in domain d03 relative to RADAR observations (Figure 4a). This is consistent with previous research that indicates WRF simulations not subject to data assimilation exhibit timing offsets when simulating extreme precipitation events (Knist et al., 2020). For this reason, and because the purpose of the current work is to examine whether 305 a CPRCM simulation can generate atmospheric hazards associated with a derecho, the model evaluation is performed within a framework such that time-synchronization is not required. The storm peak time ($t_p$) is defined independently for each ensemble member and the RADAR observations as the time of maximum exceedance of 40 dBZ during the Derecho period and the Front period, respectively. WRF output at $t_p$ is used to characterize the intensity and characteristics of each event.

The fidelity of each ensemble member with respect to storm severity and spatial extent during the Derecho and Front periods 310 is assessed using geospatial maps of composite reflectivity, precipitation accumulation and type, and maximum wind speeds, and is summarized using the following metrics:

- cREF >40 dBZ Ratio: This metric is the ratio of areal extent of WRF grid cells with composite reflectivity > 40 dBZ at $t_p$, divided by the RADAR-derived estimate. Use of cREF > 40 dBZ as the index of the spatial coverage of deep convection is based on past research (Schumacher and Johnson, 2005; Parker and Knievel, 2005). The spatial coverage 315 for other thresholds is shown in Figure 4b.

- Max Gust Ratio: This metric is the ratio of the maximum over-land 10m wind speed in each timestep from each WRF ensemble member divided by the maximum wind gust speed from any ASOS station. This is thus a basic metric of the degree to which each WRF ensemble member produces wind gusts that approach the most severe gusts observed by the ASOS network.

- Total Precipitation Ratio: This metric is the ratio of precipitation accumulation in all d03 grid cells for which RADAR retrievals are available to the RADAR observations. Stage IV precipitation data is also included. Each ensemble

member exhibits slightly higher agreement with the Stage IV precipitation product than with RADAR-only total accumulated precipitation during the Derecho period (Table 3). Hail occurrence from the WRF ensemble members is also evaluated against RADAR and ASOS observations along with the presence of 'severe hail'. Grid cells in d03 are classified as containing 'severe hail' in the WRF simulations and RADAR observations when MESH > 25mm. MESH for the WRF simulations is estimated using a weighted summation of hail kinetic energy flux for elevations above the melting layer. Hail kinetic energy fluxes are inferred as a function of reflectivity. This method was developed for use with RADAR data (Witt et al., 1998).

As described above, and indicated by Figure 4, the timing of peak intensity and transit of the derecho across the innermost domain is not consistent across the WRF simulations and/or between the WRF simulations and observations. Given this research is being performed in the context of a project designed to improve simulation of atmospheric hazards in the contemporary and possible future climates, we assess fidelity without requiring temporal synchronization. Thus, in the following we focus much of our evaluation of the simulations on their ability to reproduce the intensity and spatial extent of the derecho and thus define the time of peak intensity ($t_p$) independently for each ensemble member. While we present some of the evaluation in terms of the degree of spatial agreement with in-situ and remote sensing data using spearman correlation of geospatial values at $t_p$, we also include analyses that examine the absolute intensity of, for example, reflectivity and wind gusts without requiring geospatial coherence between the model and the observations. In these analyses we are addressing the question; was the peak intensity of the event captured even if that peak is displaced in space and time? In considering these decisions it is worth reemphasizing that the purpose and concept of this analysis is not to assess deterministic (forecast) predictability but the representation of the convective system.

The metrics of fidelity described above are considered here in the context of the environmental setting; convective available potential energy and vertical wind shear, along with descriptors of the storm dynamics; vertical velocities, cloud depth, downburst intensity and cold pool generation/intensity during the Derecho period. Many of these diagnostic analyses focus on the time of maximum coverage of high reflectivity ($t_p$) during the derecho as assessed for each individual ensemble member and/or over a window of 3 hours around that time. The metrics used are described in the following.

1. Convective Available Potential Energy (CAPE) is a measure of the available vertically integrated buoyant energy. Multiple indices of convective potential have been proposed (Kunz, 2007). Derechos are frequently associated with CAPE values in excess of 2400 $Jkg^{-1}$ in the genesis region and can increase to 4500 $Jkg^{-1}$ during the propagation of the derecho, with later observational analyses indicating that Most Unstable Convective Available Potential Energy (MU-CAPE) has a 75th percentile value of nearly 4000 $Jkg^{-1}$ and a peak of 8500 $Jkg^{-1}$ (Evans and Doswell, 2001). MU-CAPE from a WRF simulation at 3 km of a Super Derecho in Kansas on 8 May 2009 was in excess of 3000 $Jkg^{-1}$ (Weisman et al., 2013). MU-CAPE for the 3 July 2003 derecho in the Midwest ~ 500 $Jkg^{-1}$ (Metz and Bosart, 2010). The June 2012 derecho that

forms the focus of this research is remarkable not only for the number and intensity of wind gusts but also in terms of the Convective Available Potential Energy (CAPE) in the genesis region and near Washington DC. For example, CAPE estimates for the 0000Z 30 June 2012 rawinsonde sounding from Sterling VA (IAD) ~ 5500 Jkg$^{-1}$. Here we employ maximum or most unstable CAPE (MU-CAPE) as our primary index of the ability of the atmosphere to support deep convection. MU-CAPE is computed for $t_p$ and $t_p$+/-3 hours from the 3-dimensional fields of pressure, temperature, and water vapor mixing ratio using the WRF-PYTHON algorithm (https://wrf-python.readthedocs.io/en/latest/user_api/generated/wrf.cape_2d.html). This algorithm computes MU-CAPE of 500 m depth within the lowest 3000 m of the atmosphere that has maximum equivalent potential temperature. The change in average (median) MU-CAPE from all grid cells in d03 between $t_p$-3 hours and $t_p$ is used as a metric of the degree to which MU-CAPE is exhausted during passage of the derecho.

2. Wind shear from the ground to 6 km (S6) is often used to differentiate environments associated with significant severe thunderstorms from less severe events (Brooks et al., 2003). In an analysis of observational data average shear vectors in the ambient environment close to derechos ranged from shear vector magnitudes ranging from 1 to 36 m s$^{-1}$, which were slightly lower than those manifest in idealized simulations of bow echoes (Evans and Doswell, 2001). Mid-level shear has also been shown to help maintain deep convective systems (Coniglio and Stensrud, 2001; Chen et al., 2015). S6 is presented based on output at $t_p$ for all ensemble members.

3. $Z_{R20}$: Is the model height at which the 90$^{th}$ percentile base reflectivity falls below 20 dBZ. It is used as a proxy for cloud top height in areas of deep convection and thus is computed using only cells with cREF > 40 dBZ.

4. Two metrics of the intensity of vertical motions are presented. For each grid cell within 50 km of one where cREF > 40 dBZ, the layer with highest standard-deviation of vertical velocities ($\sigma(w)$) at $t_p$ is found. The magnitude of $\sigma(w)$ is used to provide information about the intensity of vertical motions, that to the first order should be a function of MU-CAPE. The height at which the maximum standard deviation of vertical velocities ($\sigma(w)$) is used to infer the intensity and vertical structure of convection. Since updrafts and downdrafts are of relatively short duration and small spatial extent, we use the spatial standard deviation $\sigma(w)$ computed using vertical velocities output from the time of maximum cREF > 40 dBZ (i.e. from the 10-min time step WRF output file at that time) considering all WRF grid cells within 50 km of cREF > 40 dBZ. This is a more descriptive metric rather than the mean velocity because the dispersion around the mean is reflective of the intensity of both downdrafts and updrafts in the column.

5. Cold pools are a key component contributing to organization and propagation of MCS (Engerer et al., 2008). They are generated by evaporative cooling, precipitation drag, and downdrafts and are key to triggering and organizing organized persistent convection (Knippertz et al., 2009; Schumacher, 2015). An analysis of cold pools associated with 39 MCS in

Oklahoma found mean surface pressure perturbations associated with cold pools range from 3.2 hPa to 4.5 hPa and mean temperature perturbations range from 9.5 to 5.4 K depending on the MCS stage (Engerer et al., 2008). To account for the presence of substantial topographic variability within d03, the intensity of cold pools at the surface associated with the derecho is quantified using anomalies from the simulation mean temperature or pressure in that grid cell over the entire simulation period. Both are computed for WRF grid cells with 50 km of all cells with cREF > 40 dBZ:

a.  95% temperature deviation: This metric is the lowest 5-percent (coldest) 2m air temperature anomalies close to the regions with most active convection.

b.  95% SLP deviation. This metric is the highest (positive) perturbation in sea-level pressure (SLP) anomalies close to the regions with most active convection.

Because variables and metrics considered here are not gaussian distributed, Spearman rank correlations (Wilks, 2011) are used to describe their co-variability. Rank correlation coefficients are computed between the model fidelity metrics and the diagnostic metrics across the 15 ensemble members to identify which model properties (wind speed, precipitation etc.) exhibit highest association with the diagnostic metrics used to examine model skill in simulating this event.

## 3    Results

### 3.1  Model fidelity assessment

For the Derecho period, model skill for this ensemble exhibits substantial sensitivity to variations in the microphysics schemes. These dependencies are not unexpected based on past research on deep convection (including squall lines) summarized in section 1. The fidelity of the ensemble members also varies with LBC, start time, use of nudging, compiler selection, number of vertical levels and use of upper-level damping. The fidelity assessment results are described here and then are explored further below in terms of how they link to diagnostic metrics of convective intensity.

The maximum areal extent of composite reflectivity (cREF) > 40 dBZ during the Derecho period (29 June 2012 21:30 – 30 June 13:30) varies widely across the WRF ensemble members but most are negatively biased relative to the RADAR (Figure 5) consistent with previous simulations of this event (Liu et al., 2023). This bias is least marked in the Morrison-XXXX and Milbrandt-XXXX simulations (where XXXX indicates the ensemble member in terms of LBC/start date/compiler from simulations that employ the Morrison and Milbrandt microphysics scheme), especially those using ERA-Interim for initial and lateral boundary conditions. The areal extent of cREF > 40 dBZ at $t_p$ for the non-nudged, simulations with Milbrandt and LBC from ERA-Interim is 84 – 95% as large as that from RADAR (Figure 5). For these two simulations (Milbrandt-626-ERA-I and Milbrandt-628-ERA-I) that differ only in terms of the time of initialization, the shape and orientation of the derecho is broadly similar to the observations, although the timing of the $t_p$ is greatly delayed (Figure 5) and the precise location of the regions of highest RADAR reflectivity

are incompletely reproduced (Table 3). The simulations with the smallest extent of high reflectivity are the nudged simulations with the Milbrandt microphysics scheme, followed by (in increasing order of coverage) Morrison, Thompson, NSSL, and Goddard (Figures 5 & 6). Use of the hail flag in the Morrison scheme does not lead to increased precipitation accumulation in contrast to past research on squall lines (Morrison et al., 2015). However, there is evidence that increasing the vertical model resolution does

increase accumulated precipitation. Accumulated precipitation is higher in simulations with Morrison that use 65 levels irrespective of whether upper-level damping is applied (Figure 7), and there is also a small increase in the extent of cREF > 40 dBZ (Figure 5). During the subsequent Front period (30 June 2012 15:20 – 1 July 2012 14:50), all schemes produce cREF > 40 dBZ that covers a larger area than is indicated in the RADAR observations (Figure 6). As described below, this appears to be linked to the weaker derechos resulting in excess MU-CAPE being available during the frontal passage. The nudged simulations produce the smallest

extent of cREF > 40 dBZ during the Front period, and thus show closest accord with the RADAR observations. Further, the Morrison 65levels+Hail simulation has higher spatial coverage of hail during the Derecho period than the equivalent simulation with 41 levels (Table 4) but the highest spatial coverage of hail is in simulations with 41 vertical levels, and that uses the Milbrandt scheme and ERA-Interim LBC.

The RADAR data indicate localized heavy precipitation in an east-west line across the north of domain d03 during the Derecho

period with total accumulations exceeding 38 mm over the 16-hour period in a few locations. However, the RADAR indicates generally low precipitation, that is also reflected in the ASOS in situ observations (Figure 7). Most WRF ensemble members exhibit a negative bias in terms of accumulated precipitation during the Derecho period, although there are marked differences between the different parameterizations (Figure 7). Ensemble members; Morrison+Hail, Thompson and NSSL exhibit very little precipitation anywhere in domain d03. The nudged simulations using ERA5 and ERA-Interim and the Milbrandt microphysics

scheme also produce very little precipitation anywhere in the domain in this period. The ensemble members using Milbrandt and ERA-Interim LBC (Milbrandt-626-ERA-I and Milbrandt-628-ERA-I), and Morrison Int, Morrison 65 and Morrison+Hail 65 show the highest precipitation totals. The closest agreement in terms of the spatial fields of accumulated precipitation are found for the simulations within ERA5 LBC and the Milbrandt microphysics scheme (Table 3).

When remapped to the WRF grid, the RADAR data indicate 824 of the almost 90,000 grid cells experienced severe hail during

the Derecho period (Table 4). These locations identified by the RADAR detection algorithm as exhibiting hail and MESH > 25 mm are distributed throughout domain d03 (Figure 7). The WRF ensemble members – particularly those that employ the Milbrandt microphysics scheme (and the Morrison 65 levels with hail simulation) indicate much greater spatial coverage of hail (Table 4). When the threshold of MESH > 25 mm is applied to the WRF output the occurrence of hail greatly decreases rather few grid cells show hail above this threshold (Table 4).

During the Front period the situation is reversed in that RADAR observations show limited areas with high precipitation totals over 40 mm and 2152 grid cells where hail was detected in clouds. Areas with substantial precipitation accumulation are only evident from RADAR in bands in the south of the domain, in regions where hail is also indicated by the RADAR detection algorithm (Figure 8). Two-thirds of the domain shows little or no precipitation in either RADAR or ASOS data. All non-nudged WRF ensemble members indicate positive bias in domain-wide precipitation and over-predict the occurrence of hail (Table 4). All four non-nudged ensemble members with the Milbrandt microphysics scheme simulations also indicate multiple locations with hail accumulation above 1 mm. The number of grid cells with RADAR detection of hail (3078) shows closest agreement with the Morrison+Hail simulation (3000) (Table 4). Using MESH > 25 mm and WRF hail accumulation of 1 mm as indicative of substantial hail, the closest accord for the Front period is found for the Milbrandt-628 ensemble member (Table 4).

The vertical cross-sections of RADAR reflectivity at $t_p$ in the Derecho period in grid cells with cREF > 40 dBZ show similar dependence on microphysics scheme to those manifest in the cREF and the precipitation analyses (Figure 9, see SM Figure S1 for the same visualization for the Front period). Vertical profiles of base reflectivity data from each 360º arc scan at each elevation angle from each RADAR at $t_p$ are also shown in Figure 9. Though this observationally constrained vertical profile is based on considerably lower data volumes than in the WRF output, it is noteworthy that the peak in reflectivity from the RADAR is located lower in the atmosphere than in most of the WRF ensemble members. Further, a greater fraction of the reflectivity values at 12 km (the highest height from which any RADAR data are available) from the RADAR observations are > 20 dBZ than in many, but not all, of the ensemble members. Analyses of output from the Morrison ensemble member indicates many grid cells with estimated base reflectivity > 35 dBZ over a deep layer (up to 10 km), while the Morrison+Hail ensemble member indicates fewer grid cells with cREF above 40 dBZ (Figure 5) and fewer of those grid cells have a base reflectivity > 30 dBZ above 5 km (Figure 9). Ten-percent of grid cells from the Thompson ensemble member that have cREF > 40 dBZ also exhibit base reflectivity at a height of approximately 4 km that exceeds 55 dBZ but the spatial variability in this metric of cloud droplet number and size concentration at a height of approximately 4 km is the highest of all ensemble members and the relatively shallow nature of the convection (i.e. depth of high base reflectivity, Figure 9) is consistent with the relatively low precipitation totals (Figure 5). The simulations that use the Milbrandt microphysics scheme tend to have deep layers with base reflectivity above 35 dBZ and lower spatial variability (Figure 9), consistent with the high production of hail (Table 4). In contrast to the other ensemble members, the nudged simulations with LBC from both ERA5 and ERA-Interim indicate the region of highest inferred RADAR base reflectivity at $t_p$ that is displaced from the ground (Figure 9). Increasing the number of vertical levels from 41 to 65 caused an increase in derived RADAR reflectivity peak (Figure 4) and total precipitation from simulations with the Morrison microphysics scheme during the Derecho period (Figure 7) and slightly increased model fidelity for those properties, but the differences relative to the simulation with coarser vertical resolution are comparatively minor (Table 5).

Links between deep convection, downdrafts and near-surface wind gusts are highly complex (Brown and Dowdy, 2021b; Geerts, 2001; Kuchera and Parker, 2006), and this combined with observational limitations mean very little previous research has quantified skill in model simulations of wind gust generated by downdrafts from deep convection. Consistent with evidence presented above of spatial displacement of the regions of deepest convection, the spatial correlation coefficients of maximum wind speeds between the individual ensemble members and ASOS wind gust observations (see time series in Figure 10a and spatial

maps in Figure 11) are also low (Table 3). As with precipitation and RADAR reflectivity, wind speeds are underestimated during the Derecho period and overestimated during the Front period. Some ensemble members (again, Milbrandt-XXXX and Morrison) produce wind gusts during the Derecho period that are within a factor of 0.6 of the ASOS maximum observed wind gust, but only one of the ensemble members generates a wind gust anywhere in domain d03 that exceeds the NWS definition of 'severe wind' (i.e. wind gusts at 10 m a.g.l. above 25.7 $ms^{-1}$) while multiple time periods and ASOS stations reported wind gusts above this

threshold (Figure 10b). Indeed, the highest 2% of modeled wind speeds is substantially lower than the equivalent near-surface gust observations (Figure 10b). Only the Morrison, Milbrandt-626-ERA-I and Mibrandt-628-ERA-I, exhibit 98[th] percentile wind speeds (sampled at the model time step in both space and time over land grid cells) that lie within 50% of the ASOS observations of wind gusts (Figure 10b). While some of the offset between observed point measurements of 3-second duration wind gusts and grid-cell average wind speeds at the model time step of 3.33 seconds is expected due to the spectral truncation inherent in grid-cell average

modeled wind speeds (Pryor et al., 2012), it is interesting to note that virtually all members of the model ensemble overestimate peak wind gusts during the frontal passage (Figure 10a and 12). The two ensemble members that use ERA-Interim IC and LBC are associated with highest wind speeds and greatest accord with near-surface measurements from ASOS during the Derecho period (Figure 10-12).

    The sensitivity to LBC in simulations with Milbrandt (e.g. Figure 5 and 11) is inconsistent with past research (Majewski, 1997).

Despite the higher resolution and larger data assimilation volumes in ERA5, simulations within ERA-Interim produced better spatial agreement with observations from RADAR and ASOS. For simulations with the Milbrandt microphysics scheme that are initialized on 26 June at 0000 UTC the correlation coefficients are -0.412 vs. 0.225 for ERA5 and ERA-Interim respectively, while for the simulations started on 28 June 0000 UTC the correlation coefficients are 0.318 and 0.669 (Table 3). The spatial correlation for peak cREF is also higher in simulations with ERA-Interim LBC (Table 3). An examination of the IC generated by WRF real

for 26 June 00Z (Figure 3) indicate higher pressure is prevalent and broader than in ERA-Interim, particularly across the derecho genesis region of the Midwest. The derecho event came at the end of an extended period of high near-surface temperatures. While the ERA-Interim and ERA5 fields at the model initialization time are superficially similar (on the two dates), some differences are evident (Figure 3). For example, on 26 June the region of elevated 2 m temperature extends further north and east in ERA-Interim and the SLP anomalies (and suppressed lower tropospheric specific humidity) associated with the anticyclone over the Great Lakes

is slightly more intense in ERA5. On 28 June the region of elevated 2 m temperatures extends further east in ERA-Interim. Much larger differences are naturally evident in the initialization from each of the reanalyses across the two start dates (26 June v 28 June). The weaker, but evident, influence from model initialization time (e.g. Figure 11) is consistent with information from the short-term forecasting community, although interestingly the spatial fields of precipitation accumulation exhibit higher agreement with observations in ensemble members initialized on 26 June. Evaluation of the initial conditions indicates a high degree of similarity between the two reanalysis products on 26 and 28 June for most properties (Figure 3). However, as described above, development of an intense elevated mixed layer (EML, 700-500 hPa) over the central US that subsequently propagated eastwards (Shourd and Kaplan, 2021) appears to have been a key ingredient in development of this Derecho. Earlier work (Banacos and Ekster, 2010) employed a definition of an EML as a layer of depth > 200 hPa with both a steep lapse rate (temperature declines of over 8°C per km) and an increase in the RH with height. Figure 3b shows the lapse rate in the four sets of IC and indicates that while both data sets correctly (relative to output from NOAA WRF-Rapid Refresh model presented in (Shourd and Kaplan, 2021)) indicate relatively low lapse rates at 0000Z 26 June (when the region with the EML was displaced further west), using the combined definition of a strong lapse rate and a strong gradient of RH (a 20% difference across the layer), the EML is, in both reanalysis products, displaced too far north at 0000Z 28 June relative to NOAA WRF-Rapid Refresh model simulations presented in (Shourd and Kaplan, 2021). The EML is, however, more consistent (across the two components) and more coherent in space in ERA-Interim. This may provide a partial explanation for why simulations with ERA-Interim initial and lateral boundary conditions exhibit higher fidelity with respect to aspects of the Derecho.

The relatively poor simulation performance for each of the ensemble members is consistent with literature summarized above regarding the specific challenge that this event presented. However, it also raised concerns regarding a possible issue with the stability of the cloud-based computational platform. Thus, simulations of two of the ensemble members were repeated on a separate computational platform (the U.S. Department of Energy NERSC Cori Cray XC40) and with a different compiler (Intel). Bit-wise reproducibility is not expected due to previously documented system architecture and compiler dependence of WRF simulations (Hacker et al., 2017; Li et al., 2016). Thus, these simulations are designed to evaluate whether use of a different system architecture and compiler yields marked improvements in terms of the fidelity with which the Derecho is simulated and to evaluate if the response to turning on the hail flag in the Morrison scheme is consistent. The results of these additional simulations are summarized in Figure 4 in terms of the time series of the number of grid cells with high cREF and in Figure 5 in terms of the cREF spatial patterns at $t_p$. These and other diagnostics indicate a high degree of similarity between the output of these simulations and the original ensemble members. Our inference is that the ensemble members generated by WRF compiled using GNU-Fortran are reliable.

### 3.2  Linking fidelity to metrics of CAPE, downbursts, and cold pool generation

As described above there is considerable spread among the ensemble members in terms of their fidelity relative to remote sensing and in situ observations. Here we seek to link model skill in reproducing aspects of derecho intensity (maximum wind gust, precipitation, and spatial coverage of cREF > 40dBZ) to metrics of convective potential specifically; MU-CAPE and wind shear between the ground and 6 km, plus metrics of convective intensity, specifically; indices of cold pool intensity, vertical velocities and cloud top height. We begin by describing the magnitudes and spatial variability of the diagnostic metrics in each ensemble

member.

MU-CAPE presented herein from rawinsonde observations is derived using the SHARPpy software (Blumberg et al., 2017) and is defined slightly differently than in codes available from the wrf-python Github page (https://doi.org/10.5065/D6W094P1), in that it is the parcel with the maximum equivalent potential temperature in the lowest 400 mb. Thus, the values are not directly comparable. Nevertheless, high values are indicative of presence of significant CAPE. Consistent with past summaries of the

environment in which the derecho was manifest, rawinsonde data from the two stations (KIAD (38.968N, -77.369E) and KWAL (38.018N and -75.236E)) within domain d03 indicate MU-CAPE values at $t_p$-3 (from RADAR) (i.e. 0000 UTC 30 June) of 6871 J/kg and 4735 J/kg (Figure S2). The surface to 6 km shear at that time are 17.2 m/s and 11.5 m/s respectively, which is consistent with the relatively weak shear evident in the WRF ensemble members (Figure S3). MU-CAPE at KIAD and KWAL dropped to 51 and 60 J/kg, respectively in the 1200 UTC 30 June sounding. This further emphasizes the profound underestimation of CAPE

consumption in the WRF ensemble during the passage of the derecho.

Consistent with estimates of parcel CAPE from rawinsonde soundings for this event and modeling of other derechos (Gatzen, 2004; Coniglio et al., 2011; Celiński-Mysław and Matuszko, 2014; Weisman et al., 2013), all of the ensemble members indicate substantial MU-CAPE leading up to and at $t_p$ (Figure 13, see also enlarged panels and timeseries of MU-CAPE in SM Figure S2, S4, S5, and S6). All have MU-CAPE above 4000 Jkg$^{-1}$ over a substantial fraction of domain d03 at $t_p$-3 (recall $t_p$ is defined

independently for all ensemble members). In some ensemble members, the boundary of the region of deep convection is clearly visible in MU-CAPE < 1000 J kg$^{-1}$ at $t_p$ over the western edge of domain 3.

There is also notable variability between ensemble members in terms of the magnitude of the vertical wind shear (0-6 km, S6 (see definition in section 2.3)) at $t_p$ (Figure 13, see also SM Figure S3). Highest shear (of up to 38 ms$^{-1}$ over this layer, or 0.006 s$^{-1}$) is shown in the Milbrandt-628, Milbrandt-628, Milbrandt-626-ERA-I and Milbrandt-628-ERA-I simulations. These values are

on the upper end of observational estimates for derecho events over the contiguous US between 1988-1993 (Evans and Doswell, 2001). The nudged ensemble members, plus Morrison+Hail and NSSL indicate relatively low shear.

The degree to which MU-CAPE decreases by $t_p$+3 varies considerably across the ensemble members (SM Figures S2, S4, and S5 and Table 5). The change in 50$^{th}$ percentile MU-CAPE values across domain d03 ranges from ~ 0 in the ensemble members

NSSL and Thompson to $\geq$ 900 Jkg$^{-1}$ in ensemble members Morrison, Milbrandt-628, Milbrandt-628, Milbrandt-626-ERA-I and

Milbrandt-628-ERA-I. Indeed, the change in median MU-CAPE is ~2000 Jkg$^{-1}$ in the Milbrandt-626-ERA-I and Milbrandt-628-ERA-I ensemble members that also showed highest agreement with observations of the spatial extent of high cREF, total precipitation accumulation, maximum wind gusts and large hail (Table 5). Other metrics that describe convective intensity that are diagnosed at $t_p$ also indicate substantial variability across the ensemble members. Modeled vertical velocity at/close to 5 km height at $t_p$ are highest in the Goddard, Morrison, Milbrandt-626-ERA-I and Milbrandt-628-ERA-I ensemble members (Figure 13, see also SM Figure S7) which also show substantial coverage of upward velocities in excess of 3 ms$^{-1}$ and also proximal regions with

substantial downdrafts of greater than 3 ms$^{-1}$. This is manifest as high values of the standard deviation of vertical velocities within 50 km of grid cells with cREF > 40 dBZ (Table 5). Goddard, Morrison, Milbrandt-626-ERA-I and Milbrandt-628-ERA-I are also the ensemble members with highest maximum near-surface wind speeds (Figure 10 and Table 5). The estimate of cloud top height derived using a threshold of base reflectivity from each model layer ranges from a low of 9 km (Morrison+Hail) to over 13.5 km

in all ensemble members that employ the Milbrandt microphysics schemes and that were not subject to nudging (Table 5).

Cold pool intensity as measured by the highest 5-percent of sea-level pressure anomalies (95[th] percentile SLP) and lowest 5-percent of temperature anomalies (i.e. 95[th] percentile negative temperature perturbations) also exhibit substantial variability between ensemble members. This is consistent with previous research that has examined microphysics scheme spread and its associated impact on cold pool properties and dynamics (Xue et al., 2017). The lowest 5-percent temperature deviations vary from

-1.38 to -5.58 K (Table 5 and example fields shown in Figure 14 for the Morrison and Milbrandt-628-ERA-I ensemble members). The upper end of this range is thus consistent with the cold pool intensities from the experiment study of Derechos from Oklahoma that indicated maximum (point) temperature anomalies of 5.4 to 9.5 K (Engerer et al., 2008). Four ensemble members (Goddard, Milbrandt-628, and the two simulations within ERA-Interim LBC) also exhibit 95[th] percentile SLP deviations of above 2 hPa (Table 5 and example fields shown in Figure 14). While it is challenging to evaluate the simulation of these cold pools due to the

limited spatial coverage of the ASOS network, the range of SLP and near-surface temperature anomalies from these ensemble members is broadly consistent with those calculated from the ASOS observations. The estimate of cloud top height derived using a threshold of base reflectivity from each model layer ranges from a low of 9 km (Morrison+Hail) to over 13.5 km in all ensemble members that employ the Milbrandt microphysics schemes and that were not subject to nudging (Table 5).

The Spearman correlation coefficients (r) between the three metrics of model fidelity from this 15 member ensemble are > 0.9

indicating that a simulation that exhibits atypically high skill with respect to maximum wind speed is also likely to perform well in describing the spatial extent of high cREF and accumulated precipitation (Table 5). The storm intensity metrics all also exhibit positive r but of varying magnitude. For example, there is only a weak association between the rank correlation of cloud top height and vertical velocities (r < 0.38).

Simulated wind gusts at the surface are a product of downdrafts/downbursts and resulting gust fronts. Accordingly, the highest

5% of downward vertical velocities exhibits a Spearman correlation coefficient (r) with the ratio of modeled to observed maximum

wind gusts of 0.90 (Table 5). All ensemble members that exhibit higher maximum wind gust ratios also exhibit stronger downdrafts

(exhibit largest negative vertical velocity), stronger vertical wind shear and show higher median MU-CAPE change. Consistent

with past research that examined ensemble spread for simulated squall lines from use of different microphysics schemes (Morrison

et al., 2015; Xue et al., 2017), the two cold pool metrics are also shown to be predictive of model fidelity for wind gusts associated

with the derecho. That is, models that generate the strongest cold-pools (as measured by either the near-surface temperature or

pressure anomalies) tend to be those that perform best in terms of the associated near-surface wind gusts (r across the 15 members

is 0.72-0.75, see Table 5). Metrics of cold pool dynamics are also predictive of other aspects of simulation fidelity (e.g. extent of

cREF) consistent with their importance for the triggering and organization of persistent convection.

Although the two ensemble members that exhibit highest fidelity with respect to the areal coverage of cREF (Morrison and

Milbrandt-628-ERA-I) also exhibit relatively high skill in reproducing precipitation and wind gusts (as shown previously), as

illustrated by Figure 14, these simulations generate different morphologies of the derecho. Specifically, the region of high cREF

is much more spatially homogeneous in Morrison than Milbrandt-628-ERA-I. Further, the cold pool intensity at $t_p$ exhibited

important differences. The region of elevated SLP is much more marked in Morrison but the associated temperature anomaly is

much smaller than that from Milbrandt-628-ERA-I, this may be linked to the lower elevation of downdraft maximum intensity in

the Morrison ensemble member (Table 5).

In those ensemble members that perform comparatively poorly in terms of reproducing key aspects of the derecho (e.g.

Morrison+Hail, NSSL, Thompson and both nudged simulations), MU-CAPE is not consumed in sufficient amounts resulting in

under-production of deep convection during the derecho (Table 5). This leaves excess MU-CAPE availability for the subsequent

frontal passage resulting in excess production of convective cells, wind gusts, cREF > 40dBZ and precipitation (Figure 6, 8 and

12). This may have implications for climate-scale (long-term) simulations from CPRCM and specifically inference regarding

temporal sequencing of deep convection and associated hazards such as flooding.

## 4    Summary and conclusions

Severe wind gusts associated with derechos represent an important natural hazard resulting from MCSs. Efforts to improve

simulations of deep convection in both weather forecasting and climate projections have been hampered by both conceptual gaps

in understanding of small scale cloud processes, lack of observations both of the associated hazards and hydrometeor properties

on the microscale (Morrison et al., 2020) and challenges in representing scale linkages in numerical models. Additionally, advanced

schemes tend to be computationally expensive (Xue et al., 2017) which may limit their utility in CPRCM simulations. Accordingly,

while a limited number of studies have sought to examine how severe convective wind environments might change in the future (Brown and Dowdy, 2021a), very few robust hindcast ensemble simulations exist for specific events that can be leveraged in a pseudo-global warming framework. Evaluating the inherent ability of models to reproduce key aspects of historic, poorly forecasted severe events will facilitate the further development of model parameterization schemes, allow selection of optimal model configuration for simulating high impact events (Dai et al., 2021) and provide context for examining how such events might change in the future.

Revisiting the main objectives of this work, we sought to evaluate an ensemble of simulations with WRF that differ in terms of the microphysics schemes applied, start date, the lateral boundary conditions, Fortran compiler and use of nudging. The main findings of this study are:

1.  This 15 member WRF ensemble tends to underestimate the spatial extent of high composite reflectivity, near-surface wind speed and precipitation during the Derecho period and overestimate cREF, wind speed and precipitation during a subsequent frontal passage. The bias with respect to the subsequent front is linked to a negative bias in MU-CAPE depletion during the derecho. The use of a double-moment, 7-class scheme with number concentrations for all species (including hail and graupel) [Milbrandt-Yau] results in the greatest model fidelity for maximum wind speeds, hail, and precipitation accumulation. This is consistent with numerous studies that have shown increased fidelity when using double-moment, bulk microphysics schemes with number concentrations for ice, graupel, and hail (Morrison et al., 2015).

2.  Model settings such as initialization time and LBC exhibit a strong signal in driving different convective conditions and results in large spread of the associated natural hazards; wind gusts and hail. The ensemble spread from changing the microphysics scheme and the resulting simulated dynamic and thermodynamic convective structures (Xue et al., 2017) is similar to that caused by changing the lateral boundary conditions. The higher fidelity associated with use of ERA-Interim reanalysis data as opposed to ERA5 is unexpected. Nested simulations of a European derecho event using the COSMO regional model found significant improvement in the simulation fidelity with use of ERA5 for the LBC over simulations using ERA-Interim. Our finding has important implications for construction of hindcast simulations for use in Surrogate or Pseudo Global Warming (PGW) numerical experiments to quantify the potential of global warming on extreme weather events using regional models (Li et al., 2019; Kröner et al., 2017; Haberlie and Ashley, 2019; Liu et al., 2017). In such simulations an historically important extreme event/period/season is first simulated using contemporary LBC and then the simulation is repeated using LBC and IC perturbed to represent the change in, for example, air temperatures and water vapor availability (Kröner et al., 2017). The difference in these two realizations is interpreted as the impact of global climate non-stationarity. A previous analysis over CONUS used ERA-Interim LBC and shifted the atmospheric profile by ± 5 °C. They found increases in both CAPE and convective inhibition, which implies shift the convective population

(Rasmussen et al., 2020). Our work indicates use of ERA5 for IC and LBC may not always result in high-fidelity baseline simulations of extreme convective events in the contemporary climate. These simulation deficiencies may render evaluation of the PGW response highly uncertain. Additional simulations using ERA5 and ERA-Interim are required before generalizable conclusions can be made about which dataset provides better boundary conditions. The relatively low skill of the 15 WRF ensemble members for this derecho, and the improvement in model skill for the simulations initialized at a later time stamp (28 June versus 26 June) is consistent with past research that has indicated forecast errors in the simulation of deep convection have a doubling time of only a few hours (Prein et al., 2015). This represents an important challenge for simulations of these atmospheric hazards.

3.  The diagnostic metrics applied here to represent pre-conditioning of the environment plus key dynamic and thermodynamic aspects of the storm (development and propagation of squall lines, downbursts and cold pool development) are highly predictive of the relative skill of individual model ensemble members. This seems to imply that although the ensemble members incompletely resolve key outcomes of the derecho (e.g. the intensity of the wind gusts), their relative ability in terms of the associated dynamics appears to indicate the better performing ensemble members are generating 'the right answers for the right reasons'.

Due to the computational demand, a spectral bin microphysics scheme was not used here, even though such schemes have been shown to outperform double moment bulk schemes in a weather forecasting context (Xue et al., 2017; Fan et al., 2017). Future work in the field of model fidelity and scheme sensitivity that examines historically significant weather events would benefit from even larger ensembles and, as computing developments allow, the use of more conceptually realistic spectral bin microphysics parameterization schemes.

**Code availability**

The WRF code version used in this study (v3.8.1) is available from:

https://www2.mmm.ucar.edu/wrf/users/download/get_sources.html

For WRF releases beginning with version 4 and above:

https://www2.mmm.ucar.edu/wrf/users/download/get_sources_new.php

**Data availability**

ERA-Interim output is available for download from **http://apps.ecmwf.int/datasets/**. The NOAA-NCEP real-time global sea surface temperature analyses are available from http:// **www.nco.ncep.noaa.gov/pmb/products/sst/**. ERA5 output is available for download from **https://cds.climate.copernicus.eu/cdsapp#!/home**. NEXRAD RADAR data, including all products used in the

current study are available from the National Climatic Data Center (https://www.ncdc.noaa.gov/data-access/radar-data). NWS ASOS data are available from ftp://ftp.ncdc.noaa.gov/pub/data/asos-fivemin/. The NOAA Storm Events Database is available at https://www.ncdc.noaa.gov/stormevents/. Stage IV precipitation data (which combines RADAR and rain gauge measurements). NCEP/EMC 4KM Gridded Data (GRIB) Stage IV Data (Du, 2011) were downloaded from https://data.eol.ucar.edu/dataset/21.093

in GRIB format and converted for processing to netCDF using the NCL command 'ncl convert2nc'. Hourly precipitation amounts were summed for the entire duration of the Derecho period. All model output used in the analyses presented here, including a sample namelist is available at:

http://portal.nersc.gov/archive/home/projects/m2645/www/public_data_derecho_case

**Author contributions**

SCP and TJS conceived the research and developed the experimental design with input from RJB. TJS performed the simulations. FL, TJS and SCP performed the analyses, and prepared the figures/tables. SCP and TJS developed the initial manuscript. All authors contributed to the final manuscript.

**Competing interests**

The authors declare that they have no conflict of interest.

**Acknowledgements**

This research is supported by the US Department of Energy (DE-SC0016438 and DE-SC0016605). Computational resources are provided by the NSF Extreme Science and Engineering Discovery Environment (XSEDE) (award TG-ATM170024) and the National Energy Research Scientific Computing Center, a DoE Office of Science User Facility supported by the Office of Science of the U.S. Department of Energy under Contract No. DE-AC02-05CH11231. The authors thank Hugh Morrison for his insightful

comments regarding his microphysics scheme, and the two anonymous reviewers for their detailed comments.

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

**Table 1: WRF simulation domains (see also Figure 1) and physics settings (see also Table 2).**

| | |
|---|---|
| Domain settings | |
| Horizontal resolution and domain size (d01, d02, d03) | 12 km (175 × 175 cells), 4 km (262 × 262 cells), 1.33 km (295 × 295 cells) |
| Vertical resolution | 41 vertical levels up to 50 hPa |
| Model time step (d01, d02, d03) | 30 sec, 10 sec, 3.33 sec |
| Model physics settings | |
| Microphysics | Varies – see Table 2 |
| Longwave radiation | RRTM (Mlawer et al., 1997) |
| Shortwave radiation | Dudhia (Dudhia, 1989) |
| Time between radiation calls | 10 minutes |
| Surface layer | Revised MM5 Monin-Obukhov scheme (Jiménez et al., 2012) |
| Land surface | Noah land surface model (Niu et al., 2011) |
| Number of soil layers | 4 |
| Number of land categories | 21 (MODIS) |
| Planetary boundary layer | MYNN level 2.5 (called every time step) (Nakanishi and Niino, 2006) |
| Cumulus parameterization | Kain-Fritsch (outer domain only, called every 5 minutes) (Kain and Fritsch, 1993; Kain, 2004) |
| Grid nudging settings (for nudged runs only) | |
| Time interval between analysis times | 360 minutes |
| Time to stop nudging after start of simulation | 144 hours |
| Calculation frequency for analysis nudging | Every time step |
| Model level nudged down to | Level 20 |
| Nudging coefficient for wind components (u,v), temperature, water vapor mixing ratio | 0.0003 sec$^{-1}$ |
| Time for ramping function | 60 minutes, ramping starts at last analysis time |
| Upper-level damping settings (for 65 vertical levels runs only) | |
| damp_opt | 3 = Rayleigh damping. dampcoef inverse time scale [1/s], e.g. 0.2; for real-data cases |


**Table 2: Overview of WRF ensemble simulation members. Note: Single-moment schemes treat only the mass of the specified variables, while double-moment schemes also treat the number concentrations of some/all of the specified variables (if only some they are denoted by *).**

| Microphysics scheme (WRF namelist #) | Name used herein (i.e. ensemble member) | Reference | Scheme characteristics | Start date & time (UTC) | LBC | Nudged |
|---|---|---|---|---|---|---|
| Goddard (7) | Goddard | | Single-moment. Cloud, rain, ice, snow, graupel & hail (if gsfcgce_hail switched is used) | 2012/06/26 00:00 | ERA5 | None |
| Morrison (10) | Morrison | | Double-moment. Cloud, rain*, ice*, snow* & graupel* | | | |
| | Morrison+Hail | | | | | |
| | Morrison Intel | | | | | |
| | Morrison Intel+Hail | | | | | |
| | Morrison 65levels | | | | | |
| | Morrison 65levels+Hail | | | | | |
| Thompson (8) | Thompson | (Thompson et al., 2008) | Double-moment. Cloud, rain*, ice*, snow & graupel | | | |
| NSSL (17) | NSSL | (Mansell et al., 2010b) | Double-moment Cloud, rain, ice, snow, graupel & hail | | | |
| Milbrandt-Yau (9) | Milbrandt-626 | (Milbrandt and Yau, 2005a) | Double-moment. Cloud, rain, ice, snow, graupel & hail. | | | |
| | Milbrandt-628 | | | 2012/06/28 00:00 | | |
| | Milbrandt-626-ERA-I | | | 2012/06/26 00:00 | ERA-Interim | |
| | Milbrandt-628-ERA-I | | | 2012/06/28 00:00 | | |
| | Nudged-ERA5 | | | 2012/06/26 00:00 | ERA5 | Down to level 20 (see Table 1) |
| | Nudged-ERA-I | | | 2012/06/26 00:00 | ERA-Interim | Down to level 20 (see Table 1) |


**Table 3: Spearman rank correlations for the spatial fields of maximum wind gusts in domain d03 during the derecho (Derecho period: 29-Jun-2012 21:30:00 to 30-Jun-2012 13:30:00) from WRF and ASOS observations. In this analysis WRF output for maximum time step wind speeds (dt = 6 sec) is sampled at the 34 ASOS locations and compared with the maximum 3-second ASOS wind gusts measurements (see spatial fields in Figure 11). Also shown are the Spearman rank correlations between spatial fields of total accumulated precipitation from WRF output relative to RADAR estimates and ASOS in situ measurements. In these analyses the correlations between WRF and the RADAR data are for all WRF grid cells sampled by the RADAR (99.4% of d03), while the comparison with ASOS measurements is for the 34 ASOS stations. The final column shows the correlations between the spatial fields of maximum composite reflectivity cREF (again in any time step during the Derecho period) from the WRF ensemble members and RADAR.**

| Ensemble member | ASOS | | RADAR | |
|---|---|---|---|---|
| | Wind gusts | Precipitation | Precipitation | cREF |
| Goddard | 0.127 | 0.343 | 0.319 | 0.279 |
| Morrison | 0.312 | 0.063 | 0.187 | 0.199 |
| Morrison+Hail | -0.557 | 0.138 | 0.181 | 0.255 |
| Morrison Intel | 0.4408 | 0.0481 | 0.2079 | 0.2430 |
| Morrison Intel+Hail | -0.3692 | 0.0961 | 0.2727 | 0.3134 |
| Morrison 65levels | 0.3418 | 0.0059 | 0.1591 | 0.2054 |
| Morrison 65levels+Hail | 0.2030 | 0.1522 | 0.2604 | 0.2561 |
| Thompson | -0.414 | -0.018 | 0.239 | 0.278 |
| NSSL | -0.482 | 0.126 | 0.119 | 0.134 |
| Milbrandt-626 | -0.412 | 0.429 | 0.351 | 0.152 |
| Milbrandt-628 | 0.318 | 0.179 | 0.299 | 0.213 |
| Milbrandt-626-ERA-I | 0.225 | 0.142 | 0.394 | 0.227 |
| Milbrandt-628-ERA-I | 0.669 | -0.179 | 0.174 | 0.250 |
| Nudged-ERA-I | -0.800 | -0.148 | 0.128 | 0.224 |
| Nudged-ERA5 | -0.410 | 0.017 | 0.140 | -0.053 |

**Table 4: Number of grid cells in domain d03 where hail is indicated by the RADARs or present in the WRF simulations during the derecho (Derecho period: 29-Jun-2012 21:30:00 to 30-Jun-2012 13:30:00) and the frontal passage (Front period: 30-Jun-2012 15:20:00 to 01-Jul-2012 14:50:00). Also shown is the number of grid cells with Maximum Estimated Size of Hail (MESH) above 25 mm from the RADAR or WRF. Recall: RADAR detection of hail is re-gridded onto the WRF grid used for domain d03 prior to use in the model evaluation.**


| | # Grid cells with hail | | # Grid cells with hail values > threshold | |
|---|---|---|---|---|
| | Derecho | Front | Derecho | Front |
| RADAR | 3078 | 2152 | 824 | 813 |
| Ensemble member | | | | |
| Goddard | 0 | 10 | 0 | 6 |
| Morrison | 0 | 24 | 0 | 0 |
| Morrison+Hail | 3000 | 74398 | 0 | 0 |
| Morrison Intel | 0 | 14 | 0 | 0 |
| Morrison Intel+Hail | 7002 | 74192 | 0 | 0 |
| Morrison 65levels | 2 | 36 | 0 | 0 |
| Morrison 65levels+Hail | 37030 | 72997 | 1 | 75 |
| Thompson | 10 | 8996 | 2 | 4909 |
| NSSL | 7446 | 79890 | 135 | 5907 |
| Milbrandt-626 | 16368 | 78276 | 167 | 5687 |
| Milbrandt-628 | 26183 | 77415 | 436 | 6461 |
| Milbrandt-626-ERA-I | 54406 | 68899 | 782 | 4928 |
| Milbrandt-628-ERA-I | 63695 | 67671 | 568 | 4028 |
| Nudged-ERA5 | 2428 | 37913 | 21 | 1226 |
| Nudged-ERA-I | 195 | 37692 | 0 | 2071 |

**Table 5:** Metrics of simulation fidelity relative to observations, and convection metrics derived from output from each WRF member during the period of the derecho passage (Derecho period: 29-Jun-2012 21:30:00 to 30-Jun-2012 13:30:00). The metrics of simulation fidelity are described in section 2.2 and are as follows: The Max Gust Ratio: the ratio of the maximum wind gust in any land grid cell from WRF output and observations at the ASOS stations. Total Precip. Ratio: the ratio of the spatial mean total accumulated precipitation from WRF to RADAR and STAGE IV, respectively, for any grid cell with common coverage. cREF>40 dBZ: the ratio of the spatial extent of grid cells with cREF above 40 dBZ at the peak coverage in WRF and RADAR. The lower portion of the table shows the Spearman rank correlation coefficients for the 15 values of each metric (one for each ensemble member). This analysis thus shows the degree to which an ensemble member that exhibit high values of a given metric also generates high values of a second metric. The color-coding used in this table is as follows; for the measures of simulation fidelity table cells colored red have low fidelity, and those indicated by cyan exhibit relatively high fidelity. For all other cells in the table, a background of orange indicates low values, while blue indicates comparatively high values. The saturation of the color indicates relative ordering of the values. The definitions of each convection metric are given in section 2.3.

| | Simulation Fidelity | | | | Convection Metric | | | | | |
|---|---|---|---|---|---|---|---|---|---|---|
| | Max Gust Ratio | Total Precip. Ratio (RADAR) | Total Precip. Ratio (Stage IV) | cREF>40 dBZ Ratio | 95% Temperature deviation [-K] | 95% SLP deviation [hPa] | Median CAPE loss [J kg$^{-1}$] | 95% -W [ms$^{-1}$] | Max std(w) height [km] | $Z_{R20}$ [km] |
| Goddard | 0.610 | 0.206 | 0.218 | 0.346 | 4.230 | 2.1 | 876 | 0.157 | 8.0 | 12.9 |
| Morrison | 0.673 | 0.413 | 0.435 | 0.788 | 3.29 | 1.850 | 1532 | 0.151 | 8.0 | 15.3 |
| Morrison +Hail | 0.460 | 0.016 | 0.017 | 0.102 | 2.12 | -0.756 | 175 | 0.117 | 8.0 | 9.0 |
| Morrison Intel | 0.687 | 0.390 | 0.411 | 0.806 | 3.96 | 4.108 | 1474 | 0.161 | 8.5 | 15.1 |
| Morrison Intel+Hail | 0.532 | 0.061 | 0.065 | 0.218 | 5.35 | 2.120 | 253 | 0.164 | 6.3 | 11.8 |
| Morrison 65levels | 0.693 | 0.454 | 0.479 | 0.972 | 4.35 | 2.726 | 1706 | 0.263 | 8.0 | 13.9 |
| Morrison 65levels+Hail | 0.820 | 0.508 | 0.536 | 0.850 | 4.47 | -0.193 | 1620 | 0.238 | 8.5 | 13.1 |
| Thompson | 0.269 | 0.006 | 0.006 | 0.044 | 1.97 | 0.478 | 61 | 0.059 | 5.6 | 12.7 |
| NSSL | 0.334 | 0.015 | 0.015 | 0.043 | 3.27 | 0.238 | -42 | 0.069 | 6.4 | 12.8 |
| Milbrandt-626 | 0.449 | 0.061 | 0.064 | 0.269 | 3.30 | 0.596 | 963 | 0.092 | 7.1 | 13.8 |
| Milbrandt-628 | 0.575 | 0.185 | 0.195 | 0.391 | 4.64 | 3.380 | 1428 | 0.093 | 8.9 | 13.7 |
| Milbrandt-626-ERA-I | 0.633 | 0.566 | 0.597 | 0.844 | 5.44 | 2.360 | 1960 | 0.152 | 8.9 | 14.4 |
| Milbrandt-628-ERA-I | 0.695 | 0.636 | 0.671 | 0.945 | 5.58 | 2.790 | 2030 | 0.146 | 7.1 | 14.9 |
| Nudged-ERA5 | 0.392 | 0.004 | 0.004 | 0.037 | 1.38 | -1.750 | 575 | 0.034 | 7.1 | 14.0 |
| Nudged-ERA-I | 0.226 | 0.000 | 0.000 | 0.004 | 4.05 | -1.220 | 182 | 0.017 | 5.6 | 11.6 |
| Spearman Rank Correlations | Max Gust Ratio | Total Precip. Ratio (RADAR) | Total Precip. Ratio (Stage IV) | cREF>40 dBZ Ratio | 95% Temperature deviation | 95% SLP deviation | Median CAPE Loss | 95% -W | Max std(w) height | $Z_{R20}$ |
| Max Gust Ratio | 1.00 | | | | | | | | | |
| Total Precip Ratio (RADAR) | 0.94 | 1.00 | | | | | | | | |

| | | | | | | | | | | |
|---|---|---|---|---|---|---|---|---|---|---|
| Total Precip Ratio (Stage IV) | 0.94 | 1.00 | 1.00 | | | | | | | |
| cREF>40 dBZ Ratio | 0.95 | 0.96 | 0.96 | 1.00 | | | | | | |
| 95% Temperature deviation | 0.60 | 0.71 | 0.71 | 0.65 | 1.00 | | | | | |
| 95% SLP deviation | 0.59 | 0.66 | 0.66 | 0.70 | 0.63 | 1.00 | | | | |
| Median CAPE Loss | 0.87 | 0.90 | 0.90 | 0.91 | 0.67 | 0.59 | 1.00 | | | |
| 95% -W | 0.83 | 0.76 | 0.76 | 0.79 | 0.56 | 0.55 | 0.59 | 1.00 | | |
| Max std(w) height | 0.74 | 0.68 | 0.68 | 0.72 | 0.39 | 0.51 | 0.67 | 0.64 | 1.00 | |
| ZR20 | 0.64 | 0.65 | 0.65 | 0.66 | 0.23 | 0.56 | 0.79 | 0.32 | 0.51 | 1.00 |

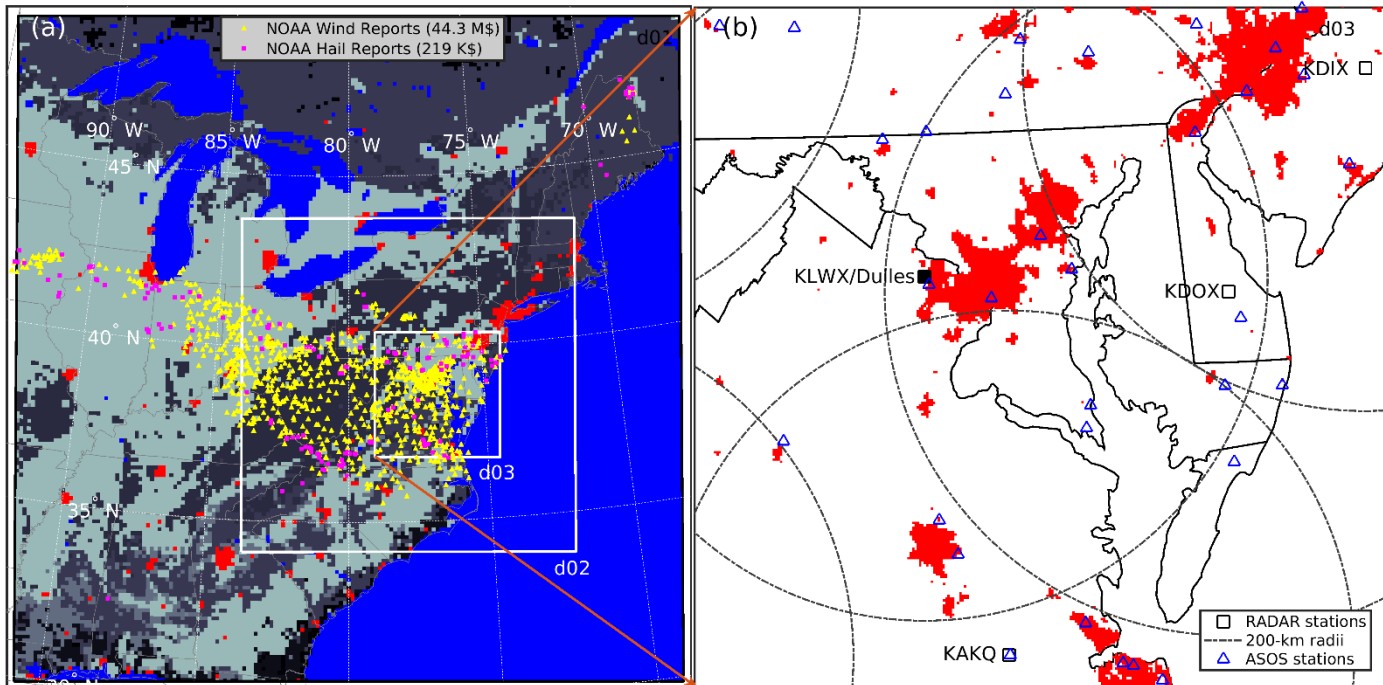


**Figure 1: a) Map of the WRF simulation domains; d01, d02, d03 with land use classes shown, where blue indicates water and red, developed areas (shown in both panels), different shades of grey denote different non-urban land use. The yellow markers indicate wind reports in the NOAA Storm Data publication during 29 and 30 June 2012, the magenta denotes reports of hail. The total amount of property damage in the NOAA Storm Data publication from wind and hail within the study area in this two-day period is given in the**

**upper legend of (a) and are 2012$ 44.3 million and 2012$ 291,000 respectively. b) Locations of RADAR (black squares, the black circles denote the 200 km radii from which data are presented here) and ASOS stations l (blue triangles) in d03. Dulles Airport is within 3 km of KLWX and both are denoted by the same filled, black square.**

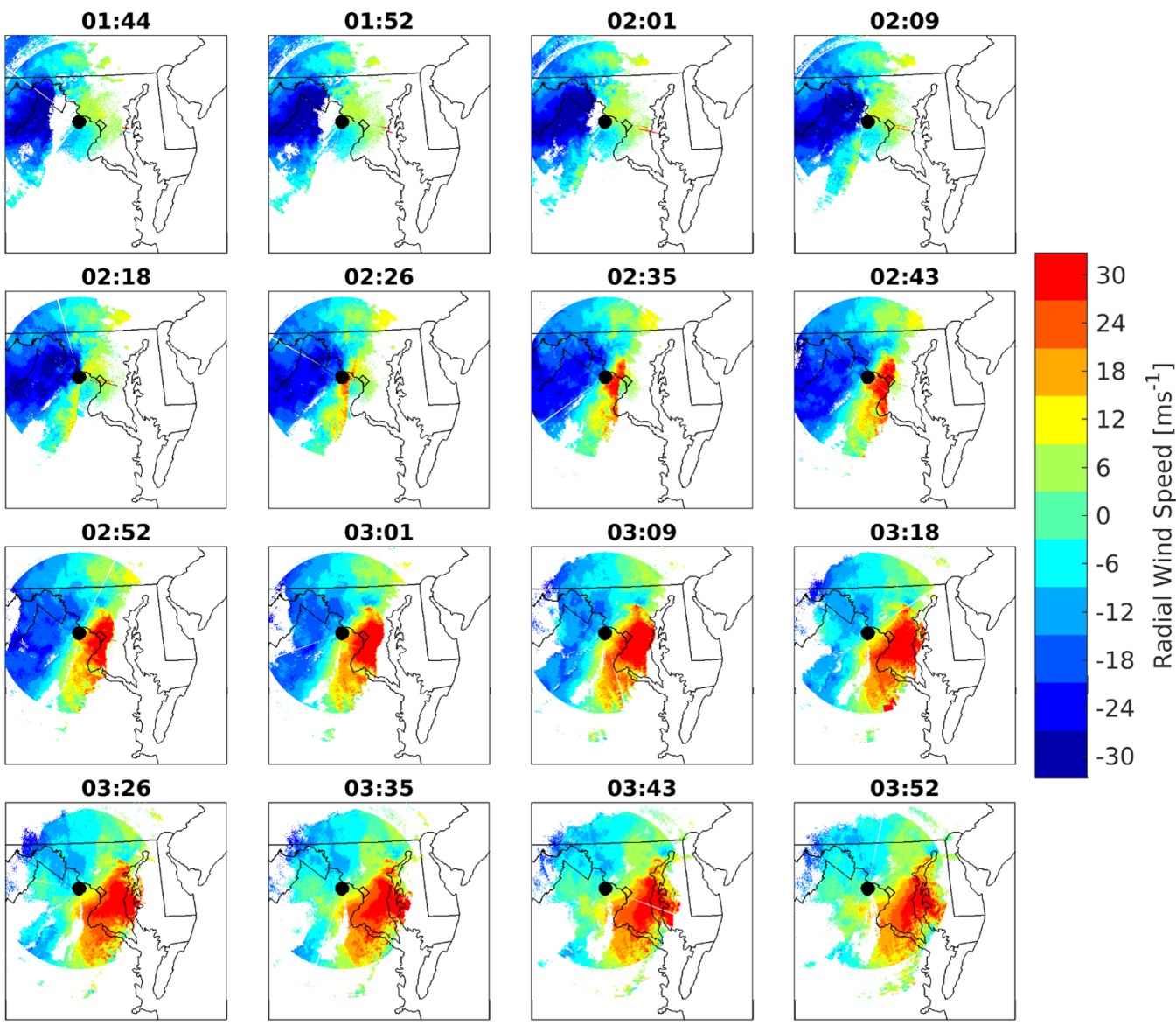

**Figure 2: Radial wind speeds from the NWS RADAR deployed at Sterling, VA (station code KLWX, shown as a black circle) the closest RADAR station to Washington Dulles airport, sampled for every second scan from the lowest elevation angle (0.5°) for about a 2-hour period surrounding the period of highest recorded wind speeds at the airport.**

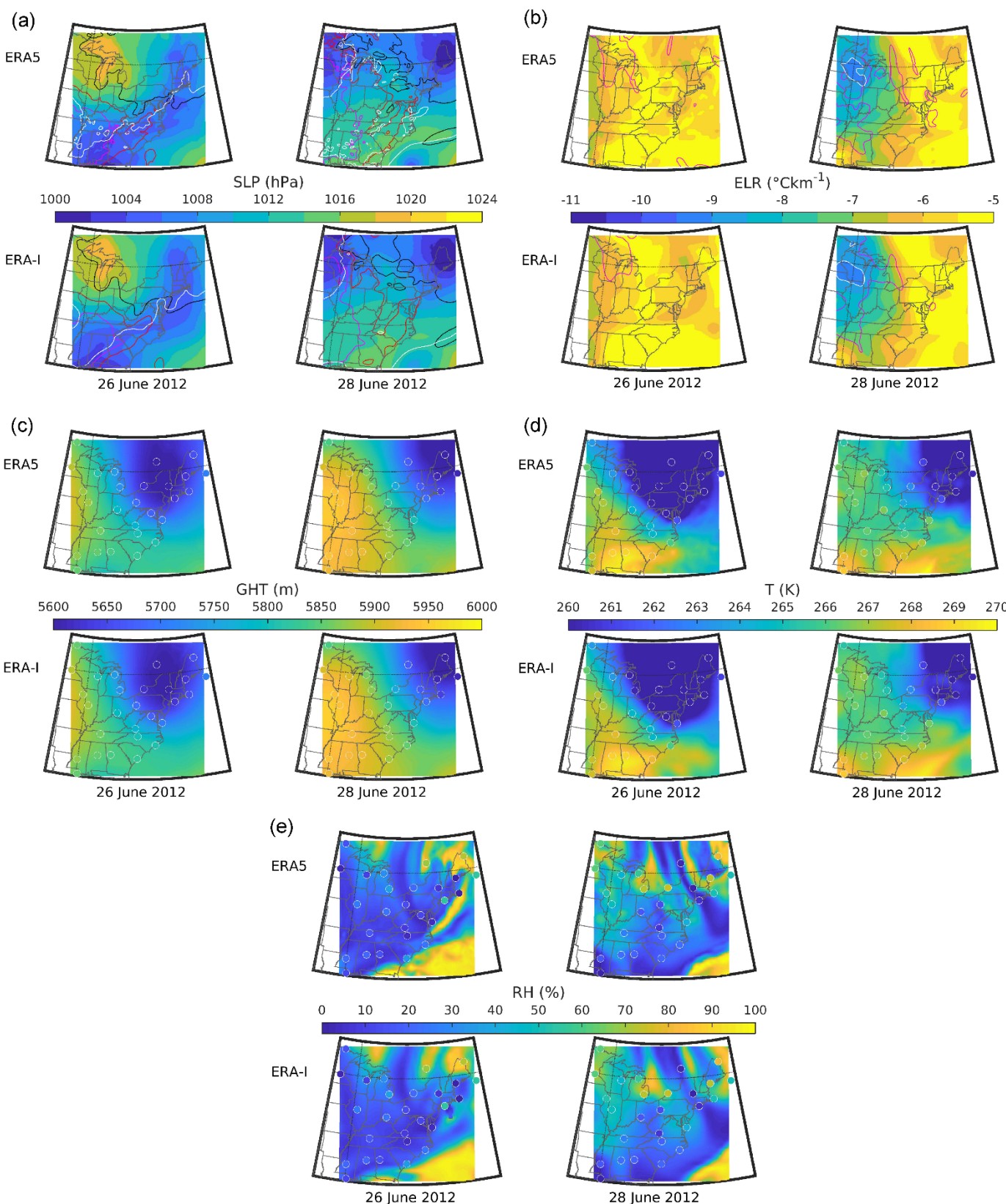

**Figure 3: (a) Spatial maps of sea level pressure (colored surface) generated by WRF real from the ERA5 and ERA-Interim reanalysis products used to provide the model LBC and initial conditions. The black, red, and magenta lines are 2-m temperature of 295 K, 300 K, and 305 K respectively. The white line represents specific humidity at 2-m of 12.5 g/kg. (b) Filled contours of lapse rates (700-500 hPa) with the -9°C/km highlighted by the white outline. Also shown by the magenta isoline is the area in which the RH increased by 20% over this layer. (c) 500 hPa geopotential height in meters. (d) 500 hPa temperature in Kelvin. (e) 500 hPa relative humidity in %. Plots in (c), (d), and (e) contain rawinsonde observations (filled circles). In all the plots, WRF real output is used from all 3 domains.**

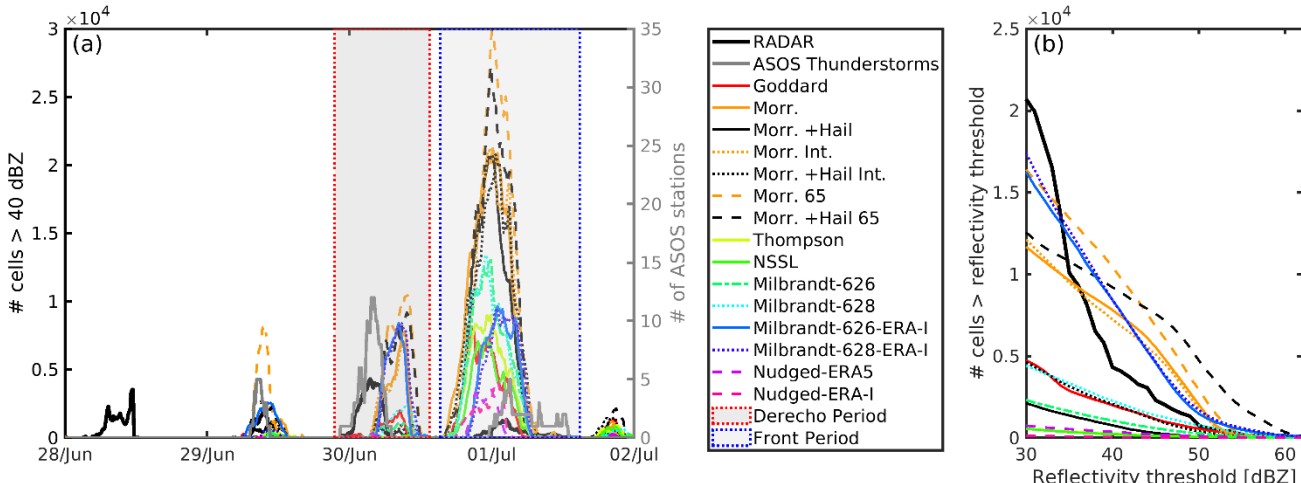

**Figure 4: (a) Time series of number of grid cells in domain d03 with composite reflectivity (cREF) > 40 dBZ from RADAR and the 15 WRF ensemble members. The number of the 34 ASOS stations in domain d03 reporting thunderstorms is shown in grey (right axis). The timing of the (Derecho period: 29-Jun-2012 21:30:00 to 30-Jun-2012 13:30:00) and the frontal passage (Front period: 30-Jun-2012 15:20:00 to 01-Jul-2012 14:50:00) are denoted by the grey backgrounds. (b) The number of grid cells in domain d03 where output from each WRF ensemble member or the RADARs exceeded the specified threshold during the time step within the derecho period when the maximum number of grid cells exceeded the threshold. For example, in the RADAR observations there is a single 10-minute period during which approximately 5000 grid cells exhibit a value above 40 dBZ.**


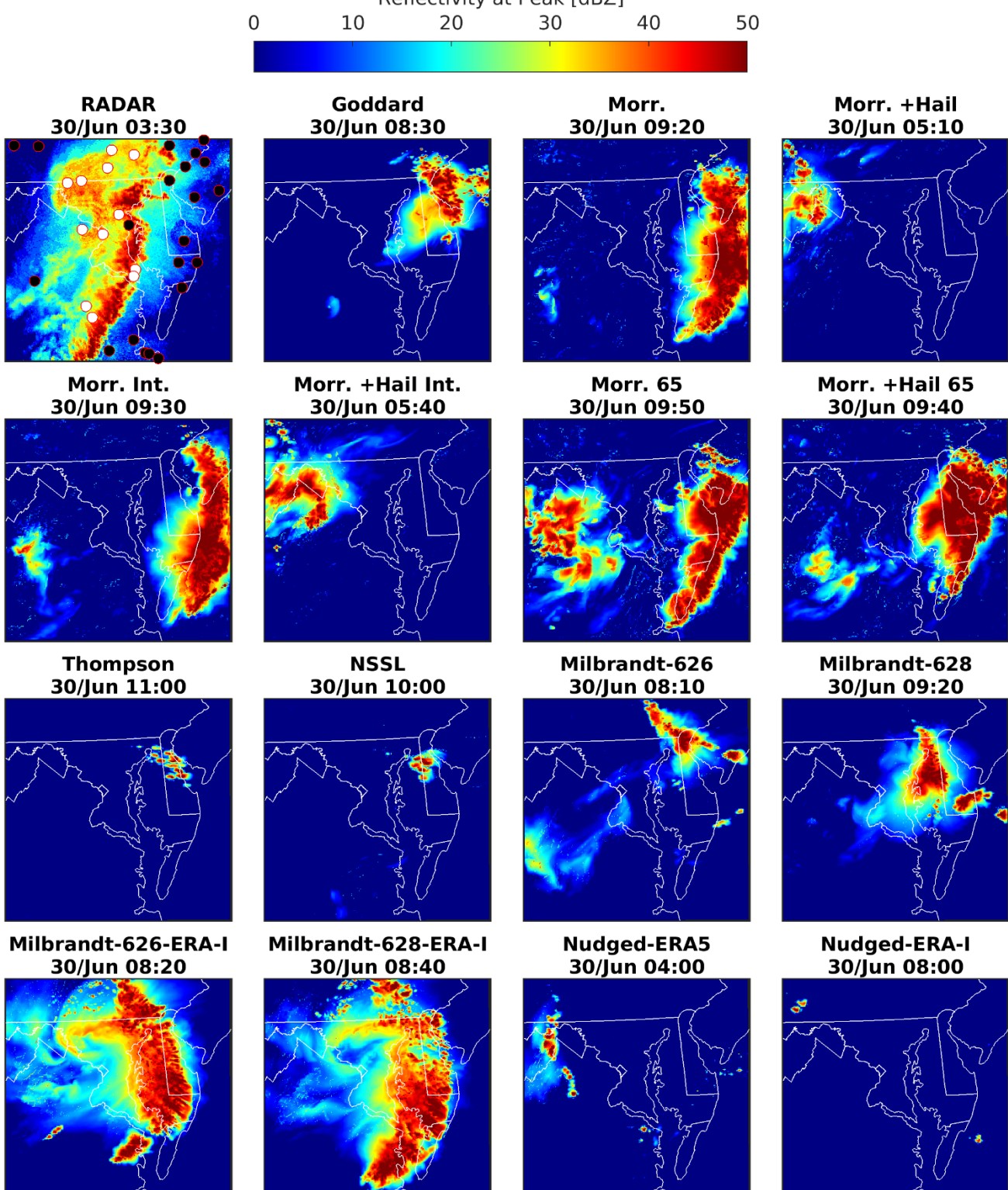

Figure 5: Composite reflectivity (cREF) in domain d03 at $t_p$ (the time when values from the maximum number of grid cells exceeded 40 dBZ) during the Derecho period from RADAR and each WRF ensemble member (times are noted in panel titles). The RADAR panel includes markers showing the presence (white) and absence (black) of thunderstorm reports from ASOS stations in domain d03 in the hour surrounding 03:30 UTC 30 June 2012. cREF > 40 dBZ varies widely across the WRF ensemble members but most are negatively biased relative to the RADAR. This bias is least marked in the Morrison-XXXX and Milbrandt-XXXX simulations (where XXXX indicates the ensemble member in terms of LBC/start date/compiler from simulations that employ the Morrison and Milbrandt microphysics scheme), especially those using ERA-Interim for initial and lateral boundary conditions.

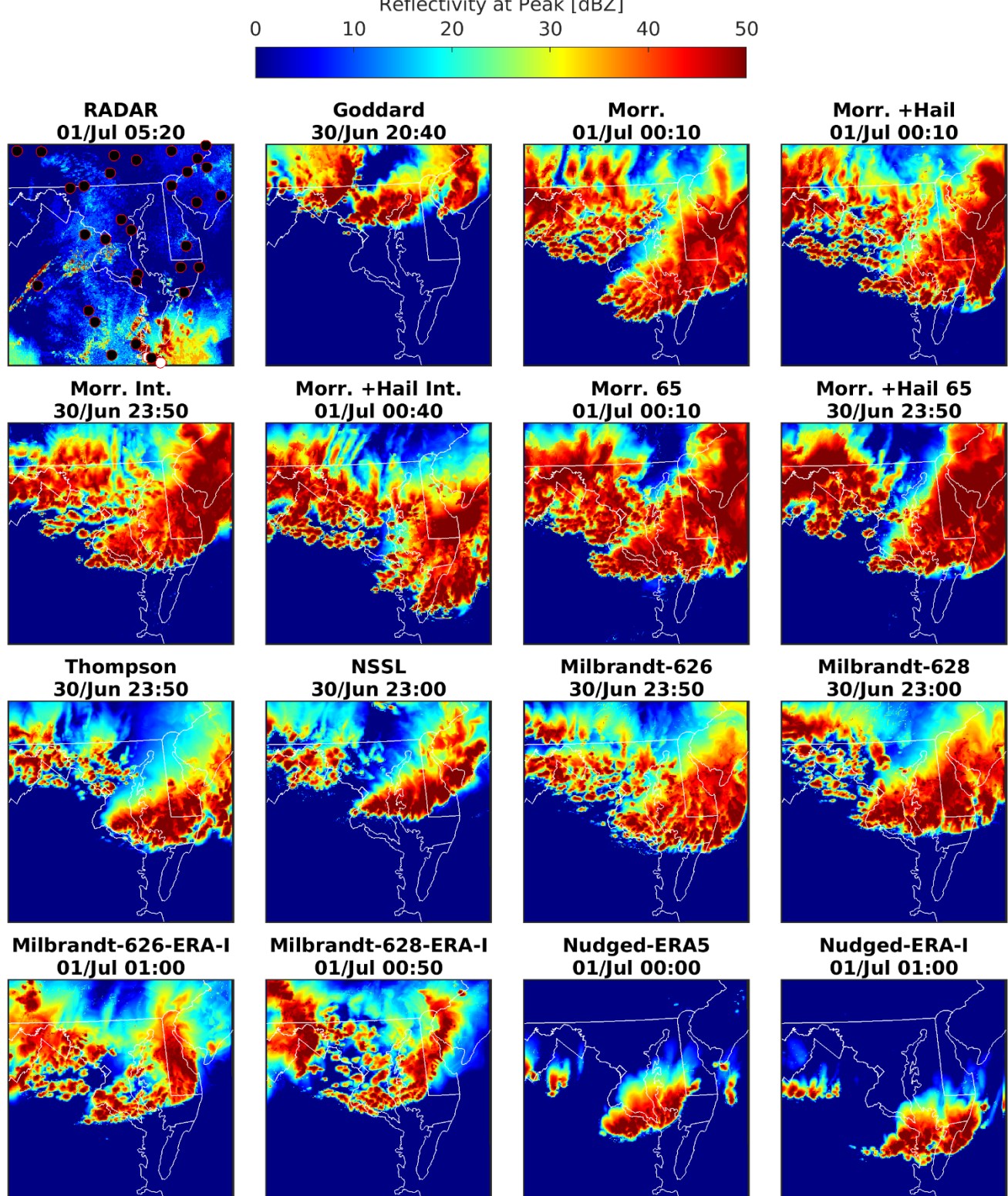

**Figure 6: Composite reflectivity (cREF) in domain d03 at $t_p$ (the time when values from the maximum number of grid cells exceeded 40 dBZ) during the Front period from RADAR and each WRF ensemble member (times are noted in panel titles). The RADAR panel includes markers showing the presence (white) and absence (black) of thunderstorm reports from ASOS stations in domain d03 in the hour surrounding 05:20 UTC 1 July 2012.**


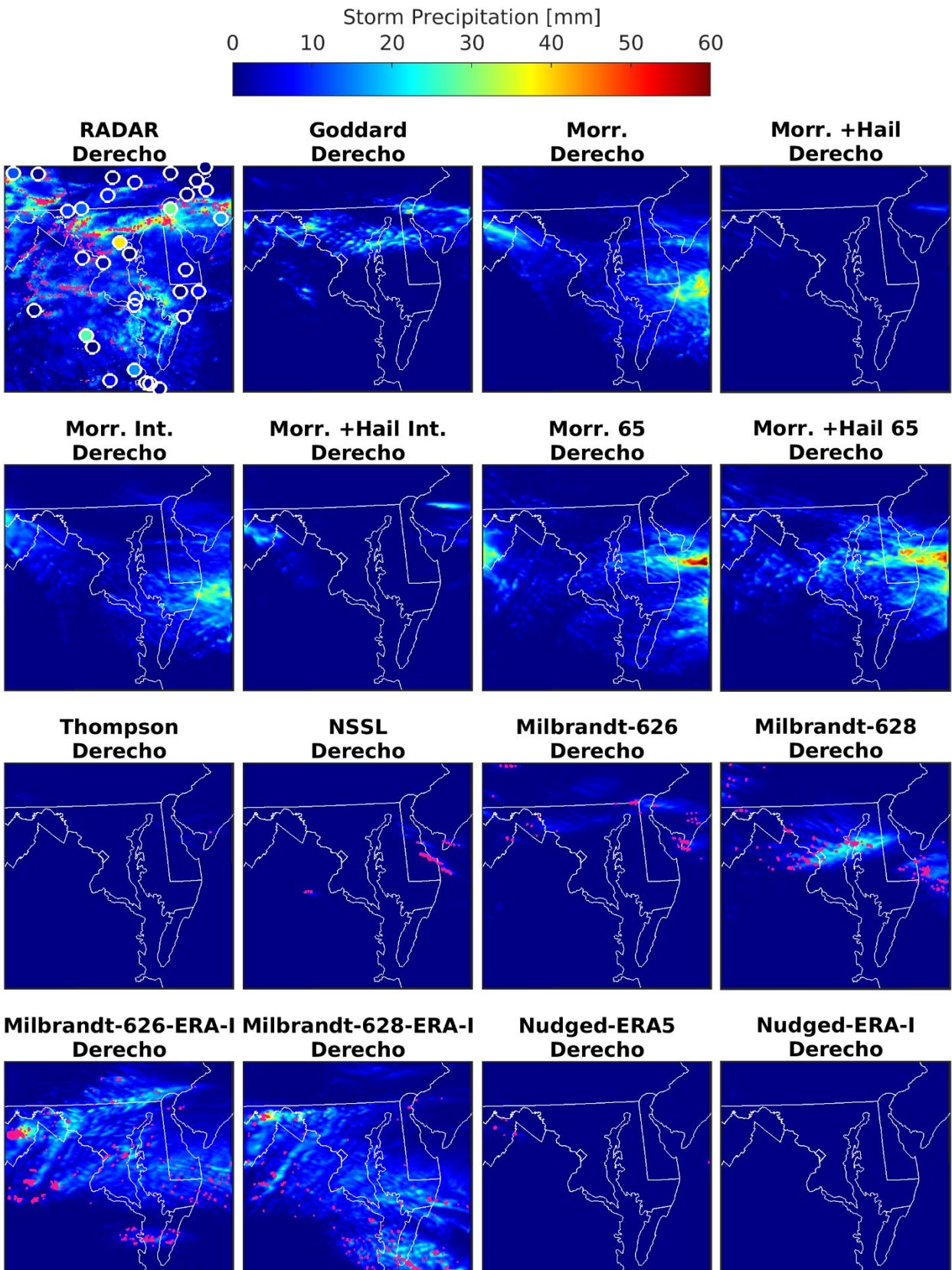

Figure 7: Total accumulated precipitation (mm) from RADAR observations and each WRF ensemble member during the Derecho period. Grid cells with MESH>25mm are marked in magenta. The RADAR indicates generally low precipitation, that is also reflected in the ASOS in situ observations (circles). Most WRF ensemble members exhibit a negative bias in terms of accumulated precipitation during the Derecho period.

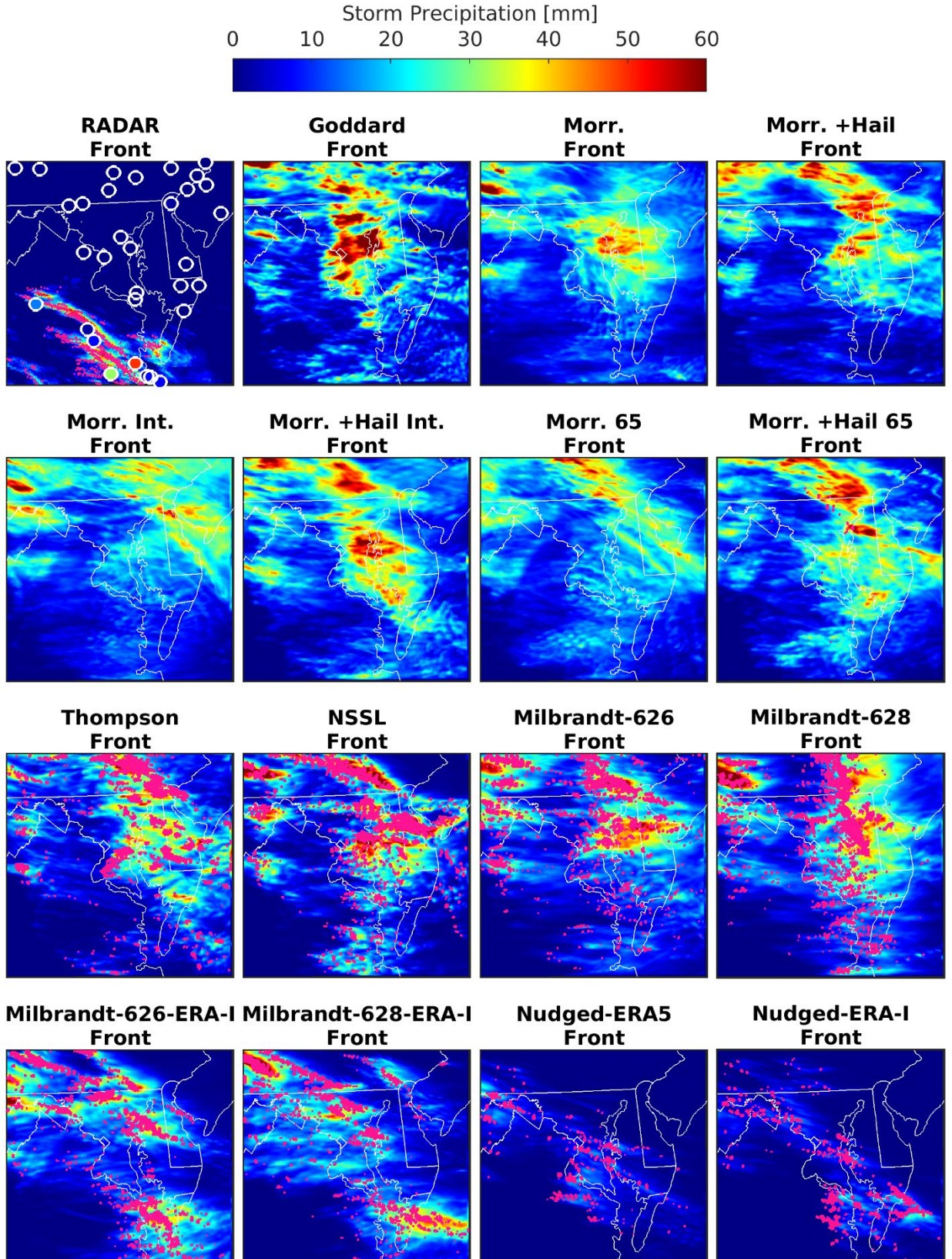

**Figure 8: Total accumulated precipitation (mm) from RADAR and each WRF ensemble member during the Front period. Grid cells with MESH>25mm are marked in magenta. In situ ASOS observations are also shown (circles). Areas with substantial precipitation accumulation are only evident from RADAR in bands in the south of the domain, in regions where hail is also indicated by the RADAR detection algorithm. Two-thirds of the domain shows little or no precipitation in either RADAR or ASOS data. All non-nudged WRF ensemble members indicate positive bias in domain-wide precipitation and over-predict the occurrence of hail.**

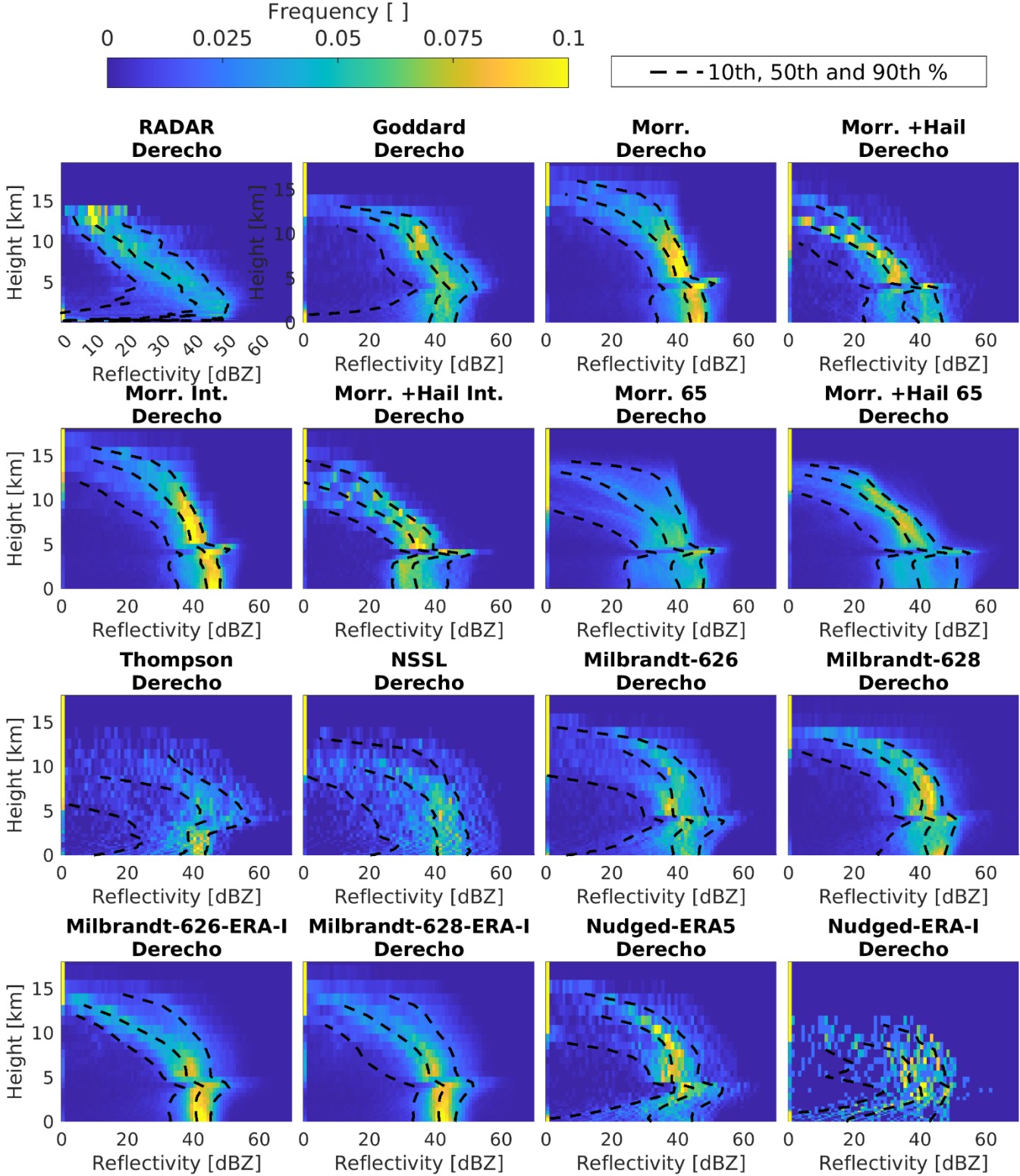


**Figure 9: Probability distributions of base reflectivity from RADAR and each WRF ensemble member at each model height at $t_p$ during the Derecho period. The plot shows the frequency with which a given reflectivity is observed at a given height in output for all domain d03 grid cells where cREF > 40 dBZ. Dotted lines show the 10th, 50th and 90th percentile reflectivity at each height.**

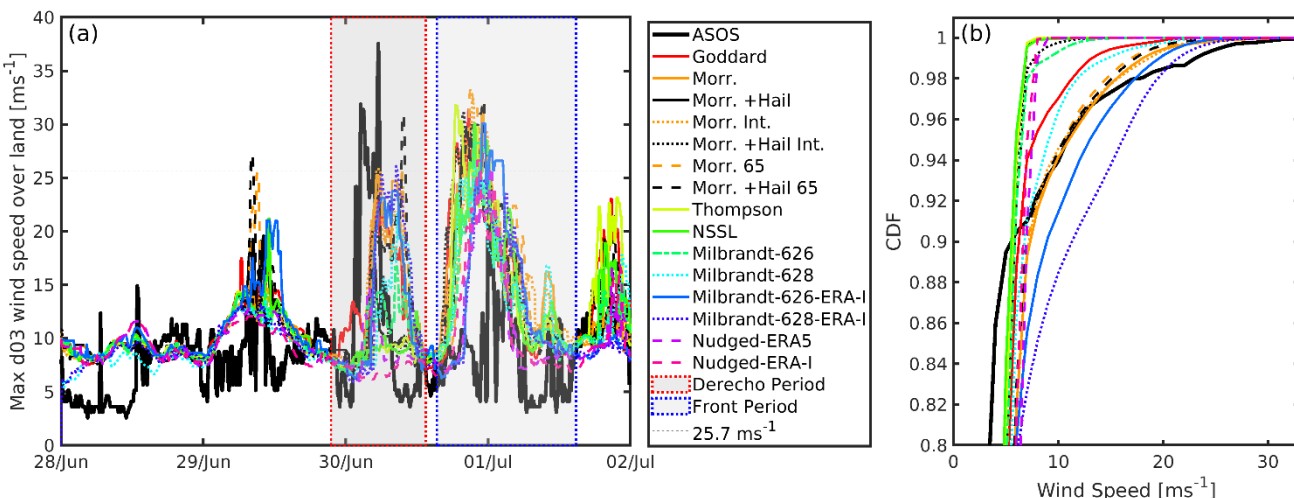

**Figure 10: (a) Time series of maximum wind gust in each 10-minute period at any ASOS station and any WRF land grid cell in domain d03, for each ensemble member (see maps in Figures 11 and 12). The timing of the (Derecho period: 29-Jun-2012 21:30:00 to 30-Jun-2012 13:30:00) and the frontal passage (Front period: 30-Jun-2012 15:20:00 to 01-Jul-2012 14:50:00) are denoted by the grey backgrounds. The horizontal grey line denotes a wind speed of 25.7 ms⁻¹ which is used by the NWS to define a damaging wind gust. (b) Spatiotemporal (every grid cell and all time steps) cumulative density functions (CDFs) of ASOS wind gusts and WRF wind speeds in**

**d03 during the Derecho period. To aid legibility only the upper 20% of values are shown. Goddard, Morrison, Milbrandt-626-ERA-I and Milbrandt-628-ERA-I are the ensemble members with highest maximum near-surface wind speeds. Only the Morrison, Milbrandt-626-ERA-I and Mibrandt-628-ERA-I, exhibit 98th percentile wind speeds (sampled at the model time step in both space and time over land grid cells) that lie within 50% of the ASOS observations of wind gusts (Figure 10b).**

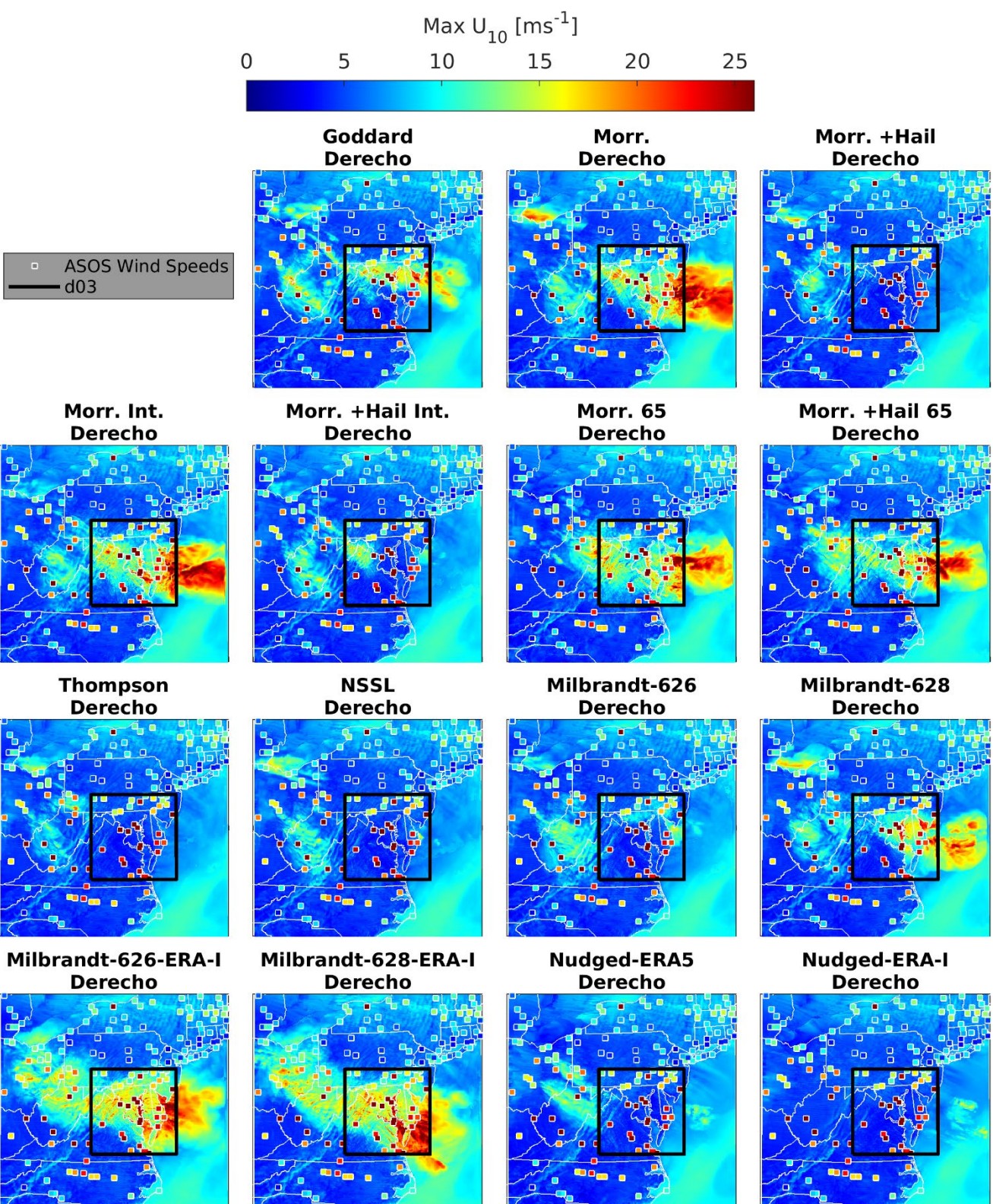


**Figure 11: Maximum wind speeds (ms$^{-1}$) at the model time step of 3.33 sec in d03 (outlined in black) and 10 seconds in d02 for each ensemble member during the Derecho period. Maximum 3-second wind gusts (ms$^{-1}$) at each ASOS station are shown by the square markers. The colorbar is truncated to aid legibility at the maximum value from any WRF ensemble member. Multiple ASOS stations reported wind gusts above 25.7 ms$^{-1}$.**


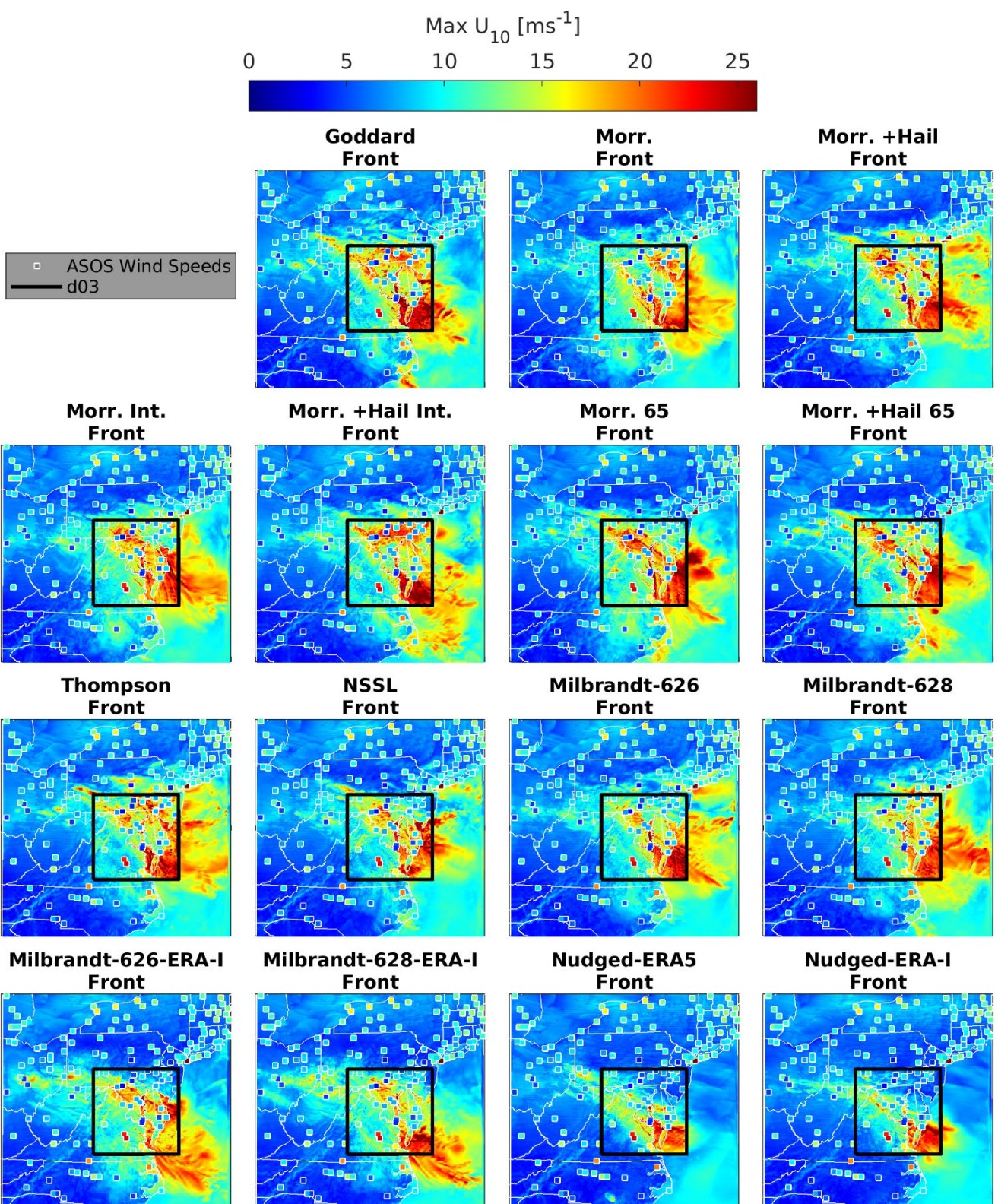

**Figure 12: Maximum wind speeds (ms⁻¹) at the model time step of 3.33 sec in d03 (outlined in black) and 10 seconds in d02 for each ensemble member during the Front period. Maximum 3-second wind gusts (ms⁻¹) at each ASOS station are shown by the square markers. The colorbar is as used in Figure 11 to aid comparisons.**


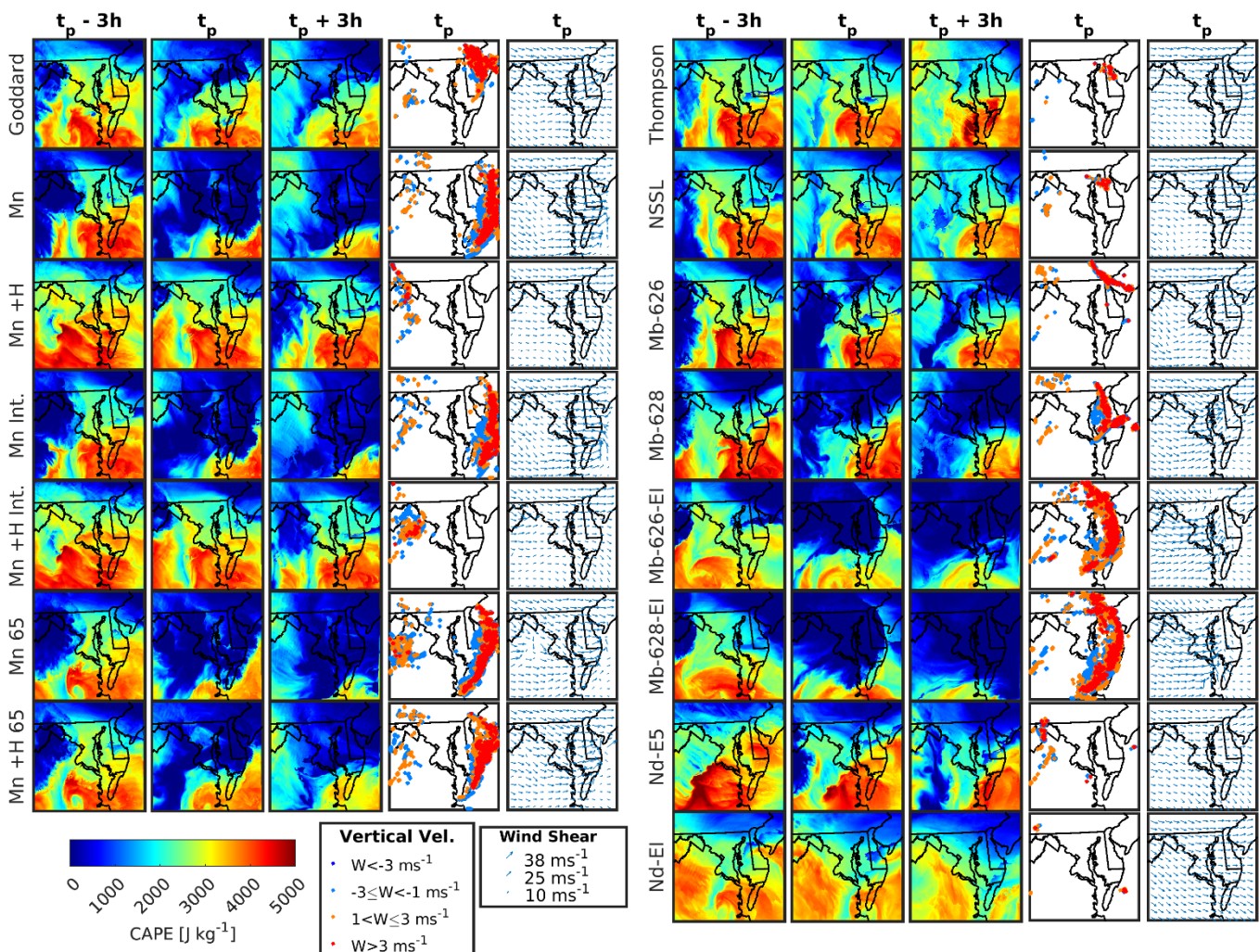

**Figure 13: MU-CAPE, vertical velocities, and wind shear within domain d03 for each WRF ensemble member. The left three columns show MU-CAPE for each member at $t_p$ -3 hours, $t_p$ (i.e. the time of peak spatial extent of cREF > 40 dBZ during the Derecho Period) and $t_p$ + 3hours. The fourth column shows the vertical wind speed (W) at 5000 m a.g.l at $t_p$. |W| within individual grid cells greatly exceeds 3 ms$^{-1}$ (values range from -10 to +10 ms$^{-1}$) these classes have been subjectively selected to capture the major regions of up and downdrafts. The right column shows the total wind shear between ground and 6000 m (S6, see definition in section 2.3). Larger versions of these maps have been provided in supplementary materials (Figures SM2-4, 5 and 7).**


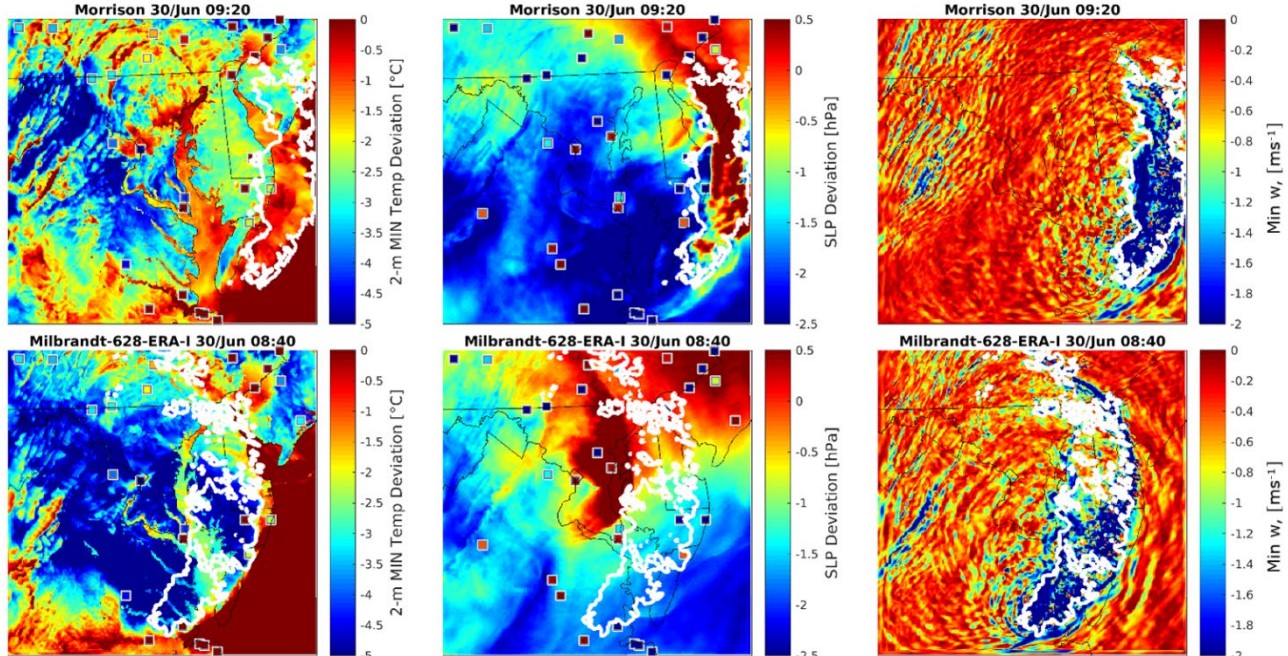

**Figure 14: WRF cold pool diagnostics at $t_p$ from two of the more skillful model ensemble members: Morrison (upper row) and Milbrandt-628-ERA-I (lower row). The metrics shown are 2 m temperature anomaly (left), sea level pressure anomaly (center) and largest negative velocity (right) for the WRF ensemble members. The title indicates the time step associated with $t_p$, i.e. when the maximum spatial coverage of cREF > 40 dBZ is simulated. Grid cells with cREF > 40 dB at $t_p$ is outlined by the white contour. Although these two ensemble members exhibit highest fidelity with respect to the areal coverage of cREF and exhibit relatively high skill in reproducing precipitation and wind gusts, as illustrated by Figure 14, these simulations generate different morphologies of the derecho.**