# Peer review of "How well are hazards associated with derechos reproduced in regional climate simulations?"

_Natural Hazards and Earth System Sciences, 2021_

## Author Comment (AC1)

Shepherd T.J., Letson F., Barthelmie R.J. and Pryor S.C. How well are hazards associated with derechos reproduced in regional climate simulations? *Natural Hazards and Earth System Sciences Discussions (*nhess-2021-373)

**Response to reviewer #1:**

We thank the reviewer for their thoughtful and thought-provoking comments. Below we provide a complete list of their comments (in black) and our responses (in green). We further note in brief how this material will be included in the manuscript once we have received all reviews.

Review of "How well are hazards associated with derechos reproduced in regional climate simulations?" by Shepherd et al.
This is a model evaluation study to evaluate WRF simulations downscaling to 1.3 km grid spacing with changes of cloud microphysics schemes, lateral boundary conditions (LBC), start date, and nudging. The focus is derecho induced from a mesoscale convective system (MCS). Since derechos cause significant infrastructure damages and economic loss, it is interesting to see if models can capture such extreme events and how the simulations are sensitive to different model setups and physical parameterizations. So, I advocate such studies. However, after looking at the results, I had to doubt whether the simulations were carried out correctly or the simulations were produced from a stable supercomputer/cluster. The model results swift from a convective system simulated with one microphysics scheme to the disappearance of the system with another microphysics scheme is something I never experienced as a senior cloud modeler. Particularly, switching graupel to hail in Morrison scheme also caused the disappearance of the MCS, which is not likely to occur since the change from graupel to hail renders minor changes relative to the entire scheme (mainly in fall speed and density). The hail option is recommended to use for continental deep convective cloud cases by the model developer but it does not simulate the MCS at all. I tested both options in several studies before and this never happened (the simulated convective cloud systems were generally very similar in morphology). In addition, there are so many literature studies with different microphysics schemes for a variety of cases that do not show such a result. As the authors stated this is expected result.
Also, none of the simulations can simulate both derecho and front stages of the observed system, then if the study have focused on why the simulations fail like this, it would still be useful. Furthermore, the sensitivity to two lateral boundary data (ERA5 and ERA-Interim) is also opposite compared with the previous studies (many literature studies showed ERA5 data is improved on ERA-Interim). With all these considered, it is very difficult for me to trust these model simulations thus I would recommend the rejection of the manuscript at this time.
Below I have some specific comments including the appropriate way to calculate the maximum hail size to compare with observations. Hope these comments will be useful for authors to improve the study.

Response: We concur that this is a very challenging event to simulate. We note it has subsequently been shown by Fierro (2014) that assimilation of lightning and RADAR data does improve the fidelity in very short term forecasts (< 6 hr lead time) as depicted visually in that work but without objective skill metrics applied. We further note this was a highly intense derecho. As Shourd and Kaplan (2021) report; "The 29–30 June 2012 "super" derecho was, up until the 10 August 2020 "Iowa Derecho", the most prolific derecho of modern times." Thus, it may present a particular challenge to models, and we believe that while there are parameter sensitivity studies in the literature, this event is worthy of special consideration. Our goal is to present an objective assessment of the inherent model skill as a function of configuration without nudging/data assimilation.

To illustrate the difficulty with simulation of this event we anticipate adding the following text at/close to line 78 in the original submission:

> A Service Assessment Team from the National Weather Service (NWS) evaluated performance during this event and found that "Unlike many major tornado outbreaks in the recent past, this

Shepherd T.J., Letson F., Barthelmie R.J. and Pryor S.C. How well are hazards associated with derechos reproduced in regional climate simulations? *Natural Hazards and Earth System Sciences Discussions* (nhess-2021-373)

**Response to reviewer #1:**

> event was not forecast well in advance." (NOAA, 2013). In part due to the multi-scale forcing of warm-season derechos, this, like other (weaker) derechos proved difficult to forecast > 12-24 hours ahead, and operational models including the North American Mesoscale (NAM) and Global Forecast System (GFS), provided "little assistance in forecasting this event more than 24 hours ahead of time". The day-3 and day-2 convective outlooks valid for 29 June showed only a 5% probability of severe thunderstorms anywhere over the eastern US, and even the Storm Prediction Center 1-day ahead convective outlook indicated only a 15% probability over most of the region that was impacted by the Derecho (NOAA, 2013). During the morning of June 29, some high-resolution, convection-permitting simulations with the High-Resolution Rapid Refresh model indicated the potential for development of intense thunderstorms and only in the afternoon of June 29 was the potential for tracking into the Mid-Atlantic coast identified (NOAA, 2013).

We appreciate the reviewer's concern about the simulations. However, we have reviewed our simulation settings and do not believe our simulations are in error. We have attached all namelist files to this response to review (and will include them all in the Supplementary Materials) and invite any comments and suggestions regarding the model configuration. With regard to the stability of the supercomputer/cluster, all simulations were performed on the Cornell University Center for Advanced Computing Aristotle Red Cloud system (https://federatedcloud.org/index.php). Each Red Cloud instance is a virtual machine in the cloud that consists of a user-defined number of cores. For all of our simulations we have used the same single instance with 28 identical cores (e.g. CPU type with identical clock speed etc.). Furthermore, we have used Docker to maintain version control. WRF was compiled inside Docker. As part of our due diligence procedure, we have undertaken rigorous testing and evaluation of possible machine sensitivity on Red Cloud. This testing included repeating simulations on the same instance, porting the Docker container WRF compile to a second instance and repeating the simulation, and repeating the simulation using different CPUs to test for machine sensitivity. In all cases the simulations we tested showed reproducibility. In the manuscript, we acknowledged the limitations of the study and provided a discussion based on the possibility of different convective forecast realizations. In response to the reviewer's comments and thus have additionally run new simulations with both Morrison settings on another compute platform (NERSC Cori). The additional Morrison simulations are simulated with an Intel compile of the same version of WRF. Below we show results from those simulations relative to original simulations on the Red Cloud system (with GNU compiler). As the reviewer will be aware, even in the absence of any 'system-errors', WRF exhibits a dependence on system architectures and the compiler (see; Hacker et al. 2017; Li et al. 2016; and Lighezzolo et al. 2018). Thus, we do not have an a priori expectation of exact bit-wise reproducibility. However, we do see a relatively high degree of agreement which leads us to have even higher confidence in our simulations. Further, critically for the reviewers point regarding the Morrison hail flag, the same sign of impact of turning on/off this flag is noted in both sets of paired simulations.

Specifically, in Figures 3-5 below, we show results from these simulations in a manner identical to Figures 3-5 in the original manuscript. As shown in Figure 3, 4, and 5 show how the Cori simulations compare to the original Morrison simulations run on the Aristotle platform with GNU. Considering differences in both the compiler and compute system, we show that the differences between the compile and compute platform are minimal. In addition, the behavior of the hail flag is consistent across the two simulations (i.e. Morrison+Hail vs. Morrison-intel+Hail). Fig 3, 4, and 5 will remain as is in the existing manuscript, but these updated Morrison comparisons will be added to the text with an additional figure showing these results.

Shepherd T.J., Letson F., Barthelmie R.J. and Pryor S.C. How well are hazards associated with derechos reproduced in regional climate simulations? *Natural Hazards and Earth System Sciences Discussions (*nhess-2021-373)

**Response to reviewer #1:**

[Figure]

Fig 3: (a) Time series of number of grid cells in domain d03 with composite reflectivity (cREF) > 40 dBZ from RADAR and the Morrison WRF ensemble members (original Morrison simulations vs. the NERSC Cori Intel compile Morrison simulations). The number of the 34 ASOS stations in domain d03 reporting thunderstorms is shown in grey (right axis). The timing of the (Derecho period: 29-Jun-2012 21:30:00 to 30-Jun-2012 13:30:00) and the frontal passage (Front period: 30-Jun-2012 15:20:00 to 01-Jul-2012 14:50:00) are denoted by the grey backgrounds. (b) The number of grid cells in domain d03 where output from each Morrison WRF ensemble member or the RADARs exceeded the specified threshold during the time step within the derecho period when the maximum number of grid cells exceeded the threshold. For example, in the RADAR observations there is a single 10-minute period during which approximately 5000 grid cells exhibit a value above 40 dBZ.

Shepherd T.J., Letson F., Barthelmie R.J. and Pryor S.C. How well are hazards associated with derechos reproduced in regional climate simulations? *Natural Hazards and Earth System Sciences Discussions (*nhess-2021-373)

**Response to reviewer #1:**

[Figure]

Fig 4: Composite reflectivity (cREF) in domain d03 at $t_p$ (the time when values from the maximum number of grid cells exceeded 40 dBZ) during the Derecho period from RADAR and each Morrison WRF ensemble member [original Morrison simulations vs. NERSC Cori Intel compile Morrison simulations] (times are noted in panel titles). The RADAR panel includes markers showing the presence (white) and absence (black) of thunderstorm reports from ASOS stations in domain d03 in the hour surrounding 03:30 UTC 30 June 2012.

Shepherd T.J., Letson F., Barthelmie R.J. and Pryor S.C. How well are hazards associated with derechos reproduced in regional climate simulations? *Natural Hazards and Earth System Sciences Discussions (*nhess-2021-373)

**Response to reviewer #1:**

[Figure]

Fig 5: Composite reflectivity (cREF) in domain d03 at $t_p$ (the time when values from the maximum number of grid cells exceeded 40 dBZ) during the Front period from RADAR and each Morrison WRF ensemble member [original Morrison simulations vs. NERSC Cori Intel compile Morrison simulations] (times are noted in panel titles). The RADAR panel includes markers showing the presence (white) and absence (black) of thunderstorm reports from ASOS stations in domain d03 in the hour surrounding 05:20 UTC 1 July 2012.

We agree with the reviewer that the results in this paper when using the Morrison scheme with/without the hail switch is indeed interesting. We believe this makes the results even more relevant for the modeling community. We had earlier contacted Hugh Morrison to discuss our findings. His personal communication reads:

*"I'm not surprised there are large differences setting the hail flag to "on". Interestingly it looks like results with this are closer overall to Thompson and NSSL. In general setting the flag to hail leads to a*

Shepherd T.J., Letson F., Barthelmie R.J. and Pryor S.C. How well are hazards associated with derechos reproduced in regional climate simulations? *Natural Hazards and Earth System Sciences Discussions (*nhess-2021-373)

**Response to reviewer #1:**

*reduced area of reflectivity but sharper precip rates in simulated squall lines (e.g. Morrison et al. 2015, JAS). …. But in "real case" simulations with realistic lateral boundary conditions, initial conditions and a longer forecast time the simulations can diverge much more. We see this every year, for example, at NSSL's spring harzardous weather testbed where, among other things, they analyze WRF ensembles with different microphysics schemes. Anecdotally, we often see large changes in convective structure, timing, placement, and mode using different schemes, especially after ~18-24 hours forecast time. To put sensitivity to microphysical changes into context for such real case runs, we've used ensembles forced with different sets of initial/lateral boundary conditions (say, from GEFS) and ensembles with small perturbations to the initial potential temperature field (Stanford et al. 2019, JAMES)."*

We envisage including the precis of the new analyses and simulations at/close to line 419 along with the following text:

> The relatively poor simulation performance for each of the ensemble members is consistent with the aforementioned literature regarding the specific challenge that this event presented. However, it also raised concerns regarding a possible issue with the stability of the computational platform. Thus, simulations of two of the ensemble members were repeated on a separate computational platform (the U.S. Department of Energy NERSC Cori Cray XC40) and with a different compiler (INTEL). Bit-wise reproducibility is not expected due to previously documented system architecture and compiler dependence of WRF simulations (Hacker et al. 2017; Li et al. 2016). Thus, these simulations are designed to evaluate whether use of a different system yields marked improvements in terms of the fidelity with which the Derecho is simulated and to evaluate if the response to turning on the hail flag in the Morrison scheme is consistent. The results of these additional simulations are summarized in Figure 11 in terms of the time series of the number of grid cells with high cREF and in Figure 12 in terms of the cREF spatial patterns at $t_p$. These and other diagnostics (not shown) indicate a high degree of similarity between the output of these simulations and the original ensemble members. Our inference is that the original ensemble members are reliable.

With respect to the importance of the LBC – while ERA5 is generally thought to be superior (and indeed this was our expectation) it is not uniformly the case. Indeed, the question of LBC remains an active area of research (e.g. Ahrens and Leps, 2021). In this specific case the improved performance in simulations with ERA-Interim may be linked to the specific multi-scale dynamics associated with this derecho (see details from Shourd and Kaplan, 2021) that were better depicted in the Era-Interim reanalysis. We have undertaken additional analyses of these IC from ERA5 and ERA-Interim as described in detail below.

The fact that our simulation results are not entirely congruent with the reviewer's (or our) expectations is entirely why we performed these analyses and submitted this manuscript.

Abstract,
"We also examine the degree to which each ensemble member differs with respect to key mesoscale drivers of convective systems (e.g. convective available potential energy and vertical wind shear) and critical manifestations of deep convection; e.g. vertical velocities, cold pool generation, and how those properties relate to correct characterization of the associated atmospheric hazards (wind gusts and hail)."
-The sentence is near the end of the abstract about it is still about the scientific approach. Suggest changing to phrasing it from the angle of describing your key findings, which is more appropriate for a scientific paper.

Shepherd T.J., Letson F., Barthelmie R.J. and Pryor S.C. How well are hazards associated with derechos reproduced in regional climate simulations? *Natural Hazards and Earth System Sciences Discussions* (nhess-2021-373)

**Response to reviewer #1:**

We regret the reviewer did not find our abstract sufficiently detailed. We note the instructions to authors with respect to the abstract read; '**Abstract**: the abstract should be intelligible to the general reader without reference to the text. After a brief introduction of the topic, the summary recapitulates the key points of the article and mentions possible directions for prospective research. Reference citations should not be included in this section, unless urgently required, and abbreviations should not be included without explanations. An abstract should be short, clear, concise, and written in English with correct spelling and good sentence structure.' We felt we were generally compliant with that instruction but based on the reviewers comment propose to modify the abstract to read:

An 11-member ensemble of convection-permitting regional simulations of the fast-moving and destructive derecho of June 29 – 30, 2012 that impacted the northeastern urban corridor of the US is presented. This event generated 1100 reports of damaging winds, significant wind gusts over an extensive area of up to 500,000 $km^2$, caused several fatalities and resulted in widespread loss of electrical power. Extreme events such as this are increasingly being used within pseudo-global warming experiments that seek to examine the sensitivity of historical, societally-important events to global climate non-stationarity and how they may evolve as a result of changing thermodynamic and dynamic context. As such it is important to examine the fidelity with which such events are described in hindcast experiments. The regional simulations presented herein are performed using the Weather Research and Forecasting (WRF) model. The resulting ensemble is used to explore simulation fidelity relative to observations for wind gust magnitudes, spatial scales of convection (as manifest in high composite reflectivity, cREF), and both rainfall and hail production as a function of model configuration (microphysics parameterization, lateral boundary conditions (LBC), start date, and use of nudging). We also examine the degree to which each ensemble member differs with respect to key mesoscale drivers of convective systems (e.g. convective available potential energy and vertical wind shear) and critical manifestations of deep convection; e.g. vertical velocities, cold pool generation, and how those properties relate to correct characterization of the associated atmospheric hazards (wind gusts and hail). Use of a double-moment, 7-class scheme with number concentrations for all species (including hail and graupel) results in the greatest fidelity of model simulated wind gusts and convective structure against the observations of this event. However, all ensemble members fail to capture the intensity of the event in terms of the spatial extent of convection and the production of high near-surface wind gusts. We further show very high sensitivity to the LBC employed and specifically that simulation fidelity is higher for simulations nested within ERA-Interim than ERA5. Excess CAPE availability in all ensemble members after the Derecho passage leads to excess production of convective cells, wind gusts, cREF > 40dBZ and precipitation during a frontal passage on the subsequent day. This event proved very challenging to forecast in real-time and to reproduce in the 11-member hindcast simulation ensemble presented here. Future work could examine if simulations with other initial and lateral boundary conditions can achieve greater fidelity.

Introduction,

"deep convection disproportionally contributes….", disproportionally does not deliver a good meaning here. Suggest rewording.
"…and three events caused more than 60% of a utilities' customers power outage; a derecho, an ice storm and a hurricane (Shield, 2021)", not three events, should be three types of events.

Response: We regret this typographic error that led to omission of the word "types".

Shepherd T.J., Letson F., Barthelmie R.J. and Pryor S.C. How well are hazards associated with derechos reproduced in regional climate simulations? *Natural Hazards and Earth System Sciences Discussions (*nhess-2021-373)

**Response to reviewer #1:**

Section 1.2: this section is for case description. There is too much text in describing the societal and economic impacts but lacking a description of the large-scale and mesoscale metrological environments in which the event developed, which is the key information for the study. Also, there should be observational analysis of this event on storm and wind properties which should be discussed to provide a better background of the events.

Response: The journal description reads as follows: "Natural Hazards and Earth System Sciences (NHESS) is a not-for-profit interdisciplinary and international journal dedicated to the public discussion and open-access publication of high-quality studies and original research on natural hazards and their consequences. Embracing a holistic Earth system science approach, NHESS serves a wide and diverse community of research scientists, practitioners, and decision makers concerned with detection of natural hazards, monitoring and modelling, vulnerability and risk assessment, and the design and implementation of mitigation and adaptation strategies, including economical, societal, and educational aspects." Thus, we felt it was appropriate to include information regarding the economical and societal aspect of the hazard. But we are happy to add further information about the meteorological environment. We thus propose to include the following text at/close to line 76:

> Prior research has suggested that Derecho events in the eastern USA are often preceded by large scale troughing over western North America (Cordeira et al. 2017). This was also evident in the June 2012 event, where associated ridging over the eastern US caused extreme near-surface air temperatures and humidity leading to issuance of heatwave advisories (Cattiaux and Yiou, 2013). Rossby wave breaking lead to development of an intense elevated mixed layer (EML, 700-500 hPa) over the central US that subsequently propagated eastwards (Shourd & Kaplan, 2021). The upper-level flow early on June 29 was dominated by ridging over the southeastern US (Figure 11 on the initial conditions) and a near-zonal Jetstream extending from the middle of Wisconsin across the Great Lakes and into New York state, with an embedded jet streak over the northern Great Lakes (Shourd & Kaplan, 2021). Near-surface conditions were dominated by a complex frontal boundary extending approximately west-east across Iowa into Pennsylvania, with very high humidity and high near-surface temperatures just to the south (Figure 11). It is noteworthy that the 12-hour forecast from the NAM model (grid-spacing of 12 km) valid at 8pm (local time) on 29 June 2012 indicated an extensive area of surface-based CAPE in excess of 4000 Jkg$^{-1}$ over the Appalachian Mountains (covering almost all of the state of west Virginia) associated with the eastward propagation of the EML but projected very little precipitation, which contributed to uncertainty in forecasting the location and intensity of the derecho (NOAA, 2013).

Section 2
Section 2.1: (a) the description of model simulations needs some clarification. I am confused by the descriptions at line 160-165. First you described 3 nested domains were used but then said "A single domain configuration and inner nest grid spacing is used in all members of the ensemble…". Are you using two types of domain settings (3 nest domains and single 1.3 km domain)? If so, please clearly describe which domain setup is used for each simulation listed in Table 2? (b) Nudging simulations are not even mentioned here but they are discussed a lot in the Result section.

Response: The text in the revised manuscript will read "The same domain configuration is used in all members of the ensemble."

Shepherd T.J., Letson F., Barthelmie R.J. and Pryor S.C. How well are hazards associated with derechos reproduced in regional climate simulations? *Natural Hazards and Earth System Sciences Discussions (*nhess-2021-373)

**Response to reviewer #1:**

Section 2.2. There are better precipitation data than retrieved precipitation rate using Z-R relationship, which in general has a large uncertainty, such as rain gauge data and Stage IV data from NOAA which combines radar and rain gauge measurements.

Response: We thank the reviewer for this suggestion. We note in situ data regarding precipitation from tipping bucket rain gauges operated as part of the NWS ASOS network are included in the original manuscript in Figures 6 and 7. Nevertheless at the reviewers request we have added Stage IV gridded data (that is a combined product of RADAR derived rainfall rates and in situ measurements) into the analysis. E.g. We have updated the fidelity metrics in Table 5 to include a precipitation comparison between WRF and Stage IV. NCEP/EMC 4KM Gridded Data (GRIB) Stage IV Data (Du, 2011). These data were downloaded from https://data.eol.ucar.edu/dataset/21.093 in grib format and converted for processing to netcdf using the NCL command ncl_convert2nc. Hourly precipitation amounts were summed for the entire duration of the Derecho period.

The revised manuscript will thus include; A modified table 5 (see below) because the following text:
- At/near lines 232: The NCEP/EMC 4KM Gridded Data Stage IV precipitation product (Du, 2011) which is a blend of RADAR-derived precipitation with in situ measurements is also used in the model fidelity assessment. The spatial fields of accumulated precipitation from the RADAR and the Stage IV product are very similar but the total domain-wide amounts during the Derecho and Frontal periods differ.
- At/near line 258: Each ensemble member exhibits slightly higher agreement with the Stage IV precipitation product than with RADAR-only total accumulated precipitation during the Derecho period (Table 5).

Shepherd T.J., Letson F., Barthelmie R.J. and Pryor S.C. How well are hazards associated with derechos reproduced in regional climate simulations? *Natural Hazards and Earth System Sciences Discussions (*nhess-2021-373)

**Response to reviewer #1:**

Table 5: Metrics of simulation fidelity relative to observations, and convection metrics derived from output from each WRF member during the period of the derecho passage (Derecho period: 29-Jun-2012 21:30:00 to 30-Jun-2012 13:30:00). The metrics of simulation fidelity are described in section 2.2 and are as follows: The Max Gust Ratio: the ratio of the maximum wind gust in any land grid cell from WRF output and observations at the ASOS stations. Total Precip. Ratio: the ratio of the spatial mean total accumulated precipitation from WRF to RADAR and the Stage IV product, respectively, for any grid cell with common coverage. cREF>40 dBZ: the ratio of the spatial extent of grid cells with cREF above 40 dBZ at the peak coverage in WRF and RADAR. The lower portion of the table shows the Spearman rank correlation for the 11 values of each metric (one for each ensemble member). This analysis thus shows the degree to which an ensemble member that exhibit high values of a given metric also generates high values of a second metric. The color-coding used in this table is as follows; for the measures of simulation fidelity table cells colored red have low fidelity, and those indicated by cyan exhibit relatively high fidelity. For all other cells in the table, a background of orange indicates low values, while blue indicates comparatively high values. The saturation of the color indicates relative ordering of the values. The definitions of each convection metric are given in section 2.3.

| | Simulation Fidelity | | | | Convection Metric | | | | | |
|---|---|---|---|---|---|---|---|---|---|---|
| | Max Gust Ratio | Total Precip. Ratio (RADAR) | Total Precip. Ratio (Stage IV) | cREF>40 dBZ Ratio | 95% Temperature deviation [-K] | 95% SLP deviation [hPa] | Median CAPE loss [J kg$^{-1}$] | 95%-W [ms$^{-1}$] | Max std(w) height [km] | $Z_{R20}$ [km] |
| Goddard | 0.61 | 0.206 | 0.218 | 0.346 | 4.23 | 2.1 | 876 | 0.15 | 8 | 12.9 |
| Morrison | 0.67 | 0.413 | 0.435 | 0.788 | 3.29 | 1.85 | 1532 | 0.15 | 8 | 15.3 |
| Morrison +Hail | 0.46 | 0.016 | 0.017 | 0.102 | 2.12 | -0.756 | 175 | 0.11 | 8 | 9 |
| Thompson | 0.26 | 0.006 | 0.006 | 0.044 | 1.97 | 0.478 | 61 | 0.05 | 5.6 | 12.7 |
| NSSL | 0.33 | 0.015 | 0.015 | 0.043 | 3.27 | 0.238 | -42 | 0.06 | 6.4 | 12.8 |
| Milbrandt-626 | 0.44 | 0.061 | 0.064 | 0.269 | 3.3 | 0.596 | 963 | 0.09 | 7.1 | 13.8 |
| Milbrandt-628 | 0.57 | 0.185 | 0.195 | 0.391 | 4.64 | 3.38 | 1428 | 0.09 | 8.9 | 13.7 |
| Milbrandt-626-ERA-I | 0.63 | 0.566 | 0.597 | 0.844 | 5.44 | 2.36 | 1960 | 0.15 | 8.9 | 14.4 |
| Milbrandt-628-ERA-I | 0.69 | 0.636 | 0.671 | 0.945 | 5.58 | 2.79 | 2030 | 0.14 | 7.1 | 14.9 |
| Nudged-ERA5 | 0.39 | 0.004 | 0.004 | 0.037 | 1.38 | -1.75 | 575 | 0.03 | 7.1 | 14 |
| Nudged-ERA-I | 0.22 | 0 | 0 | 0.004 | 4.05 | -1.22 | 182 | 0.01 | 5.6 | 11.6 |
| Spearman Rank Correlations | Max Gust Ratio | Total Precip. Ratio (RADAR) | Total Precip. Ratio (Stage IV) | cREF>40 dBZ Ratio | 95% Temperature deviation | 95% SLP deviation | Median CAPE Loss | 95%-W | Max std(w) height | $Z_{R20}$ |
| Max Gust Ratio | 1 | | | | | | | | | |
| Total Precip Ratio | 0.95 | 1 | | | | | | | | |
| Total Precip Ratio | 0.95 | 1 | 1 | | | | | | | |
| cREF>40 dBZ Ratio | 0.93 | 0.98 | 0.98 | 1 | | | | | | |
| 95% Temperature | 0.72 | 0.79 | 0.79 | 0.78 | 1 | | | | | |

Shepherd T.J., Letson F., Barthelmie R.J. and Pryor S.C. How well are hazards associated with derechos reproduced in regional climate simulations? *Natural Hazards and Earth System Sciences Discussions (*nhess-2021-373)

**Response to reviewer #1:**

| Spearman Rank Correlations | Max Gust Ratio | Total Precip. Ratio (RADAR) | Total Precip. Ratio (Stage IV) | cREF>40 dBZ Ratio | 95% Temperature deviation | 95% SLP deviation | Median CAPE Loss | 95% -W | Max std(w) height | $Z_{R20}$ |
|---|---|---|---|---|---|---|---|---|---|---|
| 95% SLP deviation | 0.75 | 0.87 | 0.87 | 0.92 | 0.81 | 1 | | | | |
| Median CAPE Loss | 0.81 | 0.77 | 0.77 | 0.81 | 0.79 | 0.72 | 1 | | | |
| 95% -W | 0.9 | 0.92 | 0.92 | 0.85 | 0.64 | 0.7 | 0.56 | 1 | | |
| Max std(w) height | 0.78 | 0.68 | 0.68 | 0.69 | 0.47 | 0.57 | 0.6 | 0.75 | 1 | |
| ZR20 | 0.65 | 0.61 | 0.61 | 0.61 | 0.5 | 0.52 | 0.81 | 0.36 | 0.37 | 1 |

Section 2.3, "Grid cells in d03 are classified as containing 'significant hail' in the WRF simulations if there is > 1 mm of hail and/or graupel accumulation, and in RADAR observations for MESH > 5mm. First, how the simulation and observations can be compared since different calculations of significant severe hail (SSH) are used? Second, What are you based on to define SSH with " > 1 mm of hail and/or graupel accumulation" in the simulations? Accumulation over what time period? In literature there are a few methods to calculate the maximum hail size based on model predicted hail/graupel size distribution such as Thompson (J. A. Milbrandt, M. K. Yau, A Multimoment Bulk Microphysics Parameterization. Part III: Control Simulation of a Hailstorm. J. Atmos. Sci. 63, 3114-3136, 2006) and Snook (N. Snook, et al. Prediction and Ensemble Forecast Verification of Hail in the Supercell Storms of 20 May 2013. Weather Forecasting 31, 811-825,2016) methods. These methods make the model-observation comparison more physically consistent. BTW, "significant hail" should be "significant severe hail" based on the conventional terminology from literature. Another comment on this section is that the metrics description for evaluating models takes too much text and can be tightened up.

Response: Thank you for your suggestion. We have reanalysed our output to derive and estimate of the maximum expected size of hail (MESH) in a manner equivalent to that used in the RADAR processing (per Snook (2013)). We have updated our hail threshold to MESH > 25 mm for both WRF and RADAR, corresponding to 'severe hail' in previous work (e.g. Labriola et al., 2019). Figures 6 and 7 and Table 4 have been updated accordingly (see below). The revised manuscript will have these revised figures and table.

**Replacement text for lines 224 to 226:** In the current work, a distinction is drawn between hail reports with MESH > 25 mm and those without. This is a diameter threshold has been previously used for identifying 'severe hail' (Labriola et al. 2019).

**Replacement text for lines 260 to 263:** Hail occurrence from the WRF ensemble members is also evaluated against RADAR and ASOS observations along with the presence of 'severe hail'. Grid cells in d03 are classified as containing 'severe hail' in the WRF simulations and RADAR observations when MESH > 25mm. MESH for the WRF simulations is estimated using a weighted summation of hail kinetic energy flux for elevations above the melting layer. Hail kinetic energy fluxes are inferred as a function of reflectivity. This method was developed for use with RADAR data (Witt et al., 1998).

Shepherd T.J., Letson F., Barthelmie R.J. and Pryor S.C. How well are hazards associated with derechos reproduced in regional climate simulations? *Natural Hazards and Earth System Sciences Discussions (*nhess-2021-373)

**Response to reviewer #1:**

**Replacement text for Lines 357 to 371:** When remapped to the WRF grid, the RADAR data indicate 824 of the almost 90,000 grid cells experienced severe hail during the Derecho period (Table 4). These locations identified by the RADAR detection algorithm as exhibiting hail and MESH > 25 mm are distributed throughout domain d03 (Figure 6). The WRF ensemble members – particularly those that employ the Milbrandt microphysics scheme indicate much greater spatial coverage of hail (Table 4). When the threshold of MESH > 25 mm is applied to the WRF output the occurrence of hail greatly decreases rather few grid cells show hail above this threshold (Table 4). During the Front period the situation is reversed. RADAR observations show limited areas with accumulated precipitation > 40 mm located in bands in the south of the domain, in regions where hail is also indicated by the RADAR detection algorithm (Figure 7). Two-thirds of the domain shows little or no precipitation in either RADAR or ASOS data. All non-nudged WRF ensemble members indicate positive bias in domain-wide precipitation and over-predict the occurrence of hail (Table 4). All four non-nudged ensemble members with the Milbrandt microphysics scheme also indicate multiple locations with MESH > 25 mm. The number of grid cells with RADAR detection of hail shows closest agreement with the Morrison+Hail simulation (Table 4). Using the MESH > 25 mm threshold as indicative of severe hail, the closest accord for the Front period is found for the Nudged-ERA5 ensemble member (Table 4).

Shepherd T.J., Letson F., Barthelmie R.J. and Pryor S.C. How well are hazards associated with derechos reproduced in regional climate simulations? *Natural Hazards and Earth System Sciences Discussions (*nhess-2021-373)

**Response to reviewer #1:**

[Figure]

Figure 6: Total accumulated precipitation (mm) from RADAR observations and each WRF ensemble member during the Derecho period. Grid cells with MESH>25mm are marked in magenta.

Shepherd T.J., Letson F., Barthelmie R.J. and Pryor S.C. How well are hazards associated with derechos reproduced in regional climate simulations? *Natural Hazards and Earth System Sciences Discussions (*nhess-2021-373)

**Response to reviewer #1:**

[Figure]

Figure 7: Total accumulated precipitation (mm) from RADAR observations and each WRF ensemble member during the Front period. Grid cells with MESH>25mm are marked in magenta.

Shepherd T.J., Letson F., Barthelmie R.J. and Pryor S.C. How well are hazards associated with derechos reproduced in regional climate simulations? *Natural Hazards and Earth System Sciences Discussions* *(*nhess-2021-373)

**Response to reviewer #1:**

Table 4: Number of grid cells in domain d03 where hail is indicated by the RADARs or present in the WRF simulations during the derecho (Derecho period: 29-Jun-2012 21:30:00 to 30-Jun-2012 13:30:00) and the frontal passage (Front period: 30-Jun-2012 15:20:00 to 01-Jul-2012 14:50:00). Also shown is the number of grid cells with Maximum Estimated Size of Hail (MESH) above 25 mm from the RADAR or WRF. Recall: RADAR detection of hail is re-gridded onto the WRF grid used for domain d03 prior to use in the model evaluation.

| | # Grid cells with hail | | # Grid cells with MESH > 25 mm | |
|---|---|---|---|---|
| | Derecho | Front | Derecho | Front |
| RADAR | 3078 | 2152 | 824 | 813 |
| Ensemble member | | | | |
| Goddard | 0 | 10 | 0 | 6 |
| Morrison | 0 | 24 | 0 | 0 |
| Morrison +Hail | 3000 | 74398 | 0 | 0 |
| Thompson | 10 | 8996 | 2 | 4909 |
| NSSL | 7446 | 79890 | 135 | 5907 |
| Milbrandt-626 | 16368 | 78276 | 167 | 5687 |
| Milbrandt-628 | 26183 | 77415 | 436 | 6461 |
| Milbrandt-626-ERA-I | 54406 | 68899 | 782 | 4928 |
| Milbrandt-628-ERA-I | 63695 | 67671 | 568 | 4028 |
| Nudged-ERA5 | 2428 | 37913 | 21 | 1226 |
| Nudged-ERA-I | 195 | 37692 | 0 | 2071 |

Section 3,
Figure 4, the simulations with changes of different microphysics only (Goddard, Morrison, Morrison+hail, Thompson, NSSL, Milbrandt-626) totally fails to simulate the convective systems except Morrison captured the linear system. Switching to another microphysics scheme in WRF usually does not make such a large storm system totally disappear (never experienced or saw this in literature). The coupling of those microphysics scheme with WRF should have no problem so it is strange this happened. What is especially suspicious is when switching the graupel option to hail in Morrison scheme, the large linear MCS system disappeared. This would not be possible in a few days of forecasting simulations since slight differences in microphysics should not have such a huge upscale effect on mesoscale systems in a few days.

Response: As noted above, one of the limitations with this (and most) modeling studies is the introduction of small-scale perturbations at t=0 that contribute to upscale growth and drive different convective forecast realizations. It is not unprecedented or unreasonable to expect sensitivity between physics schemes in WRF, and there are numerous studies in the literature that have explored this. Indeed, a study of the differences in the mass and number concentrations (where present) in each of the schemes illustrates the likelihood in the varying degree of skill that each scheme can predict the sub-grid scale processes. Again, as noted above, the use of the hail switch leads to a reduced area of reflectivity (e.g. see Morrison et al. 2015 JAS) and can indeed have large enough upscale growth from t=0 to affect system development and propagation, which is evident in the analysis. Furthermore, the noted improvement in the 28 June 00Z initialized cases is further evidence to suggest the model predictability of this event is improved the closer the model start date is to the derecho genesis on 29 June. This makes sense when appreciating the role of convective error growth over time, thus the cases initialized on 26 June 00Z would therefore be more prone to some amount of model error.

Shepherd T.J., Letson F., Barthelmie R.J. and Pryor S.C. How well are hazards associated with derechos reproduced in regional climate simulations? *Natural Hazards and Earth System Sciences Discussions* (nhess-2021-373)

**Response to reviewer #1:**

Furthermore, based on Figures 4-7, none of the simulations can simulate both derecho and front stages of the observed system, then why the simulations fail like this should be the first priority of the study.

This is indeed a key motivation for this manuscript and for section 3.2 where we seek to advance understanding of the key precursors (e.g. CAPE and vertical shear) and outcomes of the Derecho and our enhanced discussion of the initial conditions.

Figure 8, why not plot the radar measurements?

Response: We thank the reviewer for this suggestion and have included the RADAR measurements. As the reviewer will be aware the NWS RADAR scan at (typically) six elevation angles and thus the resulting data are not on the same 3D grid as WRF. Thus, we have taken the base reflectivity data from each 360º arc scan at each elevation angle from each RADAR at $t_p$ and used them to derive vertical profiles of reflectivity. We note that the RADAR observations do not provide coverage for the entire vertical structure, but we have included the measurements to the available level (these are now included on Figure 8, see below).
We have also updated Figure S1 to show RADAR observations for the Front period (see Figure S1 below).

We envisage replacing both Figure 8 and Figure S1 and including the following text at/near line 374:
> Vertical profiles of base reflectivity data from each 360º arc scan at each elevation angle from each RADAR at $t_p$ are also shown in Figure 8. Though this observationally constrained vertical profile is based on considerably lower data volumes than in the WRF output, it is worthy of note that the peak in reflectivity in the RADAR is located lower in the atmosphere than in most of the WRF ensemble members. Further, a greater fraction of the reflectivity values at 12 km (the highest height from which any RADAR data are available) from the RADAR observations are > 20 dBZ than in many, but not all, of the ensemble members.

Shepherd T.J., Letson F., Barthelmie R.J. and Pryor S.C. How well are hazards associated with derechos reproduced in regional climate simulations? *Natural Hazards and Earth System Sciences Discussions (*nhess-2021-373)

**Response to reviewer #1:**

[Figure]

Figure 8: Probability distributions of base reflectivity from RADAR and derived RADAR reflectivity from each WRF ensemble member at each model height at $t_p$ during the Derecho period. The plot shows the frequency with which a given reflectivity is observed at a given height in output for all domain d03 grid cells where cREF > 40 dBZ. Dotted lines show the $10^{th}$, $50^{th}$ and $90^{th}$ percentile reflectivity at each height.

Shepherd T.J., Letson F., Barthelmie R.J. and Pryor S.C. How well are hazards associated with derechos reproduced in regional climate simulations? *Natural Hazards and Earth System Sciences Discussions (*nhess-2021-373)

**Response to reviewer #1:**

[Figure]

Figure S1: Probability distributions of base reflectivity from RADAR and derived RADAR reflectivity from each WRF ensemble member at each model height at $t_p$ during the Front period. The plot shows the frequency with which a given reflectivity is observed at a given height in output for all domain d03 grid cells where cREF > 40 dBZ. Dotted lines show the 10[th], 50[th] and 90[th] percentile reflectivity at each height.

Shepherd T.J., Letson F., Barthelmie R.J. and Pryor S.C. How well are hazards associated with derechos reproduced in regional climate simulations? *Natural Hazards and Earth System Sciences Discussions* (nhess-2021-373)

**Response to reviewer #1:**

Figures 9 -11, because the observed MCS was not captured in most of the simulations, one can see the simulated wind speeds are also way off. I do not see a point to intercompare convective winds, cold pools, or other storm related properties since the mesoscale system does not even exist in the model simulations or the basic mesoscale storm is totally wrong as shown in Figures 4-7. Therefore, I did not review Section 3.2 "Linking fidelity to metrics of CAPE, downbursts, and cold pool generation".

Response: Now that we have established the robustness of the simulations (in terms of numerical stability), and included the results of the additional simulations, we hope that the reviewer will be able to go back and read this section. Based on the objectives and our explanations for the varying sensitivity of the storm dynamics to the microphysics schemes used should now be more coherent. Recall in this section we are describing and evaluating a suite of properties and hazards and seek to illustrate where partial fidelity exists, how that relates to the depiction of the surface hazards.

Since the sensitivity to two lateral boundary data (ERA5 and ERA-Interim) is opposite to previous studies (many studies showed ERA5 data is improved on ERA-Interim), the authors need to evaluate both datasets with observations to show this is an exception that ERA5 performs worse than ERA-Interim. Otherwise, this will make people doubt if the simulations are done correctly.

Response: We refer the reviewer to both lines 409 – 416 in the original text, and Figure S2 of the supplementary materials where we have originally undertaken an analysis of sea level pressure, temperature at 2m, and specific humidity at 2m. Upon the reviewer's suggestion, however, we have remade and improved Figure S2. We now include a more comprehensive assessment of the initial conditions and direct comparison to rawinsonde observations (i.e. atmospheric sounding data collated by the University of Wyoming http://weather.uwyo.edu/upperair/sounding.html). We now have a figure panel with 5 separate plots, one each for sea level pressure, elevated mixed layer (EML), 500 hPa temperature, 500 hPa geopotential height, and 500 hPa relative humidity. We justify this in the following two paragraphs, and this is text that will be included in the revised manuscript. The figure (see new Figure 11, revised from the SM Figure S2) is now included in the main text rather than in the supplementary materials. This figure is a summary of our enhanced evaluation of the initial conditions. In order to introduce it we will add the following text to the manuscript:

> At near line 212: We also employ data from all 28 rawinsondes within the simulation domain in the fidelity assessment of the initial conditions from each reanalysis product and start time. In these analyses the conditions on two geopotential surfaces (700 hPa and 500 hPa) as derived using WRF real from the ERA5 and ERA-Interim reanalysis products are interpolated to these pressure levels using the wrf_interp program (available at: https://github.com/pick2510/wrf_interp) and the rawinsonde observations for the closest release time.
>
> At/near line 419: Evaluation of the initial conditions indicates a high-degree of similarity between the two reanalysis products on 26 and 28 June for most properties (Figure 11). However, as described above, development of an intense elevated mixed layer (EML, 700-500 hPa) over the central US that subsequently propagated eastwards (Shourd and Kaplan, 2021) appears to have been a key ingredient in development of this Derecho. Earlier work (Banacos and Ekster, 2010) employed a definition of an EML as a layer of depth > 200 hPa with both a steep lapse rate (temperature declines of over 8°C per km) and an increase in the RH with height. Figure 11 shows the lapse rate in the four sets of IC and indicates that while both data sets correctly (relative to output from NOAA WRF-Rapid Refresh model presented in Shourd and Kaplan,

Shepherd T.J., Letson F., Barthelmie R.J. and Pryor S.C. How well are hazards associated with derechos reproduced in regional climate simulations? *Natural Hazards and Earth System Sciences Discussions (*nhess-2021-373)

**Response to reviewer #1:**

2021) indicate relatively low lapse rates at 0000Z 26 June (when the region with the EML was displaced further west), using the combined definition of a strong lapse rate and a strong gradient of RH (a 20% difference across the layer), the EML is, in both reanalysis products, displaced too far north at 0000Z 28 June relative to NOAA WRF-Rapid Refresh model simulations presented in Shourd and Kaplan (2021). The EML is, however, more consistent (across the two components) and more coherent in space in ERA-Interim. This may provide a partial explanation for why simulations with ERA-Interim initial and lateral boundary conditions exhibit higher fidelity with respect to aspects of the Derecho.

Shepherd T.J., Letson F., Barthelmie R.J. and Pryor S.C. How well are hazards associated with derechos reproduced in regional climate simulations? *Natural Hazards and Earth System Sciences Discussions (*nhess-2021-373)

**Response to reviewer #1:**

[Figure]

Figure 11: (a) Spatial maps of sea level pressure (colored surface) generated by WRF real from the ERA5 and ERA-Interim reanalysis products used to initialize the model LBC and initial conditions. The black, red, and magenta lines are 2-m temperature of 295 K, 300 K, and 305 K respectively. The white line represents specific humidity at 2-m of 12.5 g/kg. (b) Filled contours of lapse rates (700-500 hPa) with the -9°C/km highlighted by the white outline. Also shown by the magenta isoline is the area in which the RH increased by 20% over this layer. (c) 500 hPa geopotential height in meters. (d) 500 hPa temperature in Kelvin. (e) 500 hPa relative humidity in %. Plots in (c), (d), and (e) contain rawinsonde observations (filled cicles). In all the plots, WRF real output is used from all 3 domains.

Shepherd T.J., Letson F., Barthelmie R.J. and Pryor S.C. How well are hazards associated with derechos reproduced in regional climate simulations? *Natural Hazards and Earth System Sciences Discussions (*nhess-2021-373)

**Response to reviewer #1:**

As a further assessment of the initial conditions, we have also examined MU-CAPE from soundings, for direct comparison against model MU-CAPE. Figure S3 and Figure S7 have been updated to include this data. Note that the scale for Figure S3 has changed from 0 – 5000 to 0 – 6000 J kg$^{-1}$.:

At/near line 420: MU-CAPE from the SHARPpy software (Blumberg et al., 2017) is defined slightly differently than in the python WRF analysis codes, in that it is the parcel with the maximum equivalent potential temperature in the lowest 400 mb, thus the values are not directly comparable. Nevertheless, high values are indicative of presence of significant CAPE. Consistent with past summaries of the environment in which the derecho was manifest, rawinsonde data from the two stations (KIAD (38.968N, -77.369E) and KWAL (38.018N and -75.236E)) within domain d03 indicate MU-CAPE values at tp-3 (from RADAR) (i.e. 0000 UTC 30 June) of 6871 J/kg and 4735 J/kg (Figure S3). The surface to 6 km shear at that time are 17.2 m/s and 11.5 m/s respectively, which is consistent with the relatively weak shear evident in the WRF ensemble members (Figure S7). MU-CAPE at KIAD and KWAL dropped to 51 and 60 J/kg, respectively in the 1200 UTC 30 June sounding. This further emphasizes the profound underestimation of CAPE consumption in the WRF ensemble during the passage of the derecho.

Shepherd T.J., Letson F., Barthelmie R.J. and Pryor S.C. How well are hazards associated with derechos reproduced in regional climate simulations? *Natural Hazards and Earth System Sciences Discussions (*nhess-2021-373)

**Response to reviewer #1:**

[Figure]

Figure S3: Spatial patterns of MU-CAPE at $t_p$-3 (i.e. 3 hours prior to the time of peak spatial extent of cREF > 40 dBZ during the Derecho period) over domain d03 for all ensemble members. These panels are also shown in Figure 12 of the main text but are included again here, enlarged for visibility. MU-CAPE as computed from the SHARPpy program based on rawinsonde data at tp-3 (define from RADAR) (i.e. 0000 UTC 30 June) at KIAD (38.968N, -77.369E) and KWAL (38.018N and -75.236E) are shown by the filled circles.

Shepherd T.J., Letson F., Barthelmie R.J. and Pryor S.C. How well are hazards associated with derechos reproduced in regional climate simulations? *Natural Hazards and Earth System Sciences Discussions (*nhess-2021-373)

**Response to reviewer #1:**

[Figure]

Figure S7: Total wind shear between the ground and 6000 m (S6) at $t_p$ (the time of peak spatial extent of cREF > 40 dBZ during the Derecho period) for each ensemble member. These panels are also shown in Figure 12 of the main text but are included again here, enlarged for visibility. Observed shear from the surface to 6 km at the KIAD (38.968N, -77.369E) and KWAL (38.018N and -75.236E) stations are shown by the red arrows.

Shepherd T.J., Letson F., Barthelmie R.J. and Pryor S.C. How well are hazards associated with derechos reproduced in regional climate simulations? *Natural Hazards and Earth System Sciences Discussions (*nhess-2021-373)

**Response to reviewer #1:**

**References**

Ahrens, B. and Leps, N. (2021). Sensitivity of Convection Permitting Simulations to Lateral Boundary Conditions in Idealised Experiments. *Earth and Space Science Open Archive*; Washington, Feb 28, 2021. DOI:10.1002/essoar.10506295.1

Banacos, P. C., & Ekster, M. L. (2010). The association of the elevated mixed layer with significant severe weather events in the northeastern United States. *Weather and Forecasting*, *25*(4), 1082-1102.

Blumberg, W. G., Halbert, K. T., Supinie, T. A., Marsh, P. T., Thompson, R. L., & Hart, J. A. (2017). SHARPpy: An open-source sounding analysis toolkit for the atmospheric sciences. *Bulletin of the American Meteorological Society*, *98*(8), 1625-1636.

Cattiaux, J., & Yiou, P. (2013). US heat waves of spring and summer 2012 from the flow-analogue perspective. *Bulletin of the American Meteorological Society*, *94*(9), S10-S13.

Cordeira, J. M., Metz, N. D., Howarth, M. E., & Galarneau Jr, T. J. (2017). Multiscale upstream and in situ precursors to the elevated mixed layer and high-impact weather over the Midwest United States. *Weather and Forecasting*, *32*(3), 905-923.

Du, J. 2011. NCEP/EMC 4KM Gridded Data (GRIB) Stage IV Data. Version 1.0. UCAR/NCAR - Earth Observing Laboratory. https://doi.org/10.5065/D6PG1QDD. Accessed 08 Feb 2022.

Fierro, A. O., Gao, J., Ziegler, C. L., Mansell, E. R., MacGorman, D. R., and Dembek, S. R. (2014). Evaluation of a cloud-scale lightning data assimilation technique and a 3DVAR method for the analysis and short-term forecast of the 29 June 2012 derecho event, *Monthly Weather Review*, 142, 183-202.

Hacker, J. P., Exby, J., Gill, D., Jimenez, I., Maltzahn, C., See, T., ... & Fossell, K. (2017). A containerized mesoscale model and analysis toolkit to accelerate classroom learning, collaborative research, and uncertainty quantification. *Bulletin of the American Meteorological Society*, *98*(6), 1129-1138.

Labriola, J., Snook, N., Xue, M., & Thomas, K. W. (2019). Forecasting the 8 May 2017 severe hail storm in Denver, Colorado, at a convection-allowing resolution: Understanding rimed ice treatments in multimoment microphysics schemes and their effects on hail size forecasts. *Monthly Weather Review*, 147(8), 3045-3068.

Li, R., Liu, L., Yang, G., Zhang, C., & Wang, B. (2016). Bitwise identical compiling setup: Prospective for reproducibility and reliability of Earth system modeling. *Geoscientific Model Development*, **9**(2), 731–748. https://doi.org/10.5194/gmd-9-731-2016

Lighezzolo, A., Martina A., Zigarán, G., Viscardi, D., Lio, E., Rodriguez, A., Montamat, I., et al. (2018). "WRF model-sensitivity experiments to computational environment changes." In *2018 IEEE Biennial Congress of Argentina (ARGENCON)*, pp. 1-7. IEEE. doi: 10.1109/ARGENCON.2018.8645986

Morrison, H., Milbrandt, J. A., Bryan, G. H., Ikeda, K., Tessendorf, S. A., and Thompson, G. (2015). Parameterization of Cloud Microphysics Based on the Prediction of Bulk Ice Particle Properties. Part II:

Shepherd T.J., Letson F., Barthelmie R.J. and Pryor S.C. How well are hazards associated with derechos reproduced in regional climate simulations? *Natural Hazards and Earth System Sciences Discussions* (nhess-2021-373)

**Response to reviewer #1:**

Case Study Comparisons with Observations and Other Schemes, *Journal of the Atmospheric Sciences*, 72, 312-339, 10.1175/jas-d-14-0066.1.

NOAA. The Historic Derecho of 29 June 2012; Service Assessment, U.S. Department of Commerce, NOAA National Weather Service: Silver Spring, MD, USA, 2013; p. 61

Shourd, K. N., & Kaplan, M. L. (2021). The Multiscale Dynamics of the 29 June 2012 Super Derecho. *Climate*, *9*(11), 155.

Witt, A., Eilts, M. D., Stumpf, G. J., Johnson, J. T., Mitchell, E. D. W., & Thomas, K. W. (1998). An enhanced hail detection algorithm for the WSR-88D. *Weather and Forecasting*, 13(2), 286-303.

Shepherd T.J., Letson F., Barthelmie R.J. and Pryor S.C. How well are hazards associated with derechos reproduced in regional climate simulations? *Natural Hazards and Earth System Sciences Discussions (*nhess-2021-373)

**Response to reviewer #1:**

Namelist files for each simulation

Morrison

```
&time_control
 run_days                    = 6,
 run_hours                   = 0,
 run_minutes                 = 0,
 run_seconds                 = 0,
 start_year                  = 2012,  2012, 2012,
 start_month                 = 06,   06,   06,
 start_day                   = 26,   26,   26,
 start_hour                  = 00,   00,   00,
 start_minute                = 00,   00,   00,
 start_second                = 00,   00,   00,
 end_year                    = 2012, 2012, 2012,
 end_month                   = 07,   07,   07,
 end_day                     = 02,   02,   02,
 end_hour                    = 00,   00,   00,
 end_minute                  = 00,   00,   00,
 end_second                  = 00,   00,   00,
 interval_seconds            = 21600
 input_from_file             = .true.,.true.,.true.,
 history_interval            = 60, 10, 10,
 frames_per_outfile          = 1,  1, 1,
 history_outname             =
"/data/derecho/WRF_output/NE_2012/wrfout/wrfout_d<domain>_<date>"
 restart                     = .false.,
 restart_interval            = 1440,
 override_restart_timers     = .true.,
 io_form_history             = 11
 io_form_restart             = 2
 io_form_input               = 2
 io_form_boundary            = 11
 io_form_auxinput2           = 11
 io_form_auxhist2            = 11
 debug_level                 = 10
 nocolons                    = .true.,
 auxinput4_inname            = "wrflowinp_d<domain>",
 auxinput4_interval          = 1440, 1440, 1440,
 io_form_auxinput4           = 2,
 auxinput1_inname            = "/data/derecho/met_files/ERA5/met_em.d<domain>.<date>"
 iofields_filename           = "my_file_d01.txt", "my_file_d02.txt", "my_file_d03.txt",
 ignore_iofields_warning     = .true.,
 auxhist1_outname            =
"/data/derecho/WRF_output/NE_2012/aux1/auxhist1_d<domain>_<date>"
 auxhist1_interval           = 60, 60, 60,
```

Shepherd T.J., Letson F., Barthelmie R.J. and Pryor S.C. How well are hazards associated with derechos reproduced in regional climate simulations? *Natural Hazards and Earth System Sciences Discussions* (nhess-2021-373)

**Response to reviewer #1:**

```
 frames_per_auxhist1              = 1, 1, 1,
 io_form_auxhist1              = 11,
 output_diagnostics           = 1,
 auxhist3_outname             =
 "/data/derecho/WRF_output/NE_2012/wrfout/wrfxtrm_d<domain>_<date>"
 auxhist3_interval            = 60, 10, 10,
 frames_per_auxhist3             = 1, 1, 1,
 io_form_auxhist3             = 11,
 /
 &domains
 time_step               = 30,
 time_step_fract_num          = 0,
 time_step_fract_den          = 1,
 max_dom             = 3,
 e_we              = 175,   262,   295,
 e_sn              = 175,   262,   295,
 e_vert            = 41,    41,   41,
 p_top_requested          = 5000,
 sfcp_to_sfcp          = .true.,
 num_metgrid_levels           = 38,
 num_metgrid_soil_levels          = 4,
 dx             = 12000, 4000, 1333.33,
 dy             = 12000, 4000, 1333.33,
 grid_id            = 1,    2,   3,
 parent_id            = 1,    1,   2,
 i_parent_start          = 1,    60,   105,
 j_parent_start          = 1,    35,   75,
 parent_grid_ratio          = 1,    3,   3,
 parent_time_step_ratio          = 1,    3,   3,
 feedback           = 0,
 max_ts_locs          = 0,
 eta_levels          = 1.0000 , 0.9958 , 0.9916 , 0.9874 , 0.9832 ,
                     0.9790 , 0.9749 , 0.9707 , 0.9661 , 0.9609 ,
                     0.9549 , 0.9480 , 0.9398 , 0.9303 , 0.9189 ,
                     0.9054 , 0.8894 , 0.8704 , 0.8481 , 0.8221 ,
                     0.7922 , 0.7583 , 0.7205 , 0.6791 , 0.6346 ,
                     0.5877 , 0.5393 , 0.4900 , 0.4407 , 0.3922 ,
                     0.3450 , 0.2996 , 0.2564 , 0.2156 , 0.1773 ,
                     0.1417 , 0.1086 , 0.0755 , 0.0475 , 0.0224 ,
                     0.0000,
 /
 &physics
 mp_physics             = 10,    10,    10,
 ra_lw_physics          = 1,    1,   1,
 ra_sw_physics          = 1,    1,   1,
 radt            = 10,    10,   10,
 sf_sfclay_physics          = 1,    1,   1,
```

Shepherd T.J., Letson F., Barthelmie R.J. and Pryor S.C. How well are hazards associated with derechos reproduced in regional climate simulations? *Natural Hazards and Earth System Sciences Discussions (*nhess-2021-373)

**Response to reviewer #1:**

```
sf_surface_physics          = 2,   2,   2,
bl_pbl_physics              = 5,   5,   5,
bldt                    = 0,   0,   0,
cu_physics                  = 1,   0,   0,
cudt                    = 5,
isfflx                  = 1,
ifsnow                    = 1,
icloud                    = 1,
surface_input_source        = 3,
num_soil_layers            = 4,
num_land_cat               = 21,
sf_urban_physics            = 0,   0,   0,
bl_mynn_tkebudget            = 1,   1,   1,
bl_mynn_tkeadvect             = .true., .true., .true.,
rdmaxalb                = .false.,
sst_update              = 1,
tmn_update                = 1,
usemonalb                 = .true.,
lagday                  = 150,
sst_skin                = 1,
slope_rad               = 1,  1,  1,
do_radar_ref              = 1,
prec_acc_dt               = 60., 10., 10.,
fractional_seaice           = 1,
seaice_threshold            = 0.,
/
&noah_mp
dveg                = 4,
opt_crs             = 1,
opt_btr             = 2,
opt_run              = 3,
opt_sfc             = 1,
opt_frz             = 1,
opt_inf             = 1,
opt_rad              = 3,
opt_alb             = 2,
opt_snf             = 4,
opt_tbot             = 1,
opt_stc             = 3,
/
&dynamics
w_damping               = 1,
diff_opt                = 1,    1,    1,
km_opt                  = 4,    4,    4,
diff_6th_opt              = 0,    0,    0,
diff_6th_factor            = 0.12,  0.12,  0.12,
base_temp               = 290.
```

Shepherd T.J., Letson F., Barthelmie R.J. and Pryor S.C. How well are hazards associated with derechos reproduced in regional climate simulations? *Natural Hazards and Earth System Sciences Discussions (*nhess-2021-373)

**Response to reviewer #1:**

```
damp_opt                  = 0,
zdamp                     = 5000.,  5000.,  5000.,
dampcoef                  = 0.01,   0.01,  0.01,
khdif                   = 0,    0,
kvdif                   = 0,    0,
non_hydrostatic           = .true., .true., .true.,
/
&bdy_control
spec_bdy_width            = 5,
spec_zone               = 1,
relax_zone              = 4,
spec_exp                = 0.13
specified             = .true., .false., .false.,
nested                = .false., .true., .true.,
/
&grib2
/
&namelist_quilt
nio_tasks_per_group = 0,
nio_groups = 1,
/
```

Shepherd T.J., Letson F., Barthelmie R.J. and Pryor S.C. How well are hazards associated with derechos reproduced in regional climate simulations? *Natural Hazards and Earth System Sciences Discussions (*nhess-2021-373)

**Response to reviewer #1:**

**Morrison + Hail**
```
&time_control
run_days                   = 6,
run_hours                  = 0,
run_minutes                = 0,
run_seconds                = 0,
start_year                 = 2012, 2012, 2012,
start_month                = 06,   06,   06,
start_day                  = 26,   26,   26,
start_hour                 = 00,   00,   00,
start_minute               = 00,   00,   00,
start_second               = 00,   00,   00,
end_year                   = 2012, 2012, 2012,
end_month                  = 07,   07,   07,
end_day                    = 02,   02,   02,
end_hour                   = 00,   00,   00,
end_minute                 = 00,   00,   00,
end_second                 = 00,   00,   00,
interval_seconds           = 21600
input_from_file            = .true.,.true.,.true.,
history_interval           = 60, 10, 10,
frames_per_outfile         = 1,  1, 1,
history_outname            =
"/data/derecho/WRF_output/NE_2012/wrfout/wrfout_d<domain>_<date>"
restart                    = .false.,
restart_interval           = 1440,
override_restart_timers    = .true.,
io_form_history            = 11
io_form_restart            = 2
io_form_input              = 2
io_form_boundary           = 11
io_form_auxinput2          = 11
io_form_auxhist2           = 11
debug_level                = 10
nocolons                   = .true.,
auxinput4_inname           = "wrflowinp_d<domain>",
auxinput4_interval         = 1440, 1440, 1440,
io_form_auxinput4          = 2,
auxinput1_inname           = "/data/derecho/met_files/ERA5/met_em.d<domain>.<date>"
iofields_filename          = "my_file_d01.txt", "my_file_d02.txt", "my_file_d03.txt",
ignore_iofields_warning    = .true.,
auxhist1_outname           =
"/data/derecho/WRF_output/NE_2012/aux1/auxhist1_d<domain>_<date>"
auxhist1_interval          = 60, 60, 60,
frames_per_auxhist1        = 1, 1, 1,
io_form_auxhist1           = 11,
output_diagnostics         = 1,
```

Shepherd T.J., Letson F., Barthelmie R.J. and Pryor S.C. How well are hazards associated with derechos reproduced in regional climate simulations? *Natural Hazards and Earth System Sciences Discussions (*nhess-2021-373)

**Response to reviewer #1:**

```
 auxhist3_outname              =
"/data/derecho/WRF_output/NE_2012/wrfout/wrfxtrm_d<domain>_<date>"
 auxhist3_interval           = 60, 10, 10,
 frames_per_auxhist3           = 1, 1, 1,
 io_form_auxhist3              = 11,
/
 &domains
 time_step                = 30,
 time_step_fract_num           = 0,
 time_step_fract_den           = 1,
 max_dom                  = 3,
 e_we                  = 175,   262,   295,
 e_sn                  = 175,   262,   295,
 e_vert                = 41,    41,   41,
 p_top_requested             = 5000,
 sfcp_to_sfcp              = .true.,
 num_metgrid_levels            = 38,
 num_metgrid_soil_levels        = 4,
 dx                   = 12000, 4000, 1333.33,
 dy                   = 12000, 4000, 1333.33,
 grid_id               = 1,    2,   3,
 parent_id             = 1,    1,   2,
 i_parent_start           = 1,    60,  105,
 j_parent_start           = 1,    35,  75,
 parent_grid_ratio          = 1,    3,   3,
 parent_time_step_ratio        = 1,    3,   3,
 feedback               = 0,
 max_ts_locs             = 0,
 eta_levels              = 1.0000 , 0.9958 , 0.9916 , 0.9874 , 0.9832 ,
                 0.9790 , 0.9749 , 0.9707 , 0.9661 , 0.9609 ,
                 0.9549 , 0.9480 , 0.9398 , 0.9303 , 0.9189 ,
                 0.9054 , 0.8894 , 0.8704 , 0.8481 , 0.8221 ,
                 0.7922 , 0.7583 , 0.7205 , 0.6791 , 0.6346 ,
                 0.5877 , 0.5393 , 0.4900 , 0.4407 , 0.3922 ,
                 0.3450 , 0.2996 , 0.2564 , 0.2156 , 0.1773 ,
                 0.1417 , 0.1086 , 0.0755 , 0.0475 , 0.0224 ,
                 0.0000,
/
 &physics
 mp_physics                = 10,    10,    10,
 ra_lw_physics             = 1,    1,   1,
 ra_sw_physics             = 1,    1,   1,
 radt                  = 10,    10,   10,
 sf_sfclay_physics           = 1,    1,   1,
 sf_surface_physics           = 2,    2,   2,
 bl_pbl_physics             = 5,    5,   5,
 bldt                  = 0,    0,   0,
```

Shepherd T.J., Letson F., Barthelmie R.J. and Pryor S.C. How well are hazards associated with derechos reproduced in regional climate simulations? *Natural Hazards and Earth System Sciences Discussions (*nhess-2021-373)

**Response to reviewer #1:**

```
cu_physics              = 1,    0,    0,
cudt               = 5,
isfflx             = 1,
ifsnow              = 1,
icloud              = 1,
surface_input_source       = 3,
num_soil_layers          = 4,
num_land_cat           = 21,
sf_urban_physics          = 0,    0,    0,
bl_mynn_tkebudget           = 1,    1,    1,
bl_mynn_tkeadvect            = .true., .true., .true.,
rdmaxalb              = .false.,
sst_update            = 1,
tmn_update             = 1,
usemonalb             = .true.,
lagday             = 150,
sst_skin            = 1,
slope_rad             = 1,  1,  1,
do_radar_ref            = 1,
prec_acc_dt             = 60., 10., 10.,
fractional_seaice           = 1,
seaice_threshold           = 0.,
hail_opt            = 1,
/
&noah_mp
dveg               = 4,
opt_crs             = 1,
opt_btr             = 2,
opt_run              = 3,
opt_sfc             = 1,
opt_frz             = 1,
opt_inf             = 1,
opt_rad             = 3,
opt_alb             = 2,
opt_snf             = 4,
opt_tbot             = 1,
opt_stc             = 3,
/
&dynamics
w_damping              = 1,
diff_opt             = 1,    1,    1,
km_opt              = 4,    4,    4,
diff_6th_opt             = 0,    0,    0,
diff_6th_factor             = 0.12,  0.12,  0.12,
base_temp             = 290.
damp_opt             = 0,
zdamp              = 5000., 5000., 5000.,
```

Shepherd T.J., Letson F., Barthelmie R.J. and Pryor S.C. How well are hazards associated with derechos reproduced in regional climate simulations? *Natural Hazards and Earth System Sciences Discussions (*nhess-2021-373)

**Response to reviewer #1:**

```
dampcoef                  = 0.01,   0.01,  0.01,
khdif                  = 0,     0,
kvdif                  = 0,     0,
non_hydrostatic           = .true., .true., .true.,
/
&bdy_control
spec_bdy_width            = 5,
spec_zone              = 1,
relax_zone              = 4,
spec_exp                = 0.13
specified              = .true., .false., .false.,
nested                 = .false., .true., .true.,
/
&grib2
/
&namelist_quilt
nio_tasks_per_group = 0,
nio_groups = 1,
/
```

Shepherd T.J., Letson F., Barthelmie R.J. and Pryor S.C. How well are hazards associated with derechos reproduced in regional climate simulations? *Natural Hazards and Earth System Sciences Discussions (*nhess-2021-373)

**Response to reviewer #1:**

**Morrison-intel (Run on NERSC Cori)**

```
&time_control
 run_days                    = 6,
 run_hours                   = 0,
 run_minutes                 = 0,
 run_seconds                 = 0,
 start_year                  = 2012, 2012, 2012,
 start_month                 = 06,  06,  06,
 start_day                   = 26,  26,  26,
 start_hour                  = 00,  00,  00,
 start_minute                = 00,  00,  00,
 start_second                = 00,  00,  00,
 end_year                    = 2012, 2012, 2012,
 end_month                   = 07,  07,  07,
 end_day                     = 02,  02,  02,
 end_hour                    = 00,  00,  00,
 end_minute                  = 00,  00,  00,
 end_second                  = 00,  00,  00,
 interval_seconds            = 21600
 input_from_file             = .true.,.true.,.true.,
 history_interval            = 60, 10, 10,
 frames_per_outfile          = 1, 1, 1,
 history_outname             =
"/global/cscratch1/sd/tshep/WRF_derecho/WRFV3/derecho4/files/wrfout/wrfout_d<domain>_<date>"
 restart                     = .false.,
 restart_interval            = 1440,
 override_restart_timers     = .true.,
 io_form_history             = 11
 io_form_restart             = 2
 io_form_input               = 2
 io_form_boundary            = 11
 io_form_auxinput2           = 11
 io_form_auxhist2            = 11
 debug_level                 = 10
 nocolons                    = .true.,
 auxinput4_inname            = "wrflowinp_d<domain>",
 auxinput4_interval          = 1440, 1440, 1440,
 io_form_auxinput4           = 2,
 auxinput1_inname            =
"/global/cscratch1/sd/tshep/WPS_output/derecho/ERA5/met_em.d<domain>.<date>"
 iofields_filename           = "my_file_d01.txt", "my_file_d02.txt", "my_file_d03.txt",
 ignore_iofields_warning     = .true.,
 auxhist1_outname            =
"/global/cscratch1/sd/tshep/WRF_derecho/WRFV3/derecho4/files/wrfout/auxhist1_d<domain>_<date>"
 auxhist1_interval           = 60, 60, 60,
 frames_per_auxhist1         = 1, 1, 1,
 io_form_auxhist1            = 11,
```

Shepherd T.J., Letson F., Barthelmie R.J. and Pryor S.C. How well are hazards associated with derechos reproduced in regional climate simulations? *Natural Hazards and Earth System Sciences Discussions (*nhess-2021-373)

**Response to reviewer #1:**

```
output_diagnostics          =  1,
auxhist3_outname            =
"/global/cscratch1/sd/tshep/WRF_derecho/WRFV3/derecho4/files/wrfout/wrfxtrm_d<domain>_<date>"
auxhist3_interval           = 60, 10, 10,
frames_per_auxhist3         = 1, 1, 1,
io_form_auxhist3            = 11,
/
&domains
time_step                   = 30,
time_step_fract_num         = 0,
time_step_fract_den         = 1,
max_dom                     = 3,
e_we                        = 175,   262,   295,
e_sn                        = 175,   262,   295,
e_vert                      = 41,    41,    41,
p_top_requested             = 5000,
sfcp_to_sfcp                = .true.,
num_metgrid_levels          = 38,
num_metgrid_soil_levels     = 4,
dx                          = 12000, 4000, 1333.33,
dy                          = 12000, 4000, 1333.33,
grid_id                     = 1,    2,   3,
parent_id                   = 1,    1,   2,
i_parent_start              = 1,    60,   105,
j_parent_start              = 1,    35,   75,
parent_grid_ratio           = 1,    3,   3,
parent_time_step_ratio      = 1,    3,   3,
feedback                    = 0,
max_ts_locs                 = 0,
eta_levels                  = 1.0000 , 0.9958 , 0.9916 , 0.9874 , 0.9832 ,
                              0.9790 , 0.9749 , 0.9707 , 0.9661 , 0.9609 ,
                              0.9549 , 0.9480 , 0.9398 , 0.9303 , 0.9189 ,
                              0.9054 , 0.8894 , 0.8704 , 0.8481 , 0.8221 ,
                              0.7922 , 0.7583 , 0.7205 , 0.6791 , 0.6346 ,
                              0.5877 , 0.5393 , 0.4900 , 0.4407 , 0.3922 ,
                              0.3450 , 0.2996 , 0.2564 , 0.2156 , 0.1773 ,
                              0.1417 , 0.1086 , 0.0755 , 0.0475 , 0.0224 ,
                              0.0000,
/
&physics
mp_physics                  = 10,   10,   10,    5,    5,
ra_lw_physics               = 1,    1,    1,    1,    1,
ra_sw_physics               = 1,    1,    1,    1,    1,
radt                        = 10,   10,   10,   10,   10,
sf_sfclay_physics           = 1,    1,    1,    1,    1,
sf_surface_physics          = 2,    2,    2,    2,    2,
bl_pbl_physics              = 5,    5,    5,    5,    5,
```

Shepherd T.J., Letson F., Barthelmie R.J. and Pryor S.C. How well are hazards associated with derechos reproduced in regional climate simulations? *Natural Hazards and Earth System Sciences Discussions* (nhess-2021-373)

**Response to reviewer #1:**

```
bldt                     = 0,   0,   0,   0,   0,
cu_physics                = 1,   0,   0,   0,   0,
cudt                     = 5,
isfflx                   = 1,
ifsnow                    = 1,
icloud                    = 1,
surface_input_source        = 3,
num_soil_layers            = 4,
num_land_cat               = 21,
sf_urban_physics            = 0,   0,   0,   0,   0,
bl_mynn_tkebudget            = 1,   1,   1,   1,   1,
bl_mynn_tkeadvect            = .true., .true., .true., .true., .true.,
rdmaxalb                  = .false.,
sst_update                = 1,
tmn_update                 = 1,
usemonalb                  = .true.,
lagday                   = 150,
sst_skin                 = 1,
slope_rad                 = 1,  1,  1,  1,  1,
do_radar_ref               = 1,
prec_acc_dt                = 60., 10., 10., 10., 10.,
fractional_seaice            = 1,
seaice_threshold             = 0.,
/
&noah_mp
dveg                     = 4,
opt_crs                  = 1,
opt_btr                  = 2,
opt_run                   = 3,
opt_sfc                  = 1,
opt_frz                  = 1,
opt_inf                  = 1,
opt_rad                  = 3,
opt_alb                  = 2,
opt_snf                  = 4,
opt_tbot                 = 1,
opt_stc                  = 3,
/
&dynamics
w_damping                  = 1,
diff_opt                 = 1,    1,    1,    1,    1,
km_opt                   = 4,   4,    4,    4,    4,
diff_6th_opt               = 0,   0,   0,   0,    0,
diff_6th_factor             = 0.12,  0.12,  0.12,  0.12,  0.12,
base_temp                 = 290.
damp_opt                  = 0,
zdamp                    = 5000., 5000., 5000., 5000., 5000.,
```

Shepherd T.J., Letson F., Barthelmie R.J. and Pryor S.C. How well are hazards associated with derechos reproduced in regional climate simulations? *Natural Hazards and Earth System Sciences Discussions* (nhess-2021-373)

**Response to reviewer #1:**

```
dampcoef                    = 0.01,   0.01, 0.01, 0.01, 0.01,
khdif                    = 0,    0,
kvdif                    = 0,    0,
non_hydrostatic             = .true., .true., .true., .true., .true.,
/
&bdy_control
spec_bdy_width              = 5,
spec_zone                = 1,
relax_zone               = 4,
spec_exp                = 0.13
specified                = .true., .false., .false., .false., .false.,
nested                   = .false., .true., .true.,  .true.,  .true.,
/
&grib2
/
&namelist_quilt
nio_tasks_per_group = 0,
nio_groups = 1,
/
```

Shepherd T.J., Letson F., Barthelmie R.J. and Pryor S.C. How well are hazards associated with derechos reproduced in regional climate simulations? *Natural Hazards and Earth System Sciences Discussions (*nhess-2021-373)

**Response to reviewer #1:**

**Morrison-intel+hail (Run on NERSC Cori)**

```
&time_control
 run_days                 = 6,
 run_hours                = 0,
 run_minutes              = 0,
 run_seconds              = 0,
 start_year               = 2012, 2012, 2012,
 start_month              = 06,   06,   06,
 start_day                = 26,   26,   26,
 start_hour               = 00,   00,   00,
 start_minute             = 00,   00,   00,
 start_second             = 00,   00,   00,
 end_year                 = 2012, 2012, 2012,
 end_month                = 07,   07,   07,
 end_day                  = 02,   02,   02,
 end_hour                 = 00,   00,   00,
 end_minute               = 00,   00,   00,
 end_second               = 00,   00,   00,
 interval_seconds         = 21600
 input_from_file          = .true.,.true.,.true.,
 history_interval         = 60, 10, 10,
 frames_per_outfile       = 1,  1, 1,
 history_outname          =
"/global/cscratch1/sd/tshep/WRF_derecho/WRFV3/derecho4/files/wrfout/wrfout_d<domain>_<date>"
 restart                  = .false.,
 restart_interval         = 1440,
 override_restart_timers   = .true.,
 io_form_history          = 11
 io_form_restart          = 2
 io_form_input            = 2
 io_form_boundary         = 11
 io_form_auxinput2        = 11
 io_form_auxhist2         = 11
 debug_level              = 10
 nocolons                 = .true.,
 auxinput4_inname         = "wrflowinp_d<domain>",
 auxinput4_interval       = 1440, 1440, 1440,
 io_form_auxinput4        = 2,
 auxinput1_inname         =
"/global/cscratch1/sd/tshep/WPS_output/derecho/ERA5/met_em.d<domain>.<date>"
 iofields_filename        = "my_file_d01.txt", "my_file_d02.txt", "my_file_d03.txt",
 ignore_iofields_warning   = .true.,
 auxhist1_outname         =
"/global/cscratch1/sd/tshep/WRF_derecho/WRFV3/derecho4/files/wrfout/auxhist1_d<domain>_<date>"
 auxhist1_interval        = 60, 60, 60,
 frames_per_auxhist1      = 1,  1, 1,
 io_form_auxhist1         = 11,
```

Shepherd T.J., Letson F., Barthelmie R.J. and Pryor S.C. How well are hazards associated with derechos reproduced in regional climate simulations? *Natural Hazards and Earth System Sciences Discussions (*nhess-2021-373)

**Response to reviewer #1:**

```
output_diagnostics          =  1,
auxhist3_outname            =
"/global/cscratch1/sd/tshep/WRF_derecho/WRFV3/derecho4/files/wrfout/wrfxtrm_d<domain>_<date>"
auxhist3_interval           = 60, 10, 10,
frames_per_auxhist3         = 1, 1, 1,
io_form_auxhist3            = 11,
/
&domains
time_step                   = 30,
time_step_fract_num         = 0,
time_step_fract_den         = 1,
max_dom                     = 3,
e_we                        = 175,   262,   295,
e_sn                        = 175,   262,   295,
e_vert                      = 41,    41,    41,
p_top_requested             = 5000,
sfcp_to_sfcp                = .true.,
num_metgrid_levels          = 38,
num_metgrid_soil_levels     = 4,
dx                          = 12000, 4000, 1333.33,
dy                          = 12000, 4000, 1333.33,
grid_id                     = 1,    2,   3,
parent_id                   = 1,    1,   2,
i_parent_start              = 1,    60,   105,
j_parent_start              = 1,    35,   75,
parent_grid_ratio           = 1,    3,   3,
parent_time_step_ratio      = 1,    3,   3,
feedback                    = 0,
max_ts_locs                 = 0,
eta_levels                  = 1.0000 , 0.9958 , 0.9916 , 0.9874 , 0.9832 ,
                              0.9790 , 0.9749 , 0.9707 , 0.9661 , 0.9609 ,
                              0.9549 , 0.9480 , 0.9398 , 0.9303 , 0.9189 ,
                              0.9054 , 0.8894 , 0.8704 , 0.8481 , 0.8221 ,
                              0.7922 , 0.7583 , 0.7205 , 0.6791 , 0.6346 ,
                              0.5877 , 0.5393 , 0.4900 , 0.4407 , 0.3922 ,
                              0.3450 , 0.2996 , 0.2564 , 0.2156 , 0.1773 ,
                              0.1417 , 0.1086 , 0.0755 , 0.0475 , 0.0224 ,
                              0.0000,
/
&physics
mp_physics                  = 10,   10,   10,    5,   5,
ra_lw_physics               = 1,    1,    1,    1,    1,
ra_sw_physics               = 1,    1,    1,    1,    1,
radt                        = 10,   10,   10,   10,   10,
sf_sfclay_physics           = 1,    1,    1,    1,    1,
sf_surface_physics          = 2,    2,    2,    2,    2,
bl_pbl_physics              = 5,    5,    5,    5,    5,
```

Shepherd T.J., Letson F., Barthelmie R.J. and Pryor S.C. How well are hazards associated with derechos reproduced in regional climate simulations? *Natural Hazards and Earth System Sciences Discussions (*nhess-2021-373)

**Response to reviewer #1:**

```
bldt                    = 0,   0,   0,   0,   0,
cu_physics               = 1,   0,   0,   0,   0,
cudt                    = 5,
isfflx                  = 1,
ifsnow                   = 1,
icloud                   = 1,
surface_input_source        = 3,
num_soil_layers            = 4,
num_land_cat              = 21,
sf_urban_physics           = 0,   0,   0,   0,   0,
bl_mynn_tkebudget           = 1,   1,   1,   1,   1,
bl_mynn_tkeadvect           = .true., .true., .true., .true., .true.,
rdmaxalb                 = .false.,
sst_update               = 1,
tmn_update               = 1,
usemonalb                = .true.,
lagday                  = 150,
sst_skin                 = 1,
slope_rad                = 1,  1,  1,  1,  1,
do_radar_ref              = 1,
prec_acc_dt               = 60., 10., 10., 10., 10.,
fractional_seaice           = 1,
seaice_threshold            = 0.,
hail_opt                 = 1,
/
&noah_mp
dveg                    = 4,
opt_crs                  = 1,
opt_btr                  = 2,
opt_run                  = 3,
opt_sfc                  = 1,
opt_frz                  = 1,
opt_inf                  = 1,
opt_rad                  = 3,
opt_alb                  = 2,
opt_snf                  = 4,
opt_tbot                 = 1,
opt_stc                 = 3,
/
&dynamics
w_damping                 = 1,
diff_opt                = 1,    1,    1,    1,    1,
km_opt                  = 4,    4,    4,    4,    4,
diff_6th_opt              = 0,   0,   0,   0,   0,
diff_6th_factor            = 0.12,  0.12,  0.12,  0.12,  0.12,
base_temp                = 290.
damp_opt                 = 0,
```

Shepherd T.J., Letson F., Barthelmie R.J. and Pryor S.C. How well are hazards associated with derechos reproduced in regional climate simulations? *Natural Hazards and Earth System Sciences Discussions (*nhess-2021-373)

**Response to reviewer #1:**

```
zdamp                      = 5000.,  5000.,  5000.,  5000.,  5000.,
dampcoef                    = 0.01,   0.01,  0.01,   0.01,   0.01,
khdif                   = 0,     0,
kvdif                   = 0,     0,
non_hydrostatic             = .true., .true., .true., .true., .true.,
/
&bdy_control
spec_bdy_width              = 5,
spec_zone               = 1,
relax_zone              = 4,
spec_exp                = 0.13
specified               = .true., .false., .false., .false., .false.,
nested                  = .false., .true., .true.,  .true.,  .true.,
/
&grib2
/
&namelist_quilt
nio_tasks_per_group = 0,
nio_groups = 1,
/
```

Shepherd T.J., Letson F., Barthelmie R.J. and Pryor S.C. How well are hazards associated with derechos reproduced in regional climate simulations? *Natural Hazards and Earth System Sciences Discussions* (nhess-2021-373)

**Response to reviewer #1:**

**Milbrandt 26 June 00Z namelist**
```
&time_control
 run_days                  = 6,
 run_hours                 = 0,
 run_minutes               = 0,
 run_seconds               = 0,
 start_year                = 2012, 2012, 2012,
 start_month               = 06,   06,   06,
 start_day                 = 26,   26,   26,
 start_hour                = 00,   00,   00,
 start_minute              = 00,   00,   00,
 start_second              = 00,   00,   00,
 end_year                  = 2012, 2012, 2012,
 end_month                 = 07,   07,   07,
 end_day                   = 02,   02,   02,
 end_hour                  = 00,   00,   00,
 end_minute                = 00,   00,   00,
 end_second                = 00,   00,   00,
 interval_seconds          = 21600
 input_from_file           = .true.,.true.,.true.,
 history_interval          = 60, 10, 10,
 frames_per_outfile        = 1,  1, 1,
 history_outname           =
"/data/derecho/WRF_output/NE_2012/wrfout/wrfout_d<domain>_<date>"
 restart                   = .false.,
 restart_interval          = 1440,
 override_restart_timers   = .true.,
 io_form_history           = 11
 io_form_restart           = 2
 io_form_input             = 2
 io_form_boundary          = 11
 io_form_auxinput2         = 11
 io_form_auxhist2          = 11
 debug_level               = 10
 nocolons                  = .true.,
 auxinput4_inname          = "wrflowinp_d<domain>",
 auxinput4_interval        = 1440, 1440, 1440,
 io_form_auxinput4         = 2,
 auxinput1_inname          = "/data/derecho/met_files/ERA5/met_em.d<domain>.<date>"
 iofields_filename         = "my_file_d01.txt", "my_file_d02.txt", "my_file_d03.txt",
 ignore_iofields_warning   = .true.,
 auxhist1_outname          =
"/data/derecho/WRF_output/NE_2012/aux1/auxhist1_d<domain>_<date>"
 auxhist1_interval         = 60, 60, 60,
 frames_per_auxhist1       = 1, 1, 1,
 io_form_auxhist1          = 11,
 output_diagnostics        = 1,
```

Shepherd T.J., Letson F., Barthelmie R.J. and Pryor S.C. How well are hazards associated with derechos reproduced in regional climate simulations? *Natural Hazards and Earth System Sciences Discussions* (nhess-2021-373)

**Response to reviewer #1:**

```
 auxhist3_outname              =
"/data/derecho/WRF_output/NE_2012/wrfout/wrfxtrm_d<domain>_<date>"
 auxhist3_interval            = 60, 10, 10,
 frames_per_auxhist3           = 1, 1, 1,
 io_form_auxhist3             = 11,
/
 &domains
 time_step                 = 30,
 time_step_fract_num          = 0,
 time_step_fract_den          = 1,
 max_dom                  = 3,
 e_we                   = 175,   262,   295,
 e_sn                   = 175,   262,   295,
 e_vert                  = 41,    41,   41,
 p_top_requested             = 5000,
 sfcp_to_sfcp               = .true.,
 num_metgrid_levels            = 38,
 num_metgrid_soil_levels         = 4,
 dx                    = 12000, 4000, 1333.33,
 dy                    = 12000, 4000, 1333.33,
 grid_id                 = 1,    2,   3,
 parent_id                = 1,    1,   2,
 i_parent_start              = 1,    60,  105,
 j_parent_start              = 1,    35,  75,
 parent_grid_ratio            = 1,    3,   3,
 parent_time_step_ratio          = 1,    3,   3,
 feedback                 = 0,
 max_ts_locs               = 0,
 eta_levels                = 1.0000 , 0.9958 , 0.9916 , 0.9874 , 0.9832 ,
                  0.9790 , 0.9749 , 0.9707 , 0.9661 , 0.9609 ,
                  0.9549 , 0.9480 , 0.9398 , 0.9303 , 0.9189 ,
                  0.9054 , 0.8894 , 0.8704 , 0.8481 , 0.8221 ,
                  0.7922 , 0.7583 , 0.7205 , 0.6791 , 0.6346 ,
                  0.5877 , 0.5393 , 0.4900 , 0.4407 , 0.3922 ,
                  0.3450 , 0.2996 , 0.2564 , 0.2156 , 0.1773 ,
                  0.1417 , 0.1086 , 0.0755 , 0.0475 , 0.0224 ,
                  0.0000,
/
 &physics
 mp_physics                = 9,    9,   9,
 ra_lw_physics              = 1,    1,   1,
 ra_sw_physics              = 1,    1,   1,
 radt                   = 10,   10,   10,
 sf_sfclay_physics            = 1,    1,   1,
 sf_surface_physics            = 2,    2,   2,
 bl_pbl_physics              = 5,    5,   5,
 bldt                   = 0,    0,   0,
```

Shepherd T.J., Letson F., Barthelmie R.J. and Pryor S.C. How well are hazards associated with derechos reproduced in regional climate simulations? *Natural Hazards and Earth System Sciences Discussions (*nhess-2021-373)

**Response to reviewer #1:**

```
cu_physics                    = 1,    0,    0,
cudt                  = 5,
isfflx                = 1,
ifsnow                 = 1,
icloud                 = 1,
surface_input_source           = 3,
num_soil_layers             = 4,
num_land_cat               = 21,
sf_urban_physics             = 0,    0,    0,
bl_mynn_tkebudget              = 1,    1,    1,
bl_mynn_tkeadvect               = .true., .true., .true.,
rdmaxalb                 = .false.,
sst_update               = 1,
tmn_update                = 1,
usemonalb                 = .true.,
lagday                 = 150,
sst_skin               = 1,
slope_rad                = 1,  1,  1,
prec_acc_dt                 = 60., 10., 10.,
fractional_seaice               = 1,
seaice_threshold               = 0.,
/
&noah_mp
dveg                  = 4,
opt_crs                = 1,
opt_btr                = 2,
opt_run                 = 3,
opt_sfc                = 1,
opt_frz                = 1,
opt_inf                = 1,
opt_rad                = 3,
opt_alb                = 2,
opt_snf                = 4,
opt_tbot               = 1,
opt_stc                = 3,
/
&dynamics
w_damping                  = 1,
diff_opt               = 1,    1,    1,
km_opt                = 4,    4,    4,
diff_6th_opt                = 0,    0,    0,
diff_6th_factor               = 0.12,  0.12,  0.12,
base_temp               = 290.
damp_opt                = 0,
zdamp                 = 5000., 5000., 5000.,
dampcoef                = 0.01,   0.01, 0.01,
khdif                 = 0,    0,
```

Shepherd T.J., Letson F., Barthelmie R.J. and Pryor S.C. How well are hazards associated with derechos reproduced in regional climate simulations? *Natural Hazards and Earth System Sciences Discussions* *(*nhess-2021-373)

**Response to reviewer #1:**

```
kvdif                     = 0,    0,
non_hydrostatic             = .true., .true., .true.,
/
&bdy_control
spec_bdy_width               = 5,
spec_zone                = 1,
relax_zone               = 4,
spec_exp                 = 0.13
specified                = .true., .false., .false.,
nested                   = .false., .true., .true.,
/
&grib2
/
&namelist_quilt
nio_tasks_per_group = 0,
nio_groups = 1,
/
```

Shepherd T.J., Letson F., Barthelmie R.J. and Pryor S.C. How well are hazards associated with derechos reproduced in regional climate simulations? *Natural Hazards and Earth System Sciences Discussions (*nhess-2021-373)

**Response to reviewer #1:**

**Milbrandt 28 June 00Z namelist**
```
&time_control
run_days                  = 4,
run_hours                 = 0,
run_minutes               = 0,
run_seconds               = 0,
start_year                = 2012, 2012, 2012,
start_month               = 06,   06,   06,
start_day                 = 28,   28,   28,
start_hour                = 00,   00,   00,
start_minute              = 00,   00,   00,
start_second              = 00,   00,   00,
end_year                  = 2012, 2012, 2012,
end_month                 = 07,   07,   07,
end_day                   = 02,   02,   02,
end_hour                  = 00,   00,   00,
end_minute                = 00,   00,   00,
end_second                = 00,   00,   00,
interval_seconds          = 21600
input_from_file           = .true.,.true.,.true.,
history_interval          = 60, 10, 10,
frames_per_outfile        = 1,  1, 1,
history_outname           =
"/data/derecho/WRF_output/NE_2012/wrfout/wrfout_d<domain>_<date>"
restart                   = .false.,
restart_interval          = 1440,
override_restart_timers   = .true.,
io_form_history           = 11
io_form_restart           = 2
io_form_input             = 2
io_form_boundary          = 11
io_form_auxinput2         = 11
io_form_auxhist2          = 11
debug_level               = 10
nocolons                  = .true.,
auxinput4_inname          = "wrflowinp_d<domain>",
auxinput4_interval        = 1440, 1440, 1440,
io_form_auxinput4         = 2,
auxinput1_inname          = "/data/derecho/met_files/ERA5/met_em.d<domain>.<date>"
iofields_filename         = "my_file_d01.txt", "my_file_d02.txt", "my_file_d03.txt",
ignore_iofields_warning   = .true.,
auxhist1_outname          =
"/data/derecho/WRF_output/NE_2012/aux1/auxhist1_d<domain>_<date>"
auxhist1_interval         = 60, 60, 60,
frames_per_auxhist1       = 1, 1, 1,
io_form_auxhist1          = 11,
output_diagnostics        = 1,
```

Shepherd T.J., Letson F., Barthelmie R.J. and Pryor S.C. How well are hazards associated with derechos reproduced in regional climate simulations? *Natural Hazards and Earth System Sciences Discussions* *(*nhess-2021-373)

**Response to reviewer #1:**

```
 auxhist3_outname            =
"/data/derecho/WRF_output/NE_2012/wrfout/wrfxtrm_d<domain>_<date>"
 auxhist3_interval           = 60, 10, 10,
 frames_per_auxhist3          = 1, 1, 1,
 io_form_auxhist3            = 11,
/
 &domains
 time_step              = 30,
 time_step_fract_num          = 0,
 time_step_fract_den          = 1,
 max_dom               = 3,
 e_we                = 175,   262,   295,
 e_sn                = 175,   262,   295,
 e_vert               = 41,    41,    41,
 p_top_requested           = 5000,
 sfcp_to_sfcp            = .true.,
 num_metgrid_levels          = 38,
 num_metgrid_soil_levels        = 4,
 dx                 = 12000, 4000, 1333.33,
 dy                 = 12000, 4000, 1333.33,
 grid_id              = 1,    2,   3,
 parent_id             = 1,    1,   2,
 i_parent_start            = 1,    60,   105,
 j_parent_start            = 1,    35,   75,
 parent_grid_ratio           = 1,    3,   3,
 parent_time_step_ratio         = 1,    3,   3,
 feedback              = 0,
 max_ts_locs             = 0,
 eta_levels             = 1.0000 , 0.9958 , 0.9916 , 0.9874 , 0.9832 ,
                0.9790 , 0.9749 , 0.9707 , 0.9661 , 0.9609 ,
                0.9549 , 0.9480 , 0.9398 , 0.9303 , 0.9189 ,
                0.9054 , 0.8894 , 0.8704 , 0.8481 , 0.8221 ,
                0.7922 , 0.7583 , 0.7205 , 0.6791 , 0.6346 ,
                0.5877 , 0.5393 , 0.4900 , 0.4407 , 0.3922 ,
                0.3450 , 0.2996 , 0.2564 , 0.2156 , 0.1773 ,
                0.1417 , 0.1086 , 0.0755 , 0.0475 , 0.0224 ,
                0.0000,
/
 &physics
 mp_physics             = 9,   9,   9,
 ra_lw_physics            = 1,   1,   1,
 ra_sw_physics            = 1,   1,   1,
 radt                = 10,   10,   10,
 sf_sfclay_physics           = 1,   1,   1,
 sf_surface_physics           = 2,   2,   2,
 bl_pbl_physics            = 5,   5,   5,
 bldt                = 0,   0,   0,
```

Shepherd T.J., Letson F., Barthelmie R.J. and Pryor S.C. How well are hazards associated with derechos reproduced in regional climate simulations? *Natural Hazards and Earth System Sciences Discussions (*nhess-2021-373)

**Response to reviewer #1:**

```
cu_physics                    = 1,    0,    0,
cudt                  = 5,
isfflx                = 1,
ifsnow                 = 1,
icloud                 = 1,
surface_input_source           = 3,
num_soil_layers             = 4,
num_land_cat              = 21,
sf_urban_physics             = 0,    0,    0,
bl_mynn_tkebudget              = 1,    1,    1,
bl_mynn_tkeadvect               = .true., .true., .true.,
rdmaxalb               = .false.,
sst_update             = 1,
tmn_update              = 1,
usemonalb               = .true.,
lagday                = 150,
sst_skin              = 1,
slope_rad              = 1,  1,  1,
prec_acc_dt              = 60., 10., 10.,
fractional_seaice            = 1,
seaice_threshold             = 0.,
/
&noah_mp
dveg                  = 4,
opt_crs               = 1,
opt_btr               = 2,
opt_run                = 3,
opt_sfc               = 1,
opt_frz               = 1,
opt_inf               = 1,
opt_rad               = 3,
opt_alb               = 2,
opt_snf               = 4,
opt_tbot               = 1,
opt_stc               = 3,
/
&dynamics
w_damping                  = 1,
diff_opt               = 1,    1,     1,
km_opt                = 4,    4,    4,
diff_6th_opt              = 0,    0,     0,
diff_6th_factor             = 0.12,  0.12,  0.12,
base_temp              = 290.
damp_opt               = 0,
zdamp                 = 5000., 5000., 5000.,
dampcoef               = 0.01,   0.01, 0.01,
khdif                 = 0,    0,
```

Shepherd T.J., Letson F., Barthelmie R.J. and Pryor S.C. How well are hazards associated with derechos reproduced in regional climate simulations? *Natural Hazards and Earth System Sciences Discussions (*nhess-2021-373)

**Response to reviewer #1:**

```
kvdif                      = 0,     0,
non_hydrostatic              = .true., .true., .true.,
/
&bdy_control
spec_bdy_width                = 5,
spec_zone               = 1,
relax_zone              = 4,
spec_exp               = 0.13
specified              = .true., .false., .false.,
nested                 = .false., .true., .true.,
/
&grib2
/
&namelist_quilt
nio_tasks_per_group = 0,
nio_groups = 1,
/
```

Shepherd T.J., Letson F., Barthelmie R.J. and Pryor S.C. How well are hazards associated with derechos reproduced in regional climate simulations? *Natural Hazards and Earth System Sciences Discussions (*nhess-2021-373)

**Response to reviewer #1:**

**NSSL namelist**

```
&time_control
run_days                = 6,
run_hours               = 0,
run_minutes             = 0,
run_seconds             = 0,
start_year              = 2012, 2012, 2012,
start_month             = 06,   06,   06,
start_day               = 26,   26,   26,
start_hour              = 00,   00,   00,
start_minute            = 00,   00,   00,
start_second            = 00,   00,   00,
end_year                = 2012, 2012, 2012,
end_month               = 07,   07,   07,
end_day                 = 02,   02,   02,
end_hour                = 00,   00,   00,
end_minute              = 00,   00,   00,
end_second              = 00,   00,   00,
interval_seconds        = 21600
input_from_file         = .true.,.true.,.true.,
history_interval        = 60, 10, 10,
frames_per_outfile      = 1,  1, 1,
history_outname         =
"/data/derecho/WRF_output/NE_2012/wrfout/wrfout_d<domain>_<date>"
restart                 = .false.,
restart_interval        = 1440,
override_restart_timers = .true.,
io_form_history         = 11
io_form_restart         = 2
io_form_input           = 2
io_form_boundary        = 11
io_form_auxinput2       = 11
io_form_auxhist2        = 11
debug_level             = 10
nocolons                = .true.,
auxinput4_inname        = "wrflowinp_d<domain>",
auxinput4_interval      = 1440, 1440, 1440,
io_form_auxinput4       = 2,
auxinput1_inname        = "/data/derecho/met_files/ERA5/met_em.d<domain>.<date>"
iofields_filename       = "my_file_d01.txt", "my_file_d02.txt", "my_file_d03.txt",
ignore_iofields_warning = .true.,
auxhist1_outname        =
"/data/derecho/WRF_output/NE_2012/aux1/auxhist1_d<domain>_<date>"
auxhist1_interval       = 60, 60, 60,
frames_per_auxhist1     = 1, 1, 1,
io_form_auxhist1        = 11,
```

Shepherd T.J., Letson F., Barthelmie R.J. and Pryor S.C. How well are hazards associated with derechos reproduced in regional climate simulations? *Natural Hazards and Earth System Sciences Discussions* (nhess-2021-373)

**Response to reviewer #1:**

```
 output_diagnostics         =  1,
 auxhist3_outname           =
"/data/derecho/WRF_output/NE_2012/wrfout/wrfxtrm_d<domain>_<date>"
 auxhist3_interval          = 60, 10, 10,
 frames_per_auxhist3        = 1, 1, 1,
 io_form_auxhist3           = 11,
 /
 &domains
 time_step                  = 30,
 time_step_fract_num        = 0,
 time_step_fract_den        = 1,
 max_dom                    = 3,
 e_we                       = 175,   262,   295,
 e_sn                       = 175,   262,   295,
 e_vert                     = 41,    41,    41,
 p_top_requested            = 5000,
 sfcp_to_sfcp               = .true.,
 num_metgrid_levels         = 38,
 num_metgrid_soil_levels    = 4,
 dx                         = 12000, 4000, 1333.33,
 dy                         = 12000, 4000, 1333.33,
 grid_id                    = 1,    2,    3,
 parent_id                  = 1,    1,    2,
 i_parent_start             = 1,    60,   105,
 j_parent_start             = 1,    35,   75,
 parent_grid_ratio          = 1,    3,    3,
 parent_time_step_ratio     = 1,    3,    3,
 feedback                   = 0,
 max_ts_locs                = 0,
 eta_levels                 = 1.0000 , 0.9958 , 0.9916 , 0.9874 , 0.9832 ,
                             0.9790 , 0.9749 , 0.9707 , 0.9661 , 0.9609 ,
                             0.9549 , 0.9480 , 0.9398 , 0.9303 , 0.9189 ,
                             0.9054 , 0.8894 , 0.8704 , 0.8481 , 0.8221 ,
                             0.7922 , 0.7583 , 0.7205 , 0.6791 , 0.6346 ,
                             0.5877 , 0.5393 , 0.4900 , 0.4407 , 0.3922 ,
                             0.3450 , 0.2996 , 0.2564 , 0.2156 , 0.1773 ,
                             0.1417 , 0.1086 , 0.0755 , 0.0475 , 0.0224 ,
                             0.0000,
 /
 &physics
 mp_physics                 = 17,   17,   17,
 ra_lw_physics              = 1,    1,    1,
 ra_sw_physics              = 1,    1,    1,
 radt                       = 10,   10,   10,
 sf_sfclay_physics          = 1,    1,    1,
 sf_surface_physics         = 2,    2,    2,
 bl_pbl_physics             = 5,    5,    5,
```

Shepherd T.J., Letson F., Barthelmie R.J. and Pryor S.C. How well are hazards associated with derechos reproduced in regional climate simulations? *Natural Hazards and Earth System Sciences Discussions (*nhess-2021-373)

**Response to reviewer #1:**

```
bldt                    = 0,    0,    0,
cu_physics               = 1,    0,    0,
cudt                    = 5,
isfflx                  = 1,
ifsnow                   = 1,
icloud                   = 1,
surface_input_source        = 3,
num_soil_layers            = 4,
num_land_cat              = 21,
sf_urban_physics           = 0,    0,    0,
bl_mynn_tkebudget          = 1,    1,    1,
bl_mynn_tkeadvect           = .true., .true., .true.,
rdmaxalb                 = .false.,
sst_update               = 1,
tmn_update                = 1,
usemonalb                 = .true.,
lagday                  = 150,
sst_skin                 = 1,
slope_rad                = 1,  1,  1,
prec_acc_dt               = 60., 10., 10.,
fractional_seaice           = 1,
seaice_threshold            = 0.,
/
&noah_mp
dveg                    = 4,
opt_crs                  = 1,
opt_btr                  = 2,
opt_run                   = 3,
opt_sfc                  = 1,
opt_frz                  = 1,
opt_inf                  = 1,
opt_rad                  = 3,
opt_alb                  = 2,
opt_snf                  = 4,
opt_tbot                 = 1,
opt_stc                  = 3,
/
&dynamics
w_damping                 = 1,
diff_opt                 = 1,    1,    1,
km_opt                   = 4,    4,    4,
diff_6th_opt              = 0,    0,    0,
diff_6th_factor            = 0.12,  0.12,  0.12,
base_temp                = 290.
damp_opt                 = 0,
zdamp                   = 5000., 5000., 5000.,
dampcoef                 = 0.01,   0.01, 0.01,
```

Shepherd T.J., Letson F., Barthelmie R.J. and Pryor S.C. How well are hazards associated with derechos reproduced in regional climate simulations? *Natural Hazards and Earth System Sciences Discussions (*nhess-2021-373)

**Response to reviewer #1:**

```
khdif                    = 0,    0,
kvdif                    = 0,    0,
non_hydrostatic             = .true., .true., .true.,
/
&bdy_control
spec_bdy_width              = 5,
spec_zone               = 1,
relax_zone              = 4,
spec_exp                = 0.13
specified                  = .true., .false., .false.,
nested                  = .false., .true., .true.,
/
&grib2
/
&namelist_quilt
nio_tasks_per_group = 0,
nio_groups = 1,
/
```

Shepherd T.J., Letson F., Barthelmie R.J. and Pryor S.C. How well are hazards associated with derechos reproduced in regional climate simulations? *Natural Hazards and Earth System Sciences Discussions (*nhess-2021-373)

**Response to reviewer #1:**

**Thompson namelist**
```
&time_control
run_days                  = 6,
run_hours                 = 0,
run_minutes               = 0,
run_seconds               = 0,
start_year                = 2012,  2012, 2012,
start_month               = 06,   06,   06,
start_day                 = 26,   26,   26,
start_hour                = 00,   00,   00,
start_minute              = 00,   00,   00,
start_second              = 00,   00,   00,
end_year                  = 2012, 2012, 2012,
end_month                 = 07,   07,   07,
end_day                   = 02,   02,   02,
end_hour                  = 00,   00,   00,
end_minute                = 00,   00,   00,
end_second                = 00,   00,   00,
interval_seconds          = 21600
input_from_file           = .true.,.true.,.true.,
history_interval          = 60, 10, 10,
frames_per_outfile        = 1,  1, 1,
history_outname           =
"/data/derecho/WRF_output/NE_2012/wrfout/wrfout_d<domain>_<date>"
restart                   = .false.,
restart_interval          = 1440,
override_restart_timers   = .true.,
io_form_history           = 11
io_form_restart           = 2
io_form_input             = 2
io_form_boundary          = 11
io_form_auxinput2         = 11
io_form_auxhist2          = 11
debug_level               = 10
nocolons                  = .true.,
auxinput4_inname          = "wrflowinp_d<domain>",
auxinput4_interval        = 1440, 1440, 1440,
io_form_auxinput4         = 2,
auxinput1_inname          = "/data/derecho/met_files/ERA5/met_em.d<domain>.<date>"
iofields_filename         = "my_file_d01.txt", "my_file_d02.txt", "my_file_d03.txt",
ignore_iofields_warning   = .true.,
auxhist1_outname          =
"/data/derecho/WRF_output/NE_2012/aux1/auxhist1_d<domain>_<date>"
auxhist1_interval         = 60, 60, 60,
frames_per_auxhist1       = 1,  1, 1,
io_form_auxhist1          = 11,
output_diagnostics        = 1,
```

Shepherd T.J., Letson F., Barthelmie R.J. and Pryor S.C. How well are hazards associated with derechos reproduced in regional climate simulations? *Natural Hazards and Earth System Sciences Discussions (*nhess-2021-373)

**Response to reviewer #1:**

```
 auxhist3_outname              =
"/data/derecho/WRF_output/NE_2012/wrfout/wrfxtrm_d<domain>_<date>"
 auxhist3_interval           = 60, 10, 10,
 frames_per_auxhist3          = 1, 1, 1,
 io_form_auxhist3             = 11,
/
 &domains
 time_step               = 30,
 time_step_fract_num          = 0,
 time_step_fract_den          = 1,
 max_dom                 = 3,
 e_we                 = 175,   262,   295,
 e_sn                 = 175,   262,   295,
 e_vert               = 41,    41,    41,
 p_top_requested            = 5000,
 sfcp_to_sfcp             = .true.,
 num_metgrid_levels           = 38,
 num_metgrid_soil_levels        = 4,
 dx                 = 12000, 4000, 1333.33,
 dy                 = 12000, 4000, 1333.33,
 grid_id               = 1,    2,    3,
 parent_id              = 1,    1,    2,
 i_parent_start            = 1,    60,   105,
 j_parent_start            = 1,    35,   75,
 parent_grid_ratio           = 1,    3,    3,
 parent_time_step_ratio         = 1,    3,    3,
 feedback              = 0,
 max_ts_locs            = 0,
 eta_levels              = 1.0000 , 0.9958 , 0.9916 , 0.9874 , 0.9832 ,
                   0.9790 , 0.9749 , 0.9707 , 0.9661 , 0.9609 ,
                   0.9549 , 0.9480 , 0.9398 , 0.9303 , 0.9189 ,
                   0.9054 , 0.8894 , 0.8704 , 0.8481 , 0.8221 ,
                   0.7922 , 0.7583 , 0.7205 , 0.6791 , 0.6346 ,
                   0.5877 , 0.5393 , 0.4900 , 0.4407 , 0.3922 ,
                   0.3450 , 0.2996 , 0.2564 , 0.2156 , 0.1773 ,
                   0.1417 , 0.1086 , 0.0755 , 0.0475 , 0.0224 ,
                   0.0000,
/
 &physics
 mp_physics             = 8,    8,    8,
 ra_lw_physics            = 1,    1,    1,
 ra_sw_physics            = 1,    1,    1,
 radt               = 10,    10,    10,
 sf_sfclay_physics           = 1,    1,    1,
 sf_surface_physics           = 2,    2,    2,
 bl_pbl_physics            = 5,    5,    5,
 bldt               = 0,    0,    0,
```

Shepherd T.J., Letson F., Barthelmie R.J. and Pryor S.C. How well are hazards associated with derechos reproduced in regional climate simulations? *Natural Hazards and Earth System Sciences Discussions (*nhess-2021-373)

**Response to reviewer #1:**

```
cu_physics                  = 1,    0,    0,
cudt                     = 5,
isfflx                   = 1,
ifsnow                    = 1,
icloud                    = 1,
surface_input_source        = 3,
num_soil_layers             = 4,
num_land_cat               = 21,
sf_urban_physics             = 0,    0,    0,
bl_mynn_tkebudget             = 1,    1,    1,
bl_mynn_tkeadvect              = .true., .true., .true.,
rdmaxalb                  = .false.,
sst_update                = 1,
tmn_update                 = 1,
usemonalb                  = .true.,
lagday                   = 150,
sst_skin                  = 1,
slope_rad                  = 1,  1,  1,
prec_acc_dt                = 60., 10., 10.,
do_radar_ref               = 1,
fractional_seaice            = 1,
seaice_threshold             = 0.,
/
&noah_mp
dveg                    = 4,
opt_crs                   = 1,
opt_btr                   = 2,
opt_run                   = 3,
opt_sfc                   = 1,
opt_frz                   = 1,
opt_inf                   = 1,
opt_rad                   = 3,
opt_alb                   = 2,
opt_snf                   = 4,
opt_tbot                  = 1,
opt_stc                   = 3,
/
&dynamics
w_damping                   = 1,
diff_opt                 = 1,    1,    1,
km_opt                   = 4,    4,    4,
diff_6th_opt                = 0,    0,    0,
diff_6th_factor              = 0.12,  0.12,  0.12,
base_temp                  = 290.
damp_opt                  = 0,
zdamp                    = 5000., 5000., 5000.,
dampcoef                   = 0.01,   0.01, 0.01,
```

Shepherd T.J., Letson F., Barthelmie R.J. and Pryor S.C. How well are hazards associated with derechos reproduced in regional climate simulations? *Natural Hazards and Earth System Sciences Discussions (*nhess-2021-373)

**Response to reviewer #1:**

```
khdif                       = 0,    0,
kvdif                       = 0,    0,
non_hydrostatic                 = .true., .true., .true.,
/
&bdy_control
spec_bdy_width                  = 5,
spec_zone                   = 1,
relax_zone                  = 4,
spec_exp                    = 0.13
specified                   = .true., .false., .false.,
nested                      = .false., .true., .true.,
/
&grib2
/
&namelist_quilt
nio_tasks_per_group = 0,
nio_groups = 1,
/
```

Shepherd T.J., Letson F., Barthelmie R.J. and Pryor S.C. How well are hazards associated with derechos reproduced in regional climate simulations? *Natural Hazards and Earth System Sciences Discussions* (nhess-2021-373)

**Response to reviewer #1:**

**Goddard namelist**
```
&time_control
 run_days                 = 6,
 run_hours                = 0,
 run_minutes              = 0,
 run_seconds              = 0,
 start_year               = 2012, 2012, 2012,
 start_month              = 06,  06,  06,
 start_day                = 26,  26,  26,
 start_hour               = 00,  00,  00,
 start_minute             = 00,  00,  00,
 start_second             = 00,  00,  00,
 end_year                 = 2012, 2012, 2012,
 end_month                = 07,  07,  07,
 end_day                  = 02,  02,  02,
 end_hour                 = 00,  00,  00,
 end_minute               = 00,  00,  00,
 end_second               = 00,  00,  00,
 interval_seconds         = 21600
 input_from_file          = .true.,.true.,.true.,
 history_interval         = 60, 10, 10,
 frames_per_outfile       = 1,  1, 1,
 history_outname          =
"/data/derecho/WRF_output/NE_2012/wrfout/wrfout_d<domain>_<date>"
 restart                  = .false.,
 restart_interval         = 1440,
 override_restart_timers   = .true.,
 io_form_history          = 11
 io_form_restart          = 2
 io_form_input            = 2
 io_form_boundary         = 11
 io_form_auxinput2        = 11
 io_form_auxhist2         = 11
 debug_level              = 10
 nocolons                 = .true.,
 auxinput4_inname         = "wrflowinp_d<domain>",
 auxinput4_interval       = 1440, 1440, 1440,
 io_form_auxinput4        = 2,
 auxinput1_inname         = "/data/derecho/met_files/ERA5/met_em.d<domain>.<date>"
 iofields_filename        = "my_file_d01.txt", "my_file_d02.txt", "my_file_d03.txt",
 ignore_iofields_warning   = .true.,
 auxhist1_outname         =
"/data/derecho/WRF_output/NE_2012/aux1/auxhist1_d<domain>_<date>"
 auxhist1_interval        = 60, 60, 60,
 frames_per_auxhist1      = 1,  1, 1,
 io_form_auxhist1         = 11,
 output_diagnostics       = 1,
```

Shepherd T.J., Letson F., Barthelmie R.J. and Pryor S.C. How well are hazards associated with derechos reproduced in regional climate simulations? *Natural Hazards and Earth System Sciences Discussions (*nhess-2021-373)

**Response to reviewer #1:**

```
 auxhist3_outname              =
"/data/derecho/WRF_output/NE_2012/wrfout/wrfxtrm_d<domain>_<date>"
 auxhist3_interval          = 60, 10, 10,
 frames_per_auxhist3          = 1, 1, 1,
 io_form_auxhist3          = 11,
/
 &domains
 time_step             = 30,
 time_step_fract_num         = 0,
 time_step_fract_den         = 1,
 max_dom            = 3,
 e_we            = 175,   262,   295,
 e_sn            = 175,   262,   295,
 e_vert           = 41,    41,    41,
 p_top_requested          = 5000,
 sfcp_to_sfcp           = .true.,
 num_metgrid_levels          = 38,
 num_metgrid_soil_levels         = 4,
 dx             = 12000, 4000, 1333.33,
 dy             = 12000, 4000, 1333.33,
 grid_id           = 1,    2,   3,
 parent_id           = 1,    1,   2,
 i_parent_start          = 1,    60,   105,
 j_parent_start          = 1,    35,   75,
 parent_grid_ratio          = 1,    3,   3,
 parent_time_step_ratio         = 1,    3,   3,
 feedback           = 0,
 max_ts_locs           = 0,
 eta_levels            = 1.0000 , 0.9958 , 0.9916 , 0.9874 , 0.9832 ,
                 0.9790 , 0.9749 , 0.9707 , 0.9661 , 0.9609 ,
                 0.9549 , 0.9480 , 0.9398 , 0.9303 , 0.9189 ,
                 0.9054 , 0.8894 , 0.8704 , 0.8481 , 0.8221 ,
                 0.7922 , 0.7583 , 0.7205 , 0.6791 , 0.6346 ,
                 0.5877 , 0.5393 , 0.4900 , 0.4407 , 0.3922 ,
                 0.3450 , 0.2996 , 0.2564 , 0.2156 , 0.1773 ,
                 0.1417 , 0.1086 , 0.0755 , 0.0475 , 0.0224 ,
                 0.0000,
/
 &physics
 mp_physics            = 7,    7,   7,
 gsfcgce_hail           = 1,
 ra_lw_physics           = 1,    1,   1,
 ra_sw_physics           = 1,    1,   1,
 radt            = 10,   10,   10,
 sf_sfclay_physics          = 1,    1,   1,
 sf_surface_physics          = 2,    2,   2,
 bl_pbl_physics           = 5,    5,   5,
```

Shepherd T.J., Letson F., Barthelmie R.J. and Pryor S.C. How well are hazards associated with derechos reproduced in regional climate simulations? *Natural Hazards and Earth System Sciences Discussions (*nhess-2021-373)

**Response to reviewer #1:**

```
bldt                    = 0,    0,    0,
cu_physics               = 1,    0,    0,
cudt                    = 5,
isfflx                  = 1,
ifsnow                   = 1,
icloud                   = 1,
surface_input_source        = 3,
num_soil_layers            = 4,
num_land_cat              = 21,
sf_urban_physics           = 0,    0,    0,
bl_mynn_tkebudget           = 1,    1,    1,
bl_mynn_tkeadvect            = .true., .true., .true.,
rdmaxalb                 = .false.,
sst_update               = 1,
tmn_update                = 1,
usemonalb                 = .true.,
lagday                  = 150,
sst_skin                 = 1,
slope_rad                = 1,  1,  1,
prec_acc_dt               = 60., 10., 10.,
do_radar_ref               = 1,
fractional_seaice           = 1,
seaice_threshold            = 0.,
/
&noah_mp
dveg                    = 4,
opt_crs                  = 1,
opt_btr                  = 2,
opt_run                   = 3,
opt_sfc                  = 1,
opt_frz                  = 1,
opt_inf                  = 1,
opt_rad                   = 3,
opt_alb                   = 2,
opt_snf                   = 4,
opt_tbot                  = 1,
opt_stc                  = 3,
/
&dynamics
w_damping                 = 1,
diff_opt                 = 1,    1,    1,
km_opt                   = 4,    4,    4,
diff_6th_opt               = 0,    0,    0,
diff_6th_factor             = 0.12,  0.12,  0.12,
base_temp                 = 290.
damp_opt                  = 0,
zdamp                   = 5000., 5000., 5000.,
```

Shepherd T.J., Letson F., Barthelmie R.J. and Pryor S.C. How well are hazards associated with derechos reproduced in regional climate simulations? *Natural Hazards and Earth System Sciences Discussions (*nhess-2021-373)

**Response to reviewer #1:**

```
dampcoef                   = 0.01,   0.01,  0.01,
khdif                   = 0,     0,
kvdif                   = 0,     0,
non_hydrostatic            = .true., .true., .true.,
/
&bdy_control
spec_bdy_width             = 5,
spec_zone               = 1,
relax_zone               = 4,
spec_exp                = 0.13
specified               = .true., .false., .false.,
nested                 = .false., .true., .true.,
/
&grib2
/
&namelist_quilt
nio_tasks_per_group = 0,
nio_groups = 1,
/
```

Shepherd T.J., Letson F., Barthelmie R.J. and Pryor S.C. How well are hazards associated with derechos reproduced in regional climate simulations? *Natural Hazards and Earth System Sciences Discussions (*nhess-2021-373)

**Response to reviewer #1:**

**Milbrandt 26 June 00Z ERA-I namelist**

```
&time_control
run_days                 = 6,
run_hours                = 0,
run_minutes              = 0,
run_seconds              = 0,
start_year               = 2012, 2012, 2012,
start_month              = 06,   06,   06,
start_day                = 26,   26,   26,
start_hour               = 00,   00,   00,
start_minute             = 00,   00,   00,
start_second             = 00,   00,   00,
end_year                 = 2012, 2012, 2012,
end_month                = 07,   07,   07,
end_day                  = 02,   02,   02,
end_hour                 = 00,   00,   00,
end_minute               = 00,   00,   00,
end_second               = 00,   00,   00,
interval_seconds         = 21600
input_from_file          = .true.,.true.,.true.,
history_interval         = 60, 10, 10,
frames_per_outfile       = 1, 1, 1,
history_outname          =
"/data/derecho/WRF_output/NE_2012/wrfout/wrfout_d<domain>_<date>"
restart                  = .false.,
restart_interval         = 1440,
override_restart_timers  = .true.,
io_form_history          = 11
io_form_restart          = 2
io_form_input            = 2
io_form_boundary         = 11
io_form_auxinput2        = 11
io_form_auxhist2         = 11
debug_level              = 10
nocolons                 = .true.,
auxinput4_inname         = "wrflowinp_d<domain>",
auxinput4_interval       = 1440, 1440, 1440,
io_form_auxinput4        = 2,
auxinput1_inname         = "/data/derecho/met_files/ERA-I/met_em.d<domain>.<date>"
iofields_filename        = "my_file_d01.txt", "my_file_d02.txt", "my_file_d03.txt",
ignore_iofields_warning  = .true.,
auxhist1_outname         =
"/data/derecho/WRF_output/NE_2012/aux1/auxhist1_d<domain>_<date>"
auxhist1_interval        = 60, 60, 60,
frames_per_auxhist1      = 1, 1, 1,
io_form_auxhist1         = 11,
```

Shepherd T.J., Letson F., Barthelmie R.J. and Pryor S.C. How well are hazards associated with derechos reproduced in regional climate simulations? *Natural Hazards and Earth System Sciences Discussions (*nhess-2021-373)

**Response to reviewer #1:**

```
 output_diagnostics            =  1,
 auxhist3_outname              =
"/data/derecho/WRF_output/NE_2012/wrfout/wrfxtrm_d<domain>_<date>"
 auxhist3_interval             = 60, 10, 10,
 frames_per_auxhist3           = 1, 1, 1,
 io_form_auxhist3              = 11,
/
 &domains
 time_step                     = 30,
 time_step_fract_num           = 0,
 time_step_fract_den           = 1,
 max_dom                       = 3,
 e_we                          = 175,   262,   295,
 e_sn                          = 175,   262,   295,
 e_vert                        = 41,    41,   41,
 p_top_requested               = 5000,
 sfcp_to_sfcp                  = .true.,
 num_metgrid_levels            = 61,
 num_metgrid_soil_levels       = 4,
 dx                            = 12000, 4000, 1333.33,
 dy                            = 12000, 4000, 1333.33,
 grid_id                       = 1,    2,   3,
 parent_id                     = 1,    1,   2,
 i_parent_start                = 1,    60,   105,
 j_parent_start                = 1,    35,   75,
 parent_grid_ratio             = 1,    3,   3,
 parent_time_step_ratio        = 1,    3,   3,
 feedback                      = 0,
 max_ts_locs                   = 0,
 eta_levels                    = 1.0000 , 0.9958 , 0.9916 , 0.9874 , 0.9832 ,
                                 0.9790 , 0.9749 , 0.9707 , 0.9661 , 0.9609 ,
                                 0.9549 , 0.9480 , 0.9398 , 0.9303 , 0.9189 ,
                                 0.9054 , 0.8894 , 0.8704 , 0.8481 , 0.8221 ,
                                 0.7922 , 0.7583 , 0.7205 , 0.6791 , 0.6346 ,
                                 0.5877 , 0.5393 , 0.4900 , 0.4407 , 0.3922 ,
                                 0.3450 , 0.2996 , 0.2564 , 0.2156 , 0.1773 ,
                                 0.1417 , 0.1086 , 0.0755 , 0.0475 , 0.0224 ,
                                 0.0000,
/
 &physics
 mp_physics                    = 9,    9,   9,
 ra_lw_physics                 = 1,    1,   1,
 ra_sw_physics                 = 1,    1,   1,
 radt                          = 10,   10,   10,
 sf_sfclay_physics             = 1,    1,   1,
 sf_surface_physics            = 2,    2,   2,
 bl_pbl_physics                = 5,    5,   5,
```

Shepherd T.J., Letson F., Barthelmie R.J. and Pryor S.C. How well are hazards associated with derechos reproduced in regional climate simulations? *Natural Hazards and Earth System Sciences Discussions (*nhess-2021-373)

**Response to reviewer #1:**

```
bldt                    = 0,   0,   0,
cu_physics               = 1,   0,   0,
cudt                    = 5,
isfflx                  = 1,
ifsnow                   = 1,
icloud                   = 1,
surface_input_source        = 3,
num_soil_layers             = 4,
num_land_cat               = 21,
sf_urban_physics            = 0,   0,   0,
bl_mynn_tkebudget            = 1,   1,   1,
bl_mynn_tkeadvect             = .true., .true., .true.,
rdmaxalb                 = .false.,
sst_update              = 1,
tmn_update               = 1,
usemonalb                = .true.,
lagday                  = 150,
sst_skin                = 1,
slope_rad                = 1,  1,  1,
prec_acc_dt               = 60., 10., 10.,
fractional_seaice           = 1,
seaice_threshold             = 0.,
/
&noah_mp
dveg                    = 4,
opt_crs                 = 1,
opt_btr                 = 2,
opt_run                  = 3,
opt_sfc                 = 1,
opt_frz                 = 1,
opt_inf                 = 1,
opt_rad                 = 3,
opt_alb                 = 2,
opt_snf                 = 4,
opt_tbot                = 1,
opt_stc                 = 3,
/
&dynamics
w_damping                 = 1,
diff_opt                = 1,    1,    1,
km_opt                  = 4,    4,    4,
diff_6th_opt              = 0,    0,    0,
diff_6th_factor            = 0.12,  0.12,  0.12,
base_temp                = 290.
damp_opt                 = 0,
zdamp                   = 5000., 5000., 5000.,
dampcoef                  = 0.01,   0.01, 0.01,
```

Shepherd T.J., Letson F., Barthelmie R.J. and Pryor S.C. How well are hazards associated with derechos reproduced in regional climate simulations? *Natural Hazards and Earth System Sciences Discussions (*nhess-2021-373)

**Response to reviewer #1:**

```
khdif                        = 0,    0,
kvdif                        = 0,    0,
non_hydrostatic                  = .true., .true., .true.,
/
&bdy_control
spec_bdy_width                   = 5,
spec_zone                    = 1,
relax_zone                   = 4,
spec_exp                     = 0.13
specified                        = .true., .false., .false.,
nested                       = .false., .true., .true.,
/
&grib2
/
&namelist_quilt
nio_tasks_per_group = 0,
nio_groups = 1,
/
```

Shepherd T.J., Letson F., Barthelmie R.J. and Pryor S.C. How well are hazards associated with derechos reproduced in regional climate simulations? *Natural Hazards and Earth System Sciences Discussions (*nhess-2021-373)

**Response to reviewer #1:**

**Milbrant 28 June 00Z ERA-I namelist**

```
&time_control
run_days                 = 4,
run_hours                = 0,
run_minutes              = 0,
run_seconds              = 0,
start_year               = 2012, 2012, 2012,
start_month              = 06,   06,   06,
start_day                = 28,   28,   28,
start_hour               = 00,   00,   00,
start_minute             = 00,   00,   00,
start_second             = 00,   00,   00,
end_year                 = 2012, 2012, 2012,
end_month                = 07,   07,   07,
end_day                  = 02,   02,   02,
end_hour                 = 00,   00,   00,
end_minute               = 00,   00,   00,
end_second               = 00,   00,   00,
interval_seconds         = 21600
input_from_file          = .true.,.true.,.true.,
history_interval         = 60, 10, 10,
frames_per_outfile       = 1,  1, 1,
history_outname          =
"/data/derecho/WRF_output/NE_2012/wrfout/wrfout_d<domain>_<date>"
restart                  = .false.,
restart_interval         = 1440,
override_restart_timers  = .true.,
io_form_history          = 11
io_form_restart          = 2
io_form_input            = 2
io_form_boundary         = 11
io_form_auxinput2        = 11
io_form_auxhist2         = 11
debug_level              = 10
nocolons                 = .true.,
auxinput4_inname         = "wrflowinp_d<domain>",
auxinput4_interval       = 1440, 1440, 1440,
io_form_auxinput4        = 2,
auxinput1_inname         = "/data/derecho/met_files/ERA-I/met_em.d<domain>.<date>"
iofields_filename        = "my_file_d01.txt", "my_file_d02.txt", "my_file_d03.txt",
ignore_iofields_warning  = .true.,
auxhist1_outname         =
"/data/derecho/WRF_output/NE_2012/aux1/auxhist1_d<domain>_<date>"
auxhist1_interval        = 60, 60, 60,
frames_per_auxhist1      = 1, 1, 1,
io_form_auxhist1         = 11,
output_diagnostics       = 1,
```

Shepherd T.J., Letson F., Barthelmie R.J. and Pryor S.C. How well are hazards associated with derechos reproduced in regional climate simulations? *Natural Hazards and Earth System Sciences Discussions (*nhess-2021-373)

**Response to reviewer #1:**

```
 auxhist3_outname              =
"/data/derecho/WRF_output/NE_2012/wrfout/wrfxtrm_d<domain>_<date>"
 auxhist3_interval             = 60, 10, 10,
 frames_per_auxhist3           = 1, 1, 1,
 io_form_auxhist3              = 11,
/
 &domains
 time_step                     = 30,
 time_step_fract_num           = 0,
 time_step_fract_den           = 1,
 max_dom                       = 3,
 e_we                          = 175,   262,   295,
 e_sn                          = 175,   262,   295,
 e_vert                        = 41,    41,    41,
 p_top_requested               = 5000,
 sfcp_to_sfcp                  = .true.,
 num_metgrid_levels            = 61,
 num_metgrid_soil_levels       = 4,
 dx                            = 12000, 4000, 1333.33,
 dy                            = 12000, 4000, 1333.33,
 grid_id                       = 1,    2,    3,
 parent_id                     = 1,    1,    2,
 i_parent_start                = 1,    60,   105,
 j_parent_start                = 1,    35,   75,
 parent_grid_ratio             = 1,    3,    3,
 parent_time_step_ratio        = 1,    3,    3,
 feedback                      = 0,
 max_ts_locs                   = 0,
 eta_levels                    = 1.0000 , 0.9958 , 0.9916 , 0.9874 , 0.9832 ,
                                 0.9790 , 0.9749 , 0.9707 , 0.9661 , 0.9609 ,
                                 0.9549 , 0.9480 , 0.9398 , 0.9303 , 0.9189 ,
                                 0.9054 , 0.8894 , 0.8704 , 0.8481 , 0.8221 ,
                                 0.7922 , 0.7583 , 0.7205 , 0.6791 , 0.6346 ,
                                 0.5877 , 0.5393 , 0.4900 , 0.4407 , 0.3922 ,
                                 0.3450 , 0.2996 , 0.2564 , 0.2156 , 0.1773 ,
                                 0.1417 , 0.1086 , 0.0755 , 0.0475 , 0.0224 ,
                                 0.0000,
/
 &physics
 mp_physics                    = 9,    9,    9,
 ra_lw_physics                 = 1,    1,    1,
 ra_sw_physics                 = 1,    1,    1,
 radt                          = 10,   10,   10,
 sf_sfclay_physics             = 1,    1,    1,
 sf_surface_physics            = 2,    2,    2,
 bl_pbl_physics                = 5,    5,    5,
 bldt                          = 0,    0,    0,
```

Shepherd T.J., Letson F., Barthelmie R.J. and Pryor S.C. How well are hazards associated with derechos reproduced in regional climate simulations? *Natural Hazards and Earth System Sciences Discussions (*nhess-2021-373)

**Response to reviewer #1:**

```
cu_physics                = 1,    0,    0,
cudt                    = 5,
isfflx                  = 1,
ifsnow                   = 1,
icloud                   = 1,
surface_input_source       = 3,
num_soil_layers            = 4,
num_land_cat             = 21,
sf_urban_physics            = 0,    0,    0,
bl_mynn_tkebudget           = 1,    1,    1,
bl_mynn_tkeadvect           = .true., .true., .true.,
rdmaxalb                 = .false.,
sst_update               = 1,
tmn_update                = 1,
usemonalb                = .true.,
lagday                  = 150,
sst_skin                = 1,
slope_rad                = 1,  1,  1,
prec_acc_dt               = 60., 10., 10.,
fractional_seaice           = 1,
seaice_threshold            = 0.,
/
&noah_mp
dveg                    = 4,
opt_crs                  = 1,
opt_btr                  = 2,
opt_run                   = 3,
opt_sfc                  = 1,
opt_frz                  = 1,
opt_inf                  = 1,
opt_rad                  = 3,
opt_alb                  = 2,
opt_snf                  = 4,
opt_tbot                 = 1,
opt_stc                  = 3,
/
&dynamics
w_damping                  = 1,
diff_opt                 = 1,    1,    1,
km_opt                  = 4,    4,    4,
diff_6th_opt               = 0,    0,    0,
diff_6th_factor             = 0.12,  0.12,  0.12,
base_temp                = 290.
damp_opt                 = 0,
zdamp                   = 5000., 5000., 5000.,
dampcoef                 = 0.01,   0.01, 0.01,
khdif                   = 0,    0,
```

Shepherd T.J., Letson F., Barthelmie R.J. and Pryor S.C. How well are hazards associated with derechos reproduced in regional climate simulations? *Natural Hazards and Earth System Sciences Discussions (*nhess-2021-373)

**Response to reviewer #1:**

```
kvdif                     = 0,     0,
non_hydrostatic            = .true., .true., .true.,
/
&bdy_control
spec_bdy_width              = 5,
spec_zone               = 1,
relax_zone              = 4,
spec_exp                = 0.13
specified               = .true., .false., .false.,
nested                  = .false., .true., .true.,
/
&grib2
/
&namelist_quilt
nio_tasks_per_group = 0,
nio_groups = 1,
/
```

Shepherd T.J., Letson F., Barthelmie R.J. and Pryor S.C. How well are hazards associated with derechos reproduced in regional climate simulations? *Natural Hazards and Earth System Sciences Discussions* (nhess-2021-373)

**Response to reviewer #1:**

**ERA5 nudged namelist**

```
&time_control
run_days                 = 6,
run_hours                = 0,
run_minutes              = 0,
run_seconds              = 0,
start_year               = 2012,  2012, 2012,
start_month              = 06,   06,   06,
start_day                = 26,   26,   26,
start_hour               = 00,   00,   00,
start_minute             = 00,   00,   00,
start_second             = 00,   00,   00,
end_year                 = 2012, 2012, 2012,
end_month                = 07,   07,   07,
end_day                  = 02,   02,   02,
end_hour                 = 00,   00,   00,
end_minute               = 00,   00,   00,
end_second               = 00,   00,   00,
interval_seconds         = 21600
input_from_file          = .true.,.true.,.true.,
history_interval         = 60, 10, 10,
frames_per_outfile       = 1,  1, 1,
history_outname          =
"/data/derecho/WRF_output/NE_2012/wrfout/wrfout_d<domain>_<date>"
restart                  = .false.,
restart_interval         = 1440,
override_restart_timers  = .true.,
io_form_history          = 11
io_form_restart          = 2
io_form_input            = 2
io_form_boundary         = 11
io_form_auxinput2        = 11
io_form_auxhist2         = 11
debug_level              = 10
nocolons                 = .true.,
auxinput4_inname         = "wrflowinp_d<domain>",
auxinput4_interval       = 1440, 1440, 1440,
io_form_auxinput4        = 2,
auxinput1_inname         = "/data/derecho/met_files/ERA5/met_em.d<domain>.<date>"
iofields_filename        = "my_file_d01.txt", "my_file_d02.txt", "my_file_d03.txt",
ignore_iofields_warning  = .true.,
auxhist1_outname         =
"/data/derecho/WRF_output/NE_2012/aux1/auxhist1_d<domain>_<date>"
auxhist1_interval        = 60, 60, 60,
frames_per_auxhist1      = 1,  1, 1,
io_form_auxhist1         = 11,
output_diagnostics       = 1,
```

Shepherd T.J., Letson F., Barthelmie R.J. and Pryor S.C. How well are hazards associated with derechos reproduced in regional climate simulations? *Natural Hazards and Earth System Sciences Discussions (*nhess-2021-373)

**Response to reviewer #1:**

```
 auxhist3_outname            =
"/data/derecho/WRF_output/NE_2012/wrfout/wrfxtrm_d<domain>_<date>"
 auxhist3_interval           = 60, 10, 10,
 frames_per_auxhist3         = 1, 1, 1,
 io_form_auxhist3            = 11,
/
 &fdda
 grid_fdda                   = 1, 1, 1,
 gfdda_inname                = "wrffdda_d<domain>",
 gfdda_end_h                 = 144, 144, 144,
 gfdda_interval_m            = 360, 360, 360,
 fgdt                        = 0, 0, 0,
 if_no_pbl_nudging_uv        = 1, 1, 1,
 if_no_pbl_nudging_t         = 1, 1, 1,
 if_no_pbl_nudging_q         = 1, 1, 1,
 if_zfac_uv                  = 1, 1, 1,
 k_zfac_uv                   = 20, 20, 20,
 if_zfac_t                   = 1, 1, 1,
 k_zfac_t                    = 20, 20, 20,
 if_zfac_q                   = 1, 1, 1,
 k_zfac_q                    = 20, 20, 20,
 guv                         = 0.0003, 0.0003, 0.0003,
 gt                          = 0.0003, 0.0003, 0.0003,
 gq                          = 0.0003, 0.0003, 0.0003,
 if_ramping                  = 1,
 dtramp_min                  = 60.0,
 io_form_gfdda               = 11,
/
 &domains
 time_step                   = 30,
 time_step_fract_num         = 0,
 time_step_fract_den         = 1,
 max_dom                     = 3,
 e_we                        = 175,   262,   295,
 e_sn                        = 175,   262,   295,
 e_vert                      = 41,    41,   41,
 p_top_requested             = 5000,
 sfcp_to_sfcp                = .true.,
 num_metgrid_levels          = 38,
 num_metgrid_soil_levels     = 4,
 dx                          = 12000, 4000, 1333.33,
 dy                          = 12000, 4000, 1333.33,
 grid_id                     = 1,    2,    3,
 parent_id                   = 1,    1,    2,
 i_parent_start              = 1,    60,   105,
 j_parent_start              = 1,    35,   75,
 parent_grid_ratio           = 1,    3,    3,
```

Shepherd T.J., Letson F., Barthelmie R.J. and Pryor S.C. How well are hazards associated with derechos reproduced in regional climate simulations? *Natural Hazards and Earth System Sciences Discussions* (nhess-2021-373)

**Response to reviewer #1:**

```
parent_time_step_ratio        = 1,    3,   3,
feedback                = 0,
max_ts_locs              = 0,
eta_levels                = 1.0000 , 0.9958 , 0.9916 , 0.9874 , 0.9832 ,
                  0.9790 , 0.9749 , 0.9707 , 0.9661 , 0.9609 ,
                  0.9549 , 0.9480 , 0.9398 , 0.9303 , 0.9189 ,
                  0.9054 , 0.8894 , 0.8704 , 0.8481 , 0.8221 ,
                  0.7922 , 0.7583 , 0.7205 , 0.6791 , 0.6346 ,
                  0.5877 , 0.5393 , 0.4900 , 0.4407 , 0.3922 ,
                  0.3450 , 0.2996 , 0.2564 , 0.2156 , 0.1773 ,
                  0.1417 , 0.1086 , 0.0755 , 0.0475 , 0.0224 ,
                  0.0000,
/
&physics
mp_physics               = 9,    9,   9,
ra_lw_physics             = 1,    1,   1,
ra_sw_physics             = 1,    1,   1,
radt                = 10,   10,   10,
sf_sfclay_physics            = 1,    1,   1,
sf_surface_physics            = 2,    2,   2,
bl_pbl_physics              = 5,    5,   5,
bldt                = 0,    0,   0,
cu_physics               = 1,    0,   0,
cudt                = 5,
isfflx               = 1,
ifsnow                = 1,
icloud                = 1,
surface_input_source          = 3,
num_soil_layers            = 4,
num_land_cat             = 21,
sf_urban_physics             = 0,    0,    0,
bl_mynn_tkebudget             = 1,    1,    1,
bl_mynn_tkeadvect             = .true., .true., .true.,
rdmaxalb                = .false.,
sst_update               = 1,
tmn_update               = 1,
usemonalb                = .true.,
lagday                = 150,
sst_skin                = 1,
slope_rad               = 1,  1,  1,
prec_acc_dt               = 60., 10., 10.,
fractional_seaice             = 1,
seaice_threshold              = 0.,
/
&noah_mp
dveg                = 4,
opt_crs               = 1,
```

Shepherd T.J., Letson F., Barthelmie R.J. and Pryor S.C. How well are hazards associated with derechos reproduced in regional climate simulations? *Natural Hazards and Earth System Sciences Discussions (*nhess-2021-373)

**Response to reviewer #1:**

```
opt_btr                 = 2,
opt_run                 = 3,
opt_sfc                 = 1,
opt_frz                 = 1,
opt_inf                 = 1,
opt_rad                 = 3,
opt_alb                 = 2,
opt_snf                 = 4,
opt_tbot                = 1,
opt_stc                 = 3,
/
&dynamics
w_damping                  = 1,
diff_opt                = 1,    1,     1,
km_opt                  = 4,    4,     4,
diff_6th_opt              = 0,    0,     0,
diff_6th_factor           = 0.12,  0.12,   0.12,
base_temp               = 290.
damp_opt                 = 0,
zdamp                    = 5000.,  5000.,  5000.,
dampcoef                 = 0.01,   0.01,  0.01,
khdif                   = 0,    0,
kvdif                   = 0,    0,
non_hydrostatic             = .true., .true., .true.,
/
&bdy_control
spec_bdy_width              = 5,
spec_zone               = 1,
relax_zone              = 4,
spec_exp                 = 0.13
specified               = .true., .false., .false.,
nested                  = .false., .true., .true.,
/
&grib2
/
&namelist_quilt
nio_tasks_per_group = 0,
nio_groups = 1,
/
```

Shepherd T.J., Letson F., Barthelmie R.J. and Pryor S.C. How well are hazards associated with derechos reproduced in regional climate simulations? *Natural Hazards and Earth System Sciences Discussions* (nhess-2021-373)

**Response to reviewer #1:**

**ERA-I nudged namelist**

```
&time_control
run_days                = 6,
run_hours               = 0,
run_minutes             = 0,
run_seconds             = 0,
start_year              = 2012,  2012, 2012,
start_month             = 06,    06,   06,
start_day               = 26,    26,   26,
start_hour              = 00,    00,   00,
start_minute            = 00,    00,   00,
start_second            = 00,    00,   00,
end_year                = 2012, 2012, 2012,
end_month               = 07,    07,   07,
end_day                 = 02,    02,   02,
end_hour                = 00,    00,   00,
end_minute              = 00,    00,   00,
end_second              = 00,    00,   00,
interval_seconds        = 21600
input_from_file         = .true.,.true.,.true.,
history_interval        = 60, 10, 10,
frames_per_outfile      = 1,  1, 1,
history_outname         =
"/data/derecho/WRF_output/NE_2012/wrfout/wrfout_d<domain>_<date>"
restart                 = .false.,
restart_interval        = 1440,
override_restart_timers = .true.,
io_form_history         = 11
io_form_restart         = 2
io_form_input           = 2
io_form_boundary        = 11
io_form_auxinput2       = 11
io_form_auxhist2        = 11
debug_level             = 10
nocolons                = .true.,
auxinput4_inname        = "wrflowinp_d<domain>",
auxinput4_interval      = 1440, 1440, 1440,
io_form_auxinput4       = 2,
auxinput1_inname        = "/data/derecho/met_files/ERA-I/met_em.d<domain>.<date>"
iofields_filename       = "my_file_d01.txt", "my_file_d02.txt", "my_file_d03.txt",
ignore_iofields_warning = .true.,
auxhist1_outname        =
"/data/derecho/WRF_output/NE_2012/aux1/auxhist1_d<domain>_<date>"
auxhist1_interval       = 60, 60, 60,
frames_per_auxhist1     = 1, 1, 1,
io_form_auxhist1        = 11,
output_diagnostics      = 1,
```

Shepherd T.J., Letson F., Barthelmie R.J. and Pryor S.C. How well are hazards associated with derechos reproduced in regional climate simulations? *Natural Hazards and Earth System Sciences Discussions (*nhess-2021-373)

**Response to reviewer #1:**

```
 auxhist3_outname              =
"/data/derecho/WRF_output/NE_2012/wrfout/wrfxtrm_d<domain>_<date>"
 auxhist3_interval             = 60, 10, 10,
 frames_per_auxhist3           = 1, 1, 1,
 io_form_auxhist3              = 11,
/
 &fdda
 grid_fdda                     = 1, 1, 1,
 gfdda_inname                  = "wrffdda_d<domain>",
 gfdda_end_h                   = 144, 144, 144,
 gfdda_interval_m              = 360, 360, 360,
 fgdt                          = 0, 0, 0,
 if_no_pbl_nudging_uv          = 1, 1, 1,
 if_no_pbl_nudging_t           = 1, 1, 1,
 if_no_pbl_nudging_q           = 1, 1, 1,
 if_zfac_uv                    = 1, 1, 1,
 k_zfac_uv                     = 20, 20, 20,
 if_zfac_t                     = 1, 1, 1,
 k_zfac_t                      = 20, 20, 20,
 if_zfac_q                     = 1, 1, 1,
 k_zfac_q                      = 20, 20, 20,
 guv                           = 0.0003, 0.0003, 0.0003,
 gt                            = 0.0003, 0.0003, 0.0003,
 gq                            = 0.0003, 0.0003, 0.0003,
 if_ramping                    = 1,
 dtramp_min                    = 60.0,
 io_form_gfdda                 = 11,
/
 &domains
 time_step                     = 30,
 time_step_fract_num           = 0,
 time_step_fract_den           = 1,
 max_dom                       = 3,
 e_we                          = 175,   262,   295,
 e_sn                          = 175,   262,   295,
 e_vert                        = 41,    41,    41,
 p_top_requested               = 5000,
 sfcp_to_sfcp                  = .true.,
 num_metgrid_levels            = 61,
 num_metgrid_soil_levels       = 4,
 dx                            = 12000, 4000, 1333.33,
 dy                            = 12000, 4000, 1333.33,
 grid_id                       = 1,    2,    3,
 parent_id                     = 1,    1,    2,
 i_parent_start                = 1,    60,   105,
 j_parent_start                = 1,    35,   75,
 parent_grid_ratio             = 1,    3,    3,
```

Shepherd T.J., Letson F., Barthelmie R.J. and Pryor S.C. How well are hazards associated with derechos reproduced in regional climate simulations? *Natural Hazards and Earth System Sciences Discussions (*nhess-2021-373)

**Response to reviewer #1:**

```
parent_time_step_ratio        = 1,    3,   3,
feedback                = 0,
max_ts_locs              = 0,
eta_levels                = 1.0000 , 0.9958 , 0.9916 , 0.9874 , 0.9832 ,
                      0.9790 , 0.9749 , 0.9707 , 0.9661 , 0.9609 ,
                      0.9549 , 0.9480 , 0.9398 , 0.9303 , 0.9189 ,
                      0.9054 , 0.8894 , 0.8704 , 0.8481 , 0.8221 ,
                      0.7922 , 0.7583 , 0.7205 , 0.6791 , 0.6346 ,
                      0.5877 , 0.5393 , 0.4900 , 0.4407 , 0.3922 ,
                      0.3450 , 0.2996 , 0.2564 , 0.2156 , 0.1773 ,
                      0.1417 , 0.1086 , 0.0755 , 0.0475 , 0.0224 ,
                      0.0000,
/
 &physics
 mp_physics                = 9,    9,   9,
 ra_lw_physics              = 1,    1,   1,
 ra_sw_physics              = 1,    1,   1,
 radt                = 10,   10,   10,
 sf_sfclay_physics            = 1,    1,   1,
 sf_surface_physics            = 2,    2,   2,
 bl_pbl_physics              = 5,    5,   5,
 bldt                = 0,    0,   0,
 cu_physics                = 1,    0,   0,
 cudt                = 5,
 isfflx                = 1,
 ifsnow                 = 1,
 icloud                = 1,
 surface_input_source          = 3,
 num_soil_layers            = 4,
 num_land_cat              = 21,
 sf_urban_physics            = 0,    0,   0,
 bl_mynn_tkebudget            = 1,    1,   1,
 bl_mynn_tkeadvect            = .true., .true., .true.,
 rdmaxalb              = .false.,
 sst_update              = 1,
 tmn_update              = 1,
 usemonalb              = .true.,
 lagday                = 150,
 sst_skin              = 1,
 slope_rad              = 1,  1,  1,
 prec_acc_dt              = 60., 10., 10.,
 fractional_seaice            = 1,
 seaice_threshold            = 0.,
/
 &noah_mp
 dveg                = 4,
 opt_crs                = 1,
```

Shepherd T.J., Letson F., Barthelmie R.J. and Pryor S.C. How well are hazards associated with derechos reproduced in regional climate simulations? *Natural Hazards and Earth System Sciences Discussions (*nhess-2021-373)

**Response to reviewer #1:**

```
opt_btr                 = 2,
opt_run                 = 3,
opt_sfc                 = 1,
opt_frz                 = 1,
opt_inf                 = 1,
opt_rad                 = 3,
opt_alb                 = 2,
opt_snf                 = 4,
opt_tbot                = 1,
opt_stc                 = 3,
/
&dynamics
w_damping               = 1,
diff_opt                = 1,    1,     1,
km_opt                  = 4,    4,     4,
diff_6th_opt            = 0,    0,     0,
diff_6th_factor         = 0.12,  0.12,  0.12,
base_temp               = 290.
damp_opt                = 0,
zdamp                   = 5000., 5000., 5000.,
dampcoef                = 0.01,   0.01, 0.01,
khdif                   = 0,    0,
kvdif                   = 0,    0,
non_hydrostatic         = .true., .true., .true.,
/
&bdy_control
spec_bdy_width          = 5,
spec_zone               = 1,
relax_zone              = 4,
spec_exp                = 0.13
specified               = .true., .false., .false.,
nested                  = .false., .true., .true.,
/
&grib2
/
&namelist_quilt
nio_tasks_per_group = 0,
nio_groups = 1,
/
```

---

## Author Comment (AC2)

Shepherd T.J., Letson F., Barthelmie R.J. and Pryor S.C. How well are hazards associated with derechos reproduced in regional climate simulations? *Natural Hazards and Earth System Sciences Discussions (*nhess-2021-373)

**Response to reviewer #2:**

We thank the reviewer for their thoughtful and thought-provoking comments. Below we provide a complete list of their comments (in black) and our responses (in green). We further note in brief how this material will be included in the manuscript once we have received all reviews.

Review of nhess-2021-373: How well are hazards associated with derechos reproduced in regional climate simulations?
The authors used WRF as a convective-permitting regional climate model to produce 11 simulations of a severe derecho that affected the northeastern U.S. in 2012. The derecho had a major impact in terms of property damage and power outages over a large, populated region. This derecho was poorly forecast, yet we need to understand how climate change will impact such extreme mesoconvective systems. The authors examined the role of microphysical parametrization, nudging, and two different reanalysis products on 3km simulations of several days leading up to and including the derecho. The authors compared model output to surface and radar observations of precipitation, wind gust, hail, as well as variables describing the convective environment, such as vertical velocities and cold pool formation. The explanation of the methods of model assessment was particularly thoughtful.
Overall, the manuscript is well constructed with clear objectives, detailed methodology, and significant findings. I recommend that the manuscript be accepted following minor revisions.

Major comments

1. While most of the simulations poorly represented the derecho, this is not surprising given that this event was not well predicted. While I would not ask the authors to address in this manuscript, it would be intriguing to duplicate this work for a significant mesoconvective system that was well predicted.

Response: We appreciate the reviewer's comment. We do indeed plan to explore this work further, particularly in the context of pseudo global warming studies. As per minor comment 10, we have included references to such studies that are relevant to mesoscale convective systems in North America.

2. While the use of the different microphysical parameterizations in the model was well designed, it is unclear what is learned by the comparison of ERA5 and ERA-Interim. It is not clear to me that you can generalize that ERA-Interim is inherently better at producing boundary conditions for such simulations (l. 526-527), or whether the small differences in the pre-storm temperature and moisture fields in ERA-Interim (l. 413-415) fortuitously produced more realistic simulations.

Response: We thank the reviewer for presenting this concern. We do agree that the over-generalization of the initial condition results is not warranted. For this reason, we were particularly careful with the language used to not assert one dataset as being more accurate, but rather to explain why the simulations differed. We note the reviewer's concern, however, and have revised the manuscript text. The text at line 526-527 has been amended to (line 621 in the

Shepherd T.J., Letson F., Barthelmie R.J. and Pryor S.C. How well are hazards associated with derechos reproduced in regional climate simulations? *Natural Hazards and Earth System Sciences Discussions (*nhess-2021-373)

**Response to reviewer #2:**

revised text): "Additional simulations using ERA5 and ERA-I are required before generalizable conclusions can be made about which dataset provides better boundary conditions."

For lines 413-415, we note that we have already revised the manuscript text in response to reviewer 1 and their comments regarding the simulations. After additional research, the elevated mixed layer (EML) proved to be critical in this event. We note that there was a high degree of similarity between the two reanalysis products, but the key difference between the two initial conditions datasets is the coherency of the EML. This is one factor in the difference between simulations with ERA5 and ERA-I. The revised text reads (lines 480-492 in the revised manuscript):
"Evaluation of the initial conditions indicates a high degree of similarity between the two reanalysis products on 26 and 28 June for most properties (Figure 12). However, as described above, development of an intense elevated mixed layer (EML, 700-500 hPa) over the central US that subsequently propagated eastwards (Shourd and Kaplan, 2021) appears to have been a key ingredient in development of this Derecho. Earlier work (Banacos and Ekster, 2010) employed a definition of an EML as a layer of depth > 200 hPa with both a steep lapse rate (temperature declines of over 8°C per km) and an increase in the RH with height. Figure 12 shows the lapse rate in the four sets of IC and indicates that while both data sets correctly (relative to output from NOAA WRF-Rapid Refresh model presented in (Shourd and Kaplan, 2021)) indicate relatively low lapse rates at 0000Z 26 June (when the region with the EML was displaced further west), using the combined definition of a strong lapse rate and a strong gradient of RH (a 20% difference across the layer), the EML is, in both reanalysis products, displaced too far north at 0000Z 28 June relative to NOAA WRF-Rapid Refresh model simulations presented in (Shourd and Kaplan, 2021). The EML is, however, more consistent (across the two components) and more coherent in space in ERA-Interim. This may provide a partial explanation for why simulations with ERA-Interim initial and lateral boundary conditions exhibit higher fidelity with respect to aspects of the Derecho."

References

Banacos, P. C. and Ekster, M. L.: The Association of the Elevated Mixed Layer with Significant Severe Weather Events in the Northeastern United States, Weather and Forecasting, 25, 1082-1102, 10.1175/2010waf2222363.1, 2010.

Shourd, K. N. and Kaplan, M. L.: The Multiscale Dynamics of the 29 June 2012 Super Derecho, Climate, 9, 155, 2021

3. I share the concern with the first reviewer that the abstract was not sufficiently specific, but I find the proposed new abstract in the authors' response to be a significant improvement that addresses this concern.

Shepherd T.J., Letson F., Barthelmie R.J. and Pryor S.C. How well are hazards associated with derechos reproduced in regional climate simulations? *Natural Hazards and Earth System Sciences Discussions (*nhess-2021-373)

**Response to reviewer #2:**

Response: We appreciate the reviewer's comment and thank them for reading the reply to reviewer 1 where we responded with the improved abstract.

4. I share the first reviewer's concern about the need for additional context on the convective environment for this storm in the background. The additions proposed by the authors appear to address this concern.

Response: Again, we thank the reviewer for reading the reply to reviewer 1 and we are pleased that the additions address these concerns.

Minor
1. 44: Is "atmospheric phenomena" another way of saying "weather"? Or is it intended to capture more.

Response: We are using this term in the AMS glossary sense; **atmospheric phenomenon**
As commonly used in weather observing practice, an observable occurrence of particular physical (as opposed to dynamic or synoptic) significance within the atmosphere. Included are all hydrometeors (except clouds, which are usually considered separately), lithometeors,  igneous meteors, and luminous meteors. From the viewpoint of weather observations,  thunderstorms, tornadoes, waterspouts, and squalls are also included. The above usage excludes such "phenomena" as the local or large-scale characteristics of wind, pressure, and temperature; it also excludes clouds, although it includes many products of cloud development and composition. In aviation weather observation, atmospheric phenomena are divided into two categories: weather and obstructions to vision."
Which we believe is differentiable from 'weather', which again according to the AMS glossary has the following definition: **weather**
The state of the atmosphere, mainly with respect to its effects upon life and human activities.
As distinguished from climate, weather consists of the short-term (minutes to days) variations in the atmosphere. Popularly, weather is thought of in terms of temperature, humidity, precipitation,  cloudiness, visibility, and wind.
1. As used in the taking of surface weather observations, a category of individual and combined atmospheric phenomena that must be drawn upon to describe the local atmospheric activity at the time of observation.
Listed weather types include tornado, waterspout, funnel cloud, thunderstorm and severe storm, liquid precipitation(drizzle, rain, rain showers), freezing precipitation (freezing drizzle,  freezing rain), and frozen precipitation (snow, snow pellets, snow grains, hail, ice pellets,  ice crystals). These elements, with the exception of the first three, are denoted by a letter code in the observation. With the METAR code, reporting weather also includes an intensity qualifier (light, moderate, or heavy) or proximity qualifier. The weather used in synoptic weather observations and marine weather observations is reported in two categories, "present weather" and "past weather." The "present weather" table consists of 100 possible conditions, with 10 possibilities for "past weather"; both are encoded numerically. Another method, which has the advantage of

Shepherd T.J., Letson F., Barthelmie R.J. and Pryor S.C. How well are hazards associated with derechos reproduced in regional climate simulations? *Natural Hazards and Earth System Sciences Discussions (*nhess-2021-373)

**Response to reviewer #2:**

being independent of language, is the recording of weather types using symbols. There are 100 symbols that identify with the numeric codes of the synoptic observation.

2. To undergo change due to exposure to the atmosphere.'

So, we perceive 'weather' as the totality of the atmospheric state and atmospheric phenomena as notable (perhaps even atypical) conditions that may derive from a singular event (e.g. downdrafts and lightning from MCS).

Nevertheless, given the reviewers concerns we have changed 'atmospheric phenomena' to 'weather events' (line 50 in the revised manuscript).

2. 54: focuses

Response: Thanks. Corrected.

3. 51: "function model configuration" – appears a word is missing

Response: Thanks. We have corrected the sentence to read "…as a function of model configuration…" (line 57 in the revised manuscript).

4. 56: Is "advected" the right word? Perhaps "propagated"?

Response: Thanks for this suggestion. We now use 'propagated' instead of 'advected' (line 63).

5. 66: run-on sentence

Response: Thanks for spotting this. The sentence now reads (line 72): "Over 20 deaths were reported during the 29-30 June 2012 derecho event. There was also widespread property damage and extensive power outages (Halverson, 2014)."

6. 88-89: it might be helpful to explain "scale-aware convective parameterizations". How are they "scale-aware"?

Response: Thanks for this suggestion. We have added a brief definition of what scale-aware convection schemes are, by changing the sentence to read (line 118): "Emerging research has shown that using scale-aware convective parameterizations (i.e. those schemes where numerical descriptions include a parameter that modulates convective processes as a function of horizontal resolution) throughout the model gray zone resolution…".

7. 93: "degree/manner in what the model parameterization interact" is unclear. What is this trying to say?

Response: Thanks for bringing this to our attention. We have amended this sentence to remove ambiguity. The revised text in the manuscript will read (line 122): "…model fidelity is a strong function of the precise cloud microphysics scheme applied, model grid spacing, lateral boundary

Shepherd T.J., Letson F., Barthelmie R.J. and Pryor S.C. How well are hazards associated with derechos reproduced in regional climate simulations? *Natural Hazards and Earth System Sciences Discussions (*nhess-2021-373)

**Response to reviewer #2:**

conditions and the degree/manner in which the model parameterizations interact (for example, feedback between the cumulus parameterizations/cloud microphysics and the radiation scheme)…"

8. 306: Perhaps I'm missing something obvious, but why would s(w) be used as intensity for vertical motion rather than just w?

Response: We note that the text at line 308 states: "The height at which the maximum variability in vertical velocities occur is used to provide information regarding the vertical structure of convection." We have elaborated slightly to read: "The height at which the maximum standard deviation of vertical velocities ($\sigma(w)$) is used to infer the intensity and vertical structure of convection. Since updrafts and downdrafts are of relatively short duration, we use $\sigma(w)$ computed using vertical velocities output from the time of maximum cREF > 40 dBZ (i.e. from the 10-min time step WRF output file at that time) as a more descriptive as a metric rather than the mean velocity because the dispersion around the mean is reflective of the intensity of both downdrafts and updrafts in the column."

9. 322-323: This sentence (Rank correlation coefficients…") isn't clear. How does the rank correlation show which model property most greatly influences skill? The correlations show how well the model and observations agree, but the word "influences" suggests that you can determine a causal mechanism.

Response: We agree that this is ambiguous and could be misleading. We have amended the manuscript text to (line 368): "…to identify which model properties (wind speed, precipitation etc.) exhibit highest association with the diagnostic metrics used to examine model skill in simulating this event."

10. 501: The authors use the pseudo-global warming framework as a justification but don't provide references (perhaps I missed them) how this framework has been used to examine mesoconvective systems nor provide examples of how such a framework might be used.

Response: Thanks for raising this. There are indeed a few PGW studies related to MCSs, but not specifically to derecho events [e.g. Ikeda et al. (2010); Liu et al. (2017); Haberlie et al. (2019)]. We have added additional references to the manuscript text.

Thus, the previous text that read (line 521):
"Our finding has important implications for construction of hindcast simulations for use in Surrogate or Pseudo Global Warming (PGW) numerical experiments to quantify the potential of global warming on extreme weather events using regional models (Li et al., 2019). In such simulations an historically important extreme event is first simulated using contemporary LBC and then the simulation is repeated using LBC and IC perturbed to represent the change in, for example, air temperatures and water vapor availability. The difference in these two realizations

Shepherd T.J., Letson F., Barthelmie R.J. and Pryor S.C. How well are hazards associated with derechos reproduced in regional climate simulations? *Natural Hazards and Earth System Sciences Discussions (*nhess-2021-373)

**Response to reviewer #2:**

is interpreted as the impact of global climate non-stationarity. Our work indicates use of ERA5 for IC and LBC may not always result in improved baseline simulations of the extreme event in the contemporary climate, and the simulation deficiencies may render evaluation of the PGW response highly uncertain."

Has been modified to read:
"Our finding has important implications for construction of hindcast simulations for use in Surrogate or Pseudo Global Warming (PGW) numerical experiments to quantify the potential of global warming on extreme weather events using regional models (Kröner et al. 2017; Li et al., 2019). In such simulations an historically important extreme event/period/season is first simulated using contemporary LBC and then the simulation is repeated using LBC and IC perturbed to represent the change in, for example, air temperatures and water vapor availability (Kroner et al. 2017). The difference in these two realizations is interpreted as the impact of global climate non-stationarity. A previous analysis over CONUS used ERA-Interim LBC and shifted the atmospheric profile by ± 5 °C. They found increases in both CAPE and convective inhibition, which implies shift the convective population (Rasmussen et al. 2020). Our work indicates use of either ERA-Interim or ERA5 for IC and LBC may not always result in high-fidelity baseline simulations of extreme convective events in the contemporary climate. These simulation deficiencies may render evaluation of the PGW response highly uncertain."

References:

Haberlie, A. M., & Ashley, W. S. (2019). Climatological representation of mesoscale convective systems in a dynamically downscaled climate simulation. International Journal of Climatology, 39(2), 1144-1153.

Ikeda, K., Rasmussen, R., Liu, C., Gochis, D., Yates, D., Chen, F., Tewari, M., Barlage, M., Dudhia, J., Miller, K., Arsenault, K., Grubišić, V., Thompson, G. and Guttman, E. (2010) Simulation of seasonal snowfall over Colorado. *Atmospheric Research*, **97**, 462–477.

Kröner, N., Kotlarski, S., Fischer, E., Lüthi, D., Zubler, E., & Schär, C. (2017). Separating climate change signals into thermodynamic, lapse-rate and circulation effects: theory and application to the European summer climate. *Climate Dynamics*, *48*(9), 3425-3440.

Liu, C., Ikeda, K., Rasmussen, R., Barlage, M., Newman, A.J., Prein, A.F., Chen, F., Chen, L., Clark, M., Dai, A., Dudhia, J., Eidhammer, T., Gochis, D., Gutmann, E., Kurkute, S., Li, Y., Thompson, G. and Yates, D. (2017) Continental-scale convection-permitting modeling of the current and future climate of north america. *Climate Dynamics*, **49**, 71–95.

Rasmussen, K. L., Prein, A. F., Rasmussen, R. M., Ikeda, K., & Liu, C. (2020). Changes in the convective population and thermodynamic environments in convection-permitting regional climate simulations over the United States. *Climate Dynamics*, *55*(1), 383-408.

---

## Referee Report (RR1)

Thank you for your effort in understanding the capability of WRF model in reproducing the hazards associated with derechos. I enjoyed reading your manuscript and have a few suggestions to improve the manuscript.

A flowchart for the model simulation steps is missing from the manuscript. It is an important component, especially for a research paper that uses multiple model simulations and sensitivity cases. Authors should refer to the latest papers that demonstrate flowchart for the WRF model simulations, they can either provide references or produce a new flowchart for the reproducibility of similar WRF model simulations. Here are some suggestions for the relevant list of WRF literature:

Kumar, M., Kosović, B., Nayak, H.P., Porter, W.C., Randerson, J.T. and Banerjee, T., 2024. Evaluating the performance of WRF in simulating winds and surface meteorology during a Southern California wildfire event. *Frontiers in Earth Science*, *11*, p.1305124.

Saleh, N., Gharaylou, M., Farahani, M.M. and Alizadeh, O., 2023. Performance of lightning potential index, lightning threat index, and the product of CAPE and precipitation in the WRF model. *Earth and Space Science*, *10*(9), p.e2023EA003104.

Line 405: change "where XXX indicates" to "where XXXX indicates"

The font sizes of axes, ticks, titles, etc. are non-uniform in most of the figures (for example, Figures 9 & 10). Please be sure to make them uniform. The figure resolution needs to be enhanced for better readability as they appear to be of low resolution in the current version of the manuscript. In addition, some of the figure captions are inadequately written without sufficient information for the reader. Make sure to be consistent with the results and discussions provided.

---

## Author Response (AR2)

Shepherd T.J., Letson F., Barthelmie R.J. and Pryor S.C. How well are hazards associated with derechos reproduced in regional climate simulations? *Natural Hazards and Earth System Sciences Discussions* *(*nhess-2021-373)

**Response to reviewer #1:**

*We thank the reviewer for their ongoing evaluation of this work. Our response to the reviewer's comments is indicated in green for each comment.*

Note that in accordance with the reviewer's comments, we have undertaken two additional simulations with the Morrison scheme (no hail and with hail) to examine the sensitivity of this event to reviewer suggestions for: vertical levels (increased from 41 to 65 levels); and damping. All other settings remain as before. We detail the analyses of the new simulations that are presented in the paper and below here.

We appreciate that our research design is not exactly what the reviewer would have done but note that some of the disagreements really lie in personal preference regarding scheme selection and model configuration. All schemes employed here are accepted and well used schemes in WRF for many version iterations and have been subjected to extensive testing and development. We have generated a uniquely comprehensive ensemble suite and rigorously evaluated it and sought to link performance to dynamical drivers. We believe this work is valid and useful and hope that given the extraordinary lengths we have taken to address the reviewer's concerns (many additional simulations), this matter can be brought to a positive and successful conclusion.

**Reviewer 1 comments:**

The namelist that the authors provided shows some problems to me. It validates my concerns about how the simulations were conducted.

1. No shallow cumulus parameterization is used, which is surely a big problem for domain 1 which is 12 km resolution. For the 4-km, I would still recommend using it for convective system simulations.

We have used the shallow convection parameterization in the 12 km simulations. The Kain-Fritsch scheme used in the 12 km simulations (like all convective schemes) runs shallow convection by default. The user must change the Kain-Fritsch parameterization code manually and recompile WRF to switch shallow convection off (see lines 1393-94 of module_cu_kfeta.F).

NB: Perhaps the reviewer has misread the namelist provided and is reading from the line above which indicates the settings for boundary layer physics calls (0). The namelist states quite clearly that convective parameterization is used on the 12 km domain:

*bldt*                      *= 0,    0,    0,*
*cu_physics*          *= 1,    0,    0,*
*cudt*                     *= 5,*

While opinions vary, multiple previous studies have used no cumulus scheme at 4 km or below (i.e. "convection permitting" simulations) (see an example citation list below). In addition, Romine et al. (2013) states that: "*Several efforts in convection-permitting NWP (CP; horizontal grid spacing of 4 km or less), both operational and experimental, have commenced in the last decade. This is motivated in part by evidence that resolution sufficient to allow explicit representation of convection improves simulation of both the convection's diurnal characteristics (Done et al. 2004; Liu and Moncrieff 2007) and its mode and structure (Clark et al. 2007; Kain et al. 2008; Schwartz et al. 2009; Sobash et al. 2011).*"

Shepherd T.J., Letson F., Barthelmie R.J. and Pryor S.C. How well are hazards associated with derechos reproduced in regional climate simulations? *Natural Hazards and Earth System Sciences Discussions (*nhess-2021-373)

**Response to reviewer #1:**

Example citations of studies that have used only a microphysics scheme within the innermost domain:

- Khai Shen Sow, Liew Juneng, Fredolin T. Tangang, Abdul Ghapor Hussin, Mastura Mahmud, Numerical simulation of a severe late afternoon thunderstorm over Peninsular Malaysia, *Atmospheric Research*, 99, 2, 2011, https://doi.org/10.1016/j.atmosres.2010.10.014: model domains of 27, 9, 3km. "*In the third domain, several cumulus schemes were examined but the simulation result is the best with the cumulus process explicitly resolved*".

- Rajeevan, M., Kesarkar, A., Thampi, S. B., Rao, T. N., Radhakrishna, B., and Rajasekhar, M.: Sensitivity of WRF cloud microphysics to simulations of a severe thunderstorm event over Southeast India, Ann. Geophys., 28, 603–619, https://doi.org/10.5194/angeo-28-603-2010 2010: BMJ at 12, 8km, no cumulus parameterization at 2km.

- Garcia-Carreras, L., A. J. Challinor, B. J. Parkes, C. E. Birch, K. J. Nicklin, and D. J. Parker, 2015: The Impact of Parameterized Convection on the Simulation of Crop Processes. *J. Appl. Meteor. Climatol.*, 54, 1283–1296, https://doi.org/10.1175/JAMC-D-14-0226.1: "*The parameterized runs produced an unrealistic distribution of rainfall frequencies and intensities, a well-known issue with parameterizations of convection. The parameterization leads to too frequent light rainfall events (in time and space), with too few heavy rainfall showers and days with no rainfall at all. The two convection-permitting runs, on the other hand, had rainfall distributions much closer to observations, although peak rainfall intensities and rainfall totals were overestimated.*"

- Halder, M., Mukhopadhyay, P. Microphysical processes and hydrometeor distributions associated with thunderstorms over India: WRF (cloud-resolving) simulations and validations using TRMM. Nat Hazards 83, 1125–1155 (2016). https://doi.org/10.1007/s11069-016-2365-2: *Nested domains with increasing resolution of 27-, 9-, 3- and 1-km grid spacing ... Kain-Fritsch used in the 27 and 9 km domain only.*

- Xing Yu & Tae-Young Lee (2010) Role of convective parameterization in simulations of a convection band at grey-zone resolutions, Tellus A: Dynamic Meteorology and Oceanography, 62:5, 617-632, DOI: 10.1111/j.1600-0870.2010.00470.x: "*The impacts of resolution and convective parameterization at grey-zone resolutions (i.e. 3, 6 and 9 km) are then investigated. Results indicate that a grid size of 3 km is sufficient to resolve the convection band and CP for this size grid is not necessary. With 6 and 9 km grids, explicit simulations or those based on a Kain–Fritsch CP scheme do not simulate the atmospheric structure surrounding the band accurately*".

- Bodine, D. J., and K. L. Rasmussen, 2017: Evolution of Mesoscale Convective System Organizational Structure and Convective Line Propagation. Mon. Wea. Rev., 145, 3419–3440, https://doi.org/10.1175/MWR-D-16-0406.1. *27, 9, 3km domains with 44 vertical levels, and Kain-Fritsch used in the outer domains only (i.e. 27, 9 km).*

Weisman et al. (2013) NB: already cited in our study, used only a microphysics scheme in a 3km domain WRF simulation of an unusually intense bow echo and associated mesoscale vortex that were responsible for producing an extensive swath of high winds across Kansas, southern Missouri, and southern Illinois on 8 May 2009 (superderecho).

Shepherd T.J., Letson F., Barthelmie R.J. and Pryor S.C. How well are hazards associated with derechos reproduced in regional climate simulations? *Natural Hazards and Earth System Sciences Discussions (*nhess-2021-373)

**Response to reviewer #1:**

2. The radiation timesteps (Radt) are not set appropriately. It is recommended to be 1 min per km of dx based on the guide so the values for Radt should be 12, 4, and 1.3 for Domain 1, 2, and 3. The authors use 10 min for all three domains, which is particularly not appropriate for Domain 3 where the results of the study are mainly obtained from.

We have not made changes in response to this suggestion. Our reasoning is that the radiation scheme time step is set on domain 1 only and cannot be set to a different value on nested domains. From the WRF user guide on page 5-78: "minutes between radiation physics calls. Recommended 1 minute per km of dx (e.g. 10 for 10 km grid); use the same value for all nests".
See:
https://www2.mmm.ucar.edu/wrf/users/docs/user_guide_V3/user_guide_V3.8/ARWUsersGuideV3.8.pdf

3. The vertical resolution of 41 is too coarse for convective cloud system simulations.

We have implemented this suggestion and have included two new simulations using 65 vertical levels.

We appreciate the reviewer's concern. We note that using 41 levels is more than what is used by default in WRF operational/research application namelists provided by NCAR (e.g. 1 – 3 km convection permitting or hurricane applications). There is precedent in the literature for using ~40 vertical levels for WRF simulation of mesoscale convective systems (e.g. Bodine and Rasmussen (2017) used 44 vertical levels).

We have, however, as part of the two additional simulations, increased the number of vertical levels to 65. Figure 1 below shows a comparison of model vertical resolution and height above the surface between the 11 original ensemble members and the two new 65 level simulations. The change in resolution with height is more gradual and there is increased resolution above 0.5 km.

Shepherd T.J., Letson F., Barthelmie R.J. and Pryor S.C. How well are hazards associated with derechos reproduced in regional climate simulations? *Natural Hazards and Earth System Sciences Discussions (*nhess-2021-373)

**Response to reviewer #1:**

[Figure]

**Figure 1: Comparison of model vertical resolution between the 11 original 41 vertical level ensemble members, and the two new 65 level Morrison simulations.**

We now have an ensemble of 15 simulations. We reiterate that this study's objective is to examine the sensitivity of a real event to the model configuration. We have tested several hypotheses, and at the request and suggestion of the reviewer, added additional ensemble members under the hypothesis that with each iteration, the simulation fidelity should improve. We now have an increasingly large and comprehensive ensemble where we have tested sensitivities and which member most closely approaches reality. We would suggest that this is sufficient to provide a useful and interesting contribution for the current work to be published.

In summary:
1. The original work was an 11-member ensemble with 41 vertical levels, using the GNU Fortran compiler and initialized with ERA5 that varied only in terms of microphysics package. As part of this ensemble:
    a. The Morrison hail flag was also tested.
    b. Lateral boundary condition sensitivity (ERA-I vs. ERA5) was also explored using one microphysics scheme (Milbrandt).
    c. Sensitivity to model start time was explored using one microphysics scheme (Milbrandt).
2. After initial peer-review, compiler sensitivity was tested. One microphysics scheme was selected (Morrison), one LBC chosen (ERA5), with two simulations (no hail flag, with hail flag) using the Intel compiler. There were now 13 ensemble members.
3. After second peer-review, the sensitivity to the vertical resolution (and damping – see reviewer comment #4 below) has been tested, using the method in 2 above. There are now 15 ensemble members.

Shepherd T.J., Letson F., Barthelmie R.J. and Pryor S.C. How well are hazards associated with derechos reproduced in regional climate simulations? *Natural Hazards and Earth System Sciences Discussions* (nhess-2021-373)

**Response to reviewer #1:**

The two new simulations using 65 vertical levels with damping have been integrated into the manuscript. Each manuscript figure now contains all 15 ensemble members. Tables 3 – 5 have also been updated to include the two new simulations using 65 vertical levels.

In the following 5 comparison figures (Figure 2 – 6 in this reply) and subsets of the manuscript tables 3 – 5 we compare the new simulations (both have '65L' in their displayed names in the figures) to the Morrison and Morrison+Hail results that were part of the original 11-member ensemble in the paper. Note that Figures 2 – 6 in this reply correspond to Figures 4 – 8 from the manuscript.

Along with the comparison figures below, a subset for each of Tables 3 – 5 is shown here to illustrate how the two 65 level simulations compare to the previous Morrison simulations. From Table 3, the simulations with 65 levels do not result in improved agreement in terms of the spatial fields of accumulated precipitation (e.g. see ASOS/RADAR column for precipitation, for example: Morrison 0.187 vs. Morrison 65levels 0.1591). The only significant improvement is against ASOS wind gusts in the Morrison 65levels+Hail simulation vs. the original Morrison+Hail simulation.

**Table 3: Spearman rank correlations for the spatial fields of maximum wind gusts in domain d03 during the derecho (Derecho period: 29-Jun-2012 21:30:00 to 30-Jun-2012 13:30:00) from WRF and ASOS observations. In this analysis WRF output for maximum time step wind speeds (dt = 6 sec) is sampled at the 34 ASOS locations and compared with the maximum 3-second ASOS wind gusts measurements (see spatial fields in Figure 11). Also shown are the Spearman rank correlations between spatial fields of total accumulated precipitation from WRF output relative to RADAR estimates and ASOS in situ measurements. In these analyses the correlations between WRF and the RADAR data are for all WRF grid cells sampled by the RADAR (99.4% of d03), while the comparison with ASOS measurements is for the 34 ASOS stations. The final column shows the correlations between the spatial fields of maximum composite reflectivity cREF (again in any time step during the Derecho period) from the WRF ensemble members and RADAR.**

| Ensemble member | ASOS | | RADAR | |
|---|---|---|---|---|
| | Wind gusts | Precipitation | Precipitation | cREF |
| Morrison | 0.312 | 0.063 | 0.187 | 0.199 |
| Morrison+Hail | -0.557 | 0.138 | 0.181 | 0.255 |
| Morrison 65levels | 0.3418 | 0.0059 | 0.1591 | 0.2054 |
| Morrison 65levels+Hail | 0.2030 | 0.1522 | 0.2604 | 0.2561 |

**Table 4: Number of grid cells in domain d03 where hail is indicated by the RADARs or present in the WRF simulations during the derecho (Derecho period: 29-Jun-2012 21:30:00 to 30-Jun-2012 13:30:00) and the frontal passage (Front period: 30-Jun-2012 15:20:00 to 01-Jul-2012 14:50:00). Also shown is the number of grid cells with Maximum Estimated Size of Hail (MESH) above 25 mm from the RADAR or WRF. Recall: RADAR detection of hail is re-gridded onto the WRF grid used for domain d03 prior to use in the model evaluation.**

| | # Grid cells with hail | | # Grid cells with hail values > threshold | |
|---|---|---|---|---|
| | Derecho | Front | Derecho | Front |
| RADAR | 3078 | 2152 | 824 | 813 |
| Ensemble member | | | | |
| Morrison | 0 | 24 | 0 | 0 |
| Morrison+Hail | 3000 | 74398 | 0 | 0 |
| Morrison 65levels | 2 | 36 | 0 | 0 |

Shepherd T.J., Letson F., Barthelmie R.J. and Pryor S.C. How well are hazards associated with derechos reproduced in regional climate simulations? *Natural Hazards and Earth System Sciences Discussions (*nhess-2021-373)

**Response to reviewer #1:**

| Morrison 65levels+Hail | 37030 | 72997 | 1 | 75 |
|---|---|---|---|---|

For Table 4, to recap from the manuscript: *"When remapped to the WRF grid, the RADAR data indicate 824 of the almost 90,000 grid cells experienced severe hail during the Derecho period (Table 4). These locations identified by the RADAR detection algorithm as exhibiting hail and MESH > 25 mm ..."*
The new Morrison simulation using 65 levels with the hail flag on (Morrison 65levels+Hail), shows greater spatial coverage of hail compared to the original Morrison+Hail simulation (# grid cells with hail for the derecho period is 37030 vs. 3000). This is reflected below in Figure 3 in this reply and in the revised manuscript. Note that greater spatial coverage does not equate to improved fidelity. The number of grid cells with RADAR detection of hail (3078) still shows closest agreement with the Morrison+Hail simulation (3000) (Table 4). The systematic finding of more grid cells with hail when using the hail flag turned on vs. leaving the flag off is also maintained. The front period remains largely unchanged (Table 4 and Figure 4 below), as does the number of grid cells when the threshold of MESH > 25 mm is applied (0 vs. 1 for the derecho period).

A subset of Table 5 in this reply directly compares the Morrison 41 vs. 65 level simulations for the simulation fidelity and convection metrics. The Milbrandt-626-ERA-I and Milbrandt-628-ERA-I metrics are included for reference as these were the best performing ensemble members. The addition of the two new Morrison 65 levels simulations does not change that conclusion. Comparing the Morrison cases directly, the 4 metrics for simulation fidelity confirm that using 65 levels for the Morrison scheme results in enhanced simulation fidelity with and without the hail flag. Compared to the two original Morrison simulations, these two new simulations have the highest agreement with observations of the spatial extent of high cREF, total precipitation accumulation, maximum wind gusts and large hail. The convection metrics are less conclusive, but there is comparatively high values for the 95[th] percentile downward vertical velocities in the 65 levels simulations.

Shepherd T.J., Letson F., Barthelmie R.J. and Pryor S.C. How well are hazards associated with derechos reproduced in regional climate simulations? *Natural Hazards and Earth System Sciences Discussions (*nhess-2021-373)

**Response to reviewer #1:**

**Table 5: Metrics of simulation fidelity relative to observations, and convection metrics derived from output from each original Morrison WRF member compared to the new Morrison 65 level simulations during the period of the derecho passage (Derecho period: 29-Jun-2012 21:30:00 to 30-Jun-2012 13:30:00). The metrics of simulation fidelity are described in section 2.2 and are as follows: The Max Gust Ratio: the ratio of the maximum wind gust in any land grid cell from WRF output and observations at the ASOS stations. Total Precip. Ratio: the ratio of the spatial mean total accumulated precipitation from WRF to RADAR and STAGE IV, respectively, for any grid cell with common coverage. cREF>40 dBZ: the ratio of the spatial extent of grid cells with cREF above 40 dBZ at the peak coverage in WRF and RADAR. The color-coding used in this table is as follows; for the measures of simulation fidelity table cells colored red have low fidelity, and those indicated by cyan exhibit relatively high fidelity. For all other cells in the table, a background of orange indicates low values, while blue indicates comparatively high values. The saturation of the color indicates relative ordering of the values. The definitions of each convection metric are given in section 2.3.**

| | Simulation Fidelity | | | | Convection Metric | | | | | |
|---|---|---|---|---|---|---|---|---|---|---|
| | Max Gust Ratio | Total Precip. Ratio (RADAR) | Total Precip. Ratio (Stage IV) | cREF>40 dBZ Ratio | 95% Temperature deviation [-K] | 95% SLP deviation [hPa] | Median CAPE loss [J kg⁻¹] | 95% -W [ms⁻¹] | Max std(w) height [km] | $Z_{R20}$ [km] |
| Morrison | 0.673 | 0.413 | 0.435 | 0.788 | 3.29 | 1.850 | 1532 | 0.151 | 8.0 | 15.3 |
| Morrison +Hail | 0.460 | 0.016 | 0.017 | 0.102 | 2.12 | -0.756 | 175 | 0.117 | 8.0 | 9.0 |
| Morrison 65levels | 0.693 | 0.454 | 0.479 | 0.972 | 4.35 | 2.726 | 1706 | 0.263 | 8.0 | 13.9 |
| Morrison 65levels+Hail | 0.820 | 0.508 | 0.536 | 0.850 | 4.47 | -0.193 | 1620 | 0.238 | 8.5 | 13.1 |
| Milbrandt-626-ERA-I | 0.633 | 0.566 | 0.597 | 0.844 | 5.44 | 2.360 | 1960 | 0.152 | 8.9 | 14.4 |
| Milbrandt-628-ERA-I | 0.695 | 0.636 | 0.671 | 0.945 | 5.58 | 2.790 | 2030 | 0.146 | 7.1 | 14.9 |

Shepherd T.J., Letson F., Barthelmie R.J. and Pryor S.C. How well are hazards associated with derechos reproduced in regional climate simulations? *Natural Hazards and Earth System Sciences Discussions (*nhess-2021-373)

**Response to reviewer #1:**

Figure 2 (below) shows a comparison between the original Morrison simulations with 41 levels and new Morrison simulations with 65 levels and damping for the number of grid cells in domain d03 with composite reflectivity (cREF) > 40 dBZ (Figure 2a). In the derecho period it is evident that the simulations with 65 levels show a greater number of grid cells with cREF > 40 dBZ.

[Figure]

**Figure 2: Time series of number of grid cells in domain d03 with composite reflectivity (cREF) > 40 dBZ from RADAR and 4 WRF ensemble members (original Morrison simulations with 41 levels, and new Morrison simulations with 65 levels and damping). The number of the 34 ASOS stations in domain d03 reporting thunderstorms is shown in grey (right axis). The timing of the (Derecho period: 29-Jun-2012 21:30:00 to 30-Jun-2012 13:30:00) and the frontal passage (Front period: 30-Jun-2012 15:20:00 to 01-Jul-2012 14:50:00) are denoted by the grey backgrounds. (b) The number of grid cells in domain d03 where output from each WRF ensemble member or the RADARs exceeded the specified threshold during the time step within the derecho period when the maximum number of grid cells exceeded the threshold. For example, in the RADAR observations there is a single 10-minute period during which approximately 5000 grid cells exhibit a value above 40 dBZ.**

Shepherd T.J., Letson F., Barthelmie R.J. and Pryor S.C. How well are hazards associated with derechos reproduced in regional climate simulations? *Natural Hazards and Earth System Sciences Discussions (*nhess-2021-373)

**Response to reviewer #1:**

[Figure]

**Figure 3: Composite reflectivity (cREF) in domain d03 at $t_p$ (the time when values from the maximum number of grid cells exceeded 40 dBZ) during the Derecho period from RADAR and each WRF ensemble member (times are noted in panel titles). The RADAR panel includes markers showing the presence (white) and absence (black) of thunderstorm reports from ASOS stations in domain d03 in the hour surrounding 03:30 UTC 30 June 2012.**

Shepherd T.J., Letson F., Barthelmie R.J. and Pryor S.C. How well are hazards associated with derechos reproduced in regional climate simulations? *Natural Hazards and Earth System Sciences Discussions (*nhess-2021-373)

**Response to reviewer #1:**

[Figure]

**Figure 4: Composite reflectivity (cREF) in domain d03 at $t_p$ (the time when values from the maximum number of grid cells exceeded 40 dBZ) during the Front period from RADAR and each WRF ensemble member (times are noted in panel titles). The RADAR panel includes markers showing the presence (white) and absence (black) of thunderstorm reports from ASOS stations in domain d03 in the hour surrounding 05:20 UTC 1 July 2012.**

Shepherd T.J., Letson F., Barthelmie R.J. and Pryor S.C. How well are hazards associated with derechos reproduced in regional climate simulations? *Natural Hazards and Earth System Sciences Discussions (*nhess-2021-373)

**Response to reviewer #1:**

[Figure]

**Figure 5: Total accumulated precipitation (mm) from RADAR observations and each WRF ensemble member during the Derecho period. Grid cells with MESH>25mm are marked in magenta.**

Shepherd T.J., Letson F., Barthelmie R.J. and Pryor S.C. How well are hazards associated with derechos reproduced in regional climate simulations? *Natural Hazards and Earth System Sciences Discussions (*nhess-2021-373)

**Response to reviewer #1:**

[Figure]

**Figure 6: Total accumulated precipitation (mm) from RADAR and each WRF ensemble member during the Front period. Grid cells with MESH>25mm are marked in magenta.**

Shepherd T.J., Letson F., Barthelmie R.J. and Pryor S.C. How well are hazards associated with derechos reproduced in regional climate simulations? *Natural Hazards and Earth System Sciences Discussions (*nhess-2021-373)

**Response to reviewer #1:**

4. Upper-level damping is set to be zero, which is needed particularly for convective system simulations.

Past research (e.g. Chen et al. 2019) has indicated damping methods have proven less successful for real atmospheric NWP as opposed to idealized studies, although it is recommended e.g. see: https://agupubs.onlinelibrary.wiley.com/doi/full/10.1029/2019jd030968

Nevertheless, on the advice of the reviewer, we have incorporated the reviewer's suggestion and the recommendation in Chen et al. (2019) and have set upper-level damping (damp_opt = 3) for Rayleigh damping (dampcoef inverse time scale [1/s], e.g. 0.2; for real-data cases – see page 5-104 of the WRF v3.8.1 users guide) in the two new simulations undertaken for this response.

The result of using 65 levels and damping was described in comment #3 above along with the accompanying figures.

5. The authors used some physical scheme combinations that I usually do not use, such as the combination they used for the shortwave and longwave radiation, surface layer, and PBL. Therefore, I am not very sure if all those schemes are well coupled together and work well.

One of the many goals of sensitivity studies is to advance the understanding of scheme coupling and interaction, but according to Kehler-Poljak (2017), based on a literature review of mixed physics studies examining sensitivity to convective parameterization (Mayor and Mesquita 2015), microphysics (Weisman et al. 2008; Givati et al. 2012; Horvath and Vilibić 2014), radiation (Kleczek et al. 2014), land surface scheme (Mohan and Bhati 2011) and PBL scheme (Weisman et al. 2008; Acs et al. 2014; Gómez-Navarro et al. 2015; Cohen et al. 2015; Milovac et al. 2016), while the authors of such studies observed general trends, no single combination of scheme coupling for the simulation of convective events was recommended. According to Clark et al. (2010b), ensemble design and model setup should consider the type of forecast fields desired, regional area, similar types of events, but recognize that these schemes interact in a non-linear manner. Note that our study does not attempt to test multiple ensembles of mixed physics for every type of physics scheme – such a study like Clark et al. (2010b) is not the goal here and would require significant computational resources. As stated, our goal was to assess microphysics, with additional simulations to complement this and assess model configuration.

We have employed schemes that are widely used in literature. In response to the reviewer's comment, using the original publications for the schemes used (shortwave: Dudhia (Dudhia 1989); longwave: RRTM (Mlawer et al. 1997); surface layer: revised MM5 Monin-Obukhov (Jimenez et al. 2012); PBL: MYNN level 2.5 (Nakanishi and Niino 2006)), we have undertaken a citation search for deep convection, mesoscale convective system and derecho papers from the last approx. 10 years that reference those schemes, or where possible, use them as a combination. This list is not meant to be exhaustive, but indicative of the breadth of previous studies. Where possible, we have constrained this list to studies over the US. From the studies below, one can observe that even these published studies do not have a common combination of physics schemes.

Citation search results summary:

Shepherd T.J., Letson F., Barthelmie R.J. and Pryor S.C. How well are hazards associated with derechos reproduced in regional climate simulations? *Natural Hazards and Earth System Sciences Discussions* (nhess-2021-373)

**Response to reviewer #1:**

- Duda and Gallus (2013): used the RRTM and Dudhia radiation schemes with the Thompson microphysics scheme, the MYJ PBL scheme, and the Monin-Obukhov surface layer scheme in a 3 km WRF simulation of two mesoscale convective systems.

- Gensini and Mote (2014): used Dudhia and RRTM radiation schemes alongside MYJ PBL and WSM6 microphysics in dynamically downscaled WRF simulations to resolve March–May hazardous convective weather east of the U.S. Continental Divide for a historical climate period (1980–90). A hazardous convective weather model proxy is used to depict occurrences of tornadoes, damaging thunderstorm wind gusts, and large hail at hourly intervals during the period of record.

- Hariprasad et al. (2014): used RRTM, Dudhia, and revised MM5 in a study examining PBL scheme sensitivity for tropical conditions.

- Bodine and Rasmussen (2017): used RRTM and Dudhia alongside Morrison microphysics, Kain-Fritsch cumulus, and YSU PBL in a high-resolution WRF simulation of the 6 July 2015 MCS in South Dakota. Simulations with Thompson microphysics were also run, but Thompson microphysics produced less realistic stratiform rain regions and did not capture the discrete propagation events as well as the simulations using Morrison microphysics.

- Yang et al. (2017): used RRTM, YSU PBL, and Goddard radiation in a study examining mesoscale convective systems over the central US.

- Wang et al. (2022): used RRTM for longwave and shortwave, YSU PBL, revised MM5 in a study examining lake surface temperature impacts on summertime climate over the Great Lakes region. It was shown that warmer lake surface temperature reduces mesoscale convective precipitation upstream of the Great Lakes region, however, isolated deep convective precipitation is enhanced downstream.

In addition, a paper we already cite in our study (Weisman et al. 2013) notes:
*"Since 2003, real-time explicit convective forecasts at 3–4-km grid spacing have been run at NCAR during each spring to establish the capabilities of the WRF-ARW modeling system (e.g., Skamarock and Klemp 2008) to forecast severe convective events and to test recent model improvements (e.g., Done et al. 2004; Weisman et al. 2008). These forecasts have also been evaluated yearly alongside a host of forecasts obtained from differing modeling configurations and dynamical cores by a variety of modeling groups as part of the Storm Prediction Center (SPC) and National Severe Storms Laboratory (NSSL) Hazardous Weather Test Bed (HWT) Spring Experiment (e.g., Weiss et al. 2004; Kain et al. 2005, 2006, 2008; Coniglio et al. 2010; Clark et al. 2011)."* From Weisman et al. (2013), these well evaluated, real-time explicit convective forecasts use the MYJ PBL scheme, the Noah land surface model, Thompson microphysics, RRTM and Dudhia for longwave and shortwave radiation respectively. Retrospective sensitivity testing between the MYJ and YSU schemes was also undertaken. A discussion of PBL schemes is below.

We appreciate that the combination of surface layer scheme (revised MM5 Monin-Obukhov) and PBL (MYNN) is potentially a combination used in the present study where most uncertainty lies, but our selections are based on supported literature for those schemes and use of these options as standard in WRF (Samelson et al. 2020). There is greater consistency in the use of combined Dudhia and RRTM for

Shepherd T.J., Letson F., Barthelmie R.J. and Pryor S.C. How well are hazards associated with derechos reproduced in regional climate simulations? *Natural Hazards and Earth System Sciences Discussions (*nhess-2021-373)

**Response to reviewer #1:**

radiation. According to Onwukwe and Jackson (2020), the surface-layer schemes compute friction velocity and other exchange coefficients for estimating sensible and latent heat fluxes, and momentum flux from land surface models, and surface stress in the PBL schemes. Furthermore, both the Eta surface layer scheme and revised MM5 are based on similarity theory. This fact accounts for nearly half of the surface layer schemes available in v3.8.1 - the other schemes are less cited in the literature (Pleim-Xiu, QNSE, and TEMF). The Eta similarity, however, includes a parameterization of viscous sublayer according to Janjić (2002), but the underlying theory for each scheme is based on similarity theory. For these reasons, we do not believe the combination used in the present study is a cause for significant concern, especially in the context of existing literature.

We note that several studies have used these schemes in combination. This is a small selection:

- Kumar et al. (2024): used RRTM, Dudhia, MYNN, and revised MM5 in a WRF simulation of a Southern California wildfire event.

- Huang et al. (2023): used revised MM5 and MYNN in convection-permitting simulations of precipitation over the Peruvian Central Andes.

- Minder et al. (2020): used revised MM5 and MYNN in a multi-scheme sensitivity study to help separate sensitivity to PBL and surface layer scheme.

We note that we have previously published this combination of schemes for deep convection simulations over the southern Great Plains – see peer reviewed AMS publications (Letson et al. (2020b); Pryor et al. 2023). Those simulations exhibited some degree of fidelity for key meteorological properties.

Notably, from the citation search, there is a split between use of the YSU and MYJ PBL schemes for the mesoscale convective system studies, with no recommended preference, further reiterating the arbitrary nature of scheme selection by the user. These two PBL schemes use different formulations (non-local [YSU] vs. local closure [MYJ/MYNN]). According to Minder et al. (2020), these variations in PBL scheme formulation have been found to have large impacts on PBL structure and convective storms (e.g., Coniglio et al. 2013; García-Díez et al. 2013; Cohen et al. 2015; Milovac et al. 2016). There are documented biases in the literature for different types of PBL scheme. Weisman et al. (2008) identified biases in a sensitivity study between YSU and MYJ: *"The YSU scheme tends to create boundary layers that are deeper and drier, and is also very aggressive in eliminating capping inversions. On the other hand, the MYJ scheme tends to deepen the boundary layer more slowly, resulting in PBL conditions that are characteristically cooler, moister, and more strongly capped." and "...cases run with the MYJ scheme show that it was consistently better at maintaining the boundary layer moisture than YSU".*

Note that MYNN uses a similar baseline as MYJ, but with considerable modifications of certain parameterizations, e.g., mixing length scale, eddy diffusivities, and stability functions in the stable PBL. The scheme has been modified and improved since its inclusion in WRF. The fact it is based on MYJ makes it a reasonable analog to biases in MYJ, but should be viewed in the context of improvements made to MYNN in recent years.

A study examining the sensitivity and interaction between the land surface schemes (i.e. Noah) and PBL schemes in a case study over western Germany (Milovac et al. 2016), showed that the representation of land surface processes significantly impacts the simulation of mixing properties within the convective

Shepherd T.J., Letson F., Barthelmie R.J. and Pryor S.C. How well are hazards associated with derechos reproduced in regional climate simulations? *Natural Hazards and Earth System Sciences Discussions (*nhess-2021-373)

**Response to reviewer #1:**

boundary layer. In addition, the nonlocal PBL schemes (YSU) simulated a deeper and drier boundary layer than the local schemes (MYJ and MYNN), which is consistent with Weisman et al. (2008). Note that our land surface scheme selection in the present study (Noah) is consistent with the literature.

In summary, we appreciate the reviewer's comments about scheme selection and coupling – our choices were not random, but based on existing literature, known trade-offs and biases, and ultimately informed by science.

6. Another note is that the radiation schemes they only take the hydrometeor mass from the cloud microphysics calculation. The cloud droplet and ice effective radii are fixed in the radiation scheme and not from cloud microphysics calculation. For better coupling of cloud microphysics with radiation, the default WRF is not enough and the users need to do the coupling in cloud and ice effective radii.

We appreciate the reviewer's comment, but such an exercise is a model/scheme development study and is outside the scope of the current study (an ensemble sensitivity study). Indeed, previous sensitivity studies like the present study run the microphysics schemes with no user-specific code modification. The literature we have cited here that uses the specific radiation schemes, includes no specific modification to the default WRF schemes.

The large differences (100% or even more) in their tests between NERSC Cori and their university supercomputer for the Morrison scheme with the hail option (Figure 13b) is an indication that the code or computer is not stable. Changing compilers or computer would never cause such large differences which basically make the model simulations not trustable. Note the code instability may not be shown in any tests and might only be triggered in some conditions. Since WRF has so many options for physics and model configurations, and many options were plugged in only for some specific schemes and might not be tested for other schemes, users need to be very careful and better work with experienced WRF modelers or relevant scheme developers to get things figured out when unexpected/unusual results are obtained.

We showed in our previous response that the simulations between platforms, and the Intel vs GNU tests are quite similar. We have worked with scheme developers– we spoke with Hugh Morrison to gain an appreciation for the difference between the Morrison simulations and we communicated that in our previous response.

With appreciation for the reviewers careful attention to our paper, and our lengthy and detailed responses and additional simulations we hope that this process can now be drawn to a conclusion and our paper move into publication.
Thanks to everyone for their time and effort on this.

**References**

Acs, F., Gyöngyösi, A. Z., Breuer, H., Horváth, A., Mona, T., & Rajkai, K. (2014). Sensitivity of WRF-simulated planetary boundary layer height to land cover and soil changes. Meteorologische Zeitschrift, 23, 279–293.

Shepherd T.J., Letson F., Barthelmie R.J. and Pryor S.C. How well are hazards associated with derechos reproduced in regional climate simulations? *Natural Hazards and Earth System Sciences Discussions (*nhess-2021-373)

**Response to reviewer #1:**

Bodine, D. J., and K. L. Rasmussen, 2017: Evolution of Mesoscale Convective System Organizational Structure and Convective Line Propagation. Mon. Wea. Rev., 145, 3419–3440, https://doi.org/10.1175/MWR-D-16-0406.1.

Chen, Y., Yuan, H., and Gao, S.: A High-Resolution Simulation of Roll Convection Over the Yellow Sea During a Cold Air Outbreak, Journal of Geophysical Research: Atmospheres, 124, 10608-10625, https://doi.org/10.1029/2019JD030968, 2019.

Clark, A. J., W. Gallus, and T. Chen, 2007: Comparison of the diurnal precipitation cycle in convection-resolving and non-convection-resolving mesoscale models. Mon. Wea. Rev., 135, 3456–3473.

Clark, A. J., W. A. Gallus , M. Xue, and F. Kong, 2010b: Growth of Spread in Convection-Allowing and Convection-Parameterizing Ensembles. Wea. Forecasting, 25, 594–612, https://doi.org/10.1175/2009WAF2222318.1.

Cohen, A. E., Cavallo, S. M., Coniglio, M. C., & Brooks, H. E. (2015). A review of planetary boundary layer parameterization schemes and their sensitivity in simulating southeastern U.S. cold season severe weather environments. Journal of Applied Meteorology and Climatology, 30, 591–612.

Coniglio, M. C., J. Correia Jr., P. T. Marsh, and F. Kong, 2013: Verification of convection-allowing WRF model forecasts of the planetary boundary layer using sounding observations. Wea. Forecasting, 28, 842–862, https://doi.org/10.1175/WAF-D-12-00103.1.

Done, J., C. A. Davis, and M. L. Weisman, 2004: The next generation of NWP: Explicit forecasts of convection using the Weather Research and Forecast (WRF) model. Atmos. Sci. Lett., 5, 110–117.

Duda, J. D., and W. A. Gallus , 2013: The Impact of Large-Scale Forcing on Skill of Simulated Convective Initiation and Upscale Evolution with Convection-Allowing Grid Spacings in the WRF. Wea. Forecasting, 28, 994–1018, https://doi.org/10.1175/WAF-D-13-00005.1.

Garcia-Carreras, L., A. J. Challinor, B. J. Parkes, C. E. Birch, K. J. Nicklin, and D. J. Parker, 2015: The Impact of Parameterized Convection on the Simulation of Crop Processes. *J. Appl. Meteor. Climatol.*, 54, 1283–1296, https://doi.org/10.1175/JAMC-D-14-0226.1

García-Díez, M., J. Fernández, L. Fita, and C. Yagüe, 2013: Seasonal dependence of WRF model biases and sensitivity to PBL schemes over Europe. Quart. J. Roy. Meteor. Soc., 139, 501–514, https://doi.org/10.1002/qj.1976.

Gensini, V. A., and T. L. Mote, 2014: Estimations of Hazardous Convective Weather in the United States Using Dynamical Downscaling. J. Climate, 27, 6581–6589, https://doi.org/10.1175/JCLI-D-13-00777.1.

Givati, A., Lynn, B., Liu, Y., & Rimmer, A. (2012). Using the WRF model in an operational stream forecast system for the Jordan River. Journal of Applied Meteorology and Climatology, 51, 285–299.

Shepherd T.J., Letson F., Barthelmie R.J. and Pryor S.C. How well are hazards associated with derechos reproduced in regional climate simulations? *Natural Hazards and Earth System Sciences Discussions (*nhess-2021-373)

**Response to reviewer #1:**

Gómez-Navarro, J. J., Raible, C. C., & Dierer, S. (2015). Sensitivity of the WRF model to PBL parametrisations and nesting techniques: Evaluation of wind storms over complex terrain. Geoscientific Model Development, 8, 3349–3363.

Halder, M., Mukhopadhyay, P. Microphysical processes and hydrometeor distributions associated with thunderstorms over India: WRF (cloud-resolving) simulations and validations using TRMM. Nat Hazards 83, 1125–1155 (2016). https://doi.org/10.1007/s11069-016-2365-2

Hariprasad, K.B.R.R., C.V. Srinivas, A.Bagavath Singh, S. Vijaya Bhaskara Rao, R. Baskaran, B. Venkatraman, Numerical simulation and intercomparison of boundary layer structure with different PBL schemes in WRF using experimental observations at a tropical site, Atmospheric Research, 2014, https://doi.org/10.1016/j.atmosres.2014.03.023.

Horvath, K., & Vilibić, I. (2014). Atmospheric mesoscale conditions during the Boothbay meteotsunami: a numerical sensitivity study using a high-resolution mesoscale model. Natural Hazards, 74, 55–74.

Huang, Y., M. Xue, X. Hu, E. Martin, H. M. Novoa, R. A. McPherson, A. Perez, and I. Y. Morales, 2023: Convection-Permitting Simulations of Precipitation over the Peruvian Central Andes: Strong Sensitivity to Planetary Boundary Layer Parameterization. J. Hydrometeor., 24, 1969–1990, https://doi.org/10.1175/JHM-D-22-0173.1.

Janjić, Z. I., 2002: Nonsingular implementation of the Mellor-Yamada level 2.5 scheme in the NCEP meso model. NCEP Office Note 437, 61 pp., https://repository.library.noaa.gov/view/noaa/11409.

Kain, J. S., and Coauthors, 2008: Some practical considerations regarding horizontal resolution in the first generation of operational convection-allowing NWP. Wea. Forecasting, 23, 931–952.

Kehler-Poljak, G., Telišman Prtenjak, M., Kvakić, M. et al. Interaction of Sea Breeze and Deep Convection over the Northeastern Adriatic Coast: An Analysis of Sensitivity Experiments Using a High-Resolution Mesoscale Model. Pure Appl. Geophys. 174, 4197–4224 (2017). https://doi.org/10.1007/s00024-017-1607-x

Khai Shen Sow, Liew Juneng, Fredolin T. Tangang, Abdul Ghapor Hussin, Mastura Mahmud, Numerical simulation of a severe late afternoon thunderstorm over Peninsular Malaysia, *Atmospheric Research*, 99, 2, 2011, https://doi.org/10.1016/j.atmosres.2010.10.014

Kleczek, M. A., Steeneveld, G.-J., & Holtslag, A. A. M. (2014). Evaluation of the weather research and forecasting mesoscale model for GABLS3: Impact of boundary-layer schemes, boundary conditions and spin-up. Boundary-Layer Meteorology, 152, 213–243.

Kumar, M., Kosović, B., Nayak, H. P., Porter, W. C., Randerson, J. T., and Banerjee, T.: Evaluating the performance of WRF in simulating winds and surface meteorology during a Southern California wildfire event, Frontiers in Earth Science, 11, 10.3389/feart.2023.1305124, 2024.

Letson, F., T. J. Shepherd, R. J. Barthelmie, and S. C. Pryor, 2020b: WRF modelling of deep convection and hail for wind power applications. J. Appl. Meteor. Climatol., 59, 1717–1733, https://doi.org/10.1175/JAMC-D-20-0033.1.

Shepherd T.J., Letson F., Barthelmie R.J. and Pryor S.C. How well are hazards associated with derechos reproduced in regional climate simulations? *Natural Hazards and Earth System Sciences Discussions (*nhess-2021-373)

**Response to reviewer #1:**

Liu, C., and M. W. Moncrieff, 2007: Sensitivity of cloud-resolving simulations of warm-season convection to cloud microphysics parameterizations. Mon. Wea. Rev., 135, 2854–2868.

Mayor, Y. G., & Mesquita, M. D. S. (2015). Numerical simulations of the 1 May 2012 deep convection event over Cuba: Sensitivity to cumulus and microphysical schemes in a high-resolution model. Advances in Meteorology, 2015. doi:10.1155/2015/973151

Milovac, J., Warrach-Sagi, K., Behrendt, A., Späth, F., Ingwersen, J., & Wulfmeyer, V. (2016). Investigation of PBL schemes combining the WRF model simulations with scanning water vapor differential absorption lidar measurements. Journal of Geophysical Research, 121, 624–649 (2016). doi:10.1002/2015JD023927.

Minder, J. R., W. M. Bartolini, C. Spence, N. R. Hedstrom, P. D. Blanken, and J. D. Lenters, 2020: Characterizing and Constraining Uncertainty Associated with Surface and Boundary Layer Turbulent Fluxes in Simulations of Lake-Effect Snowfall. Wea. Forecasting, 35, 467–488, https://doi.org/10.1175/WAF-D-19-0153.1.

Mohan, M., & Bhati S., (2011). Analysis of WRF model performance over subtropical region of Delhi, India. Advances in Meteorology, 2011. doi:10.1155/2011/621235.

Onwukwe, C., and P. L. Jackson, 2020: Meteorological Downscaling with WRF Model, Version 4.0, and Comparative Evaluation of Planetary Boundary Layer Schemes over a Complex Coastal Airshed. J. Appl. Meteor. Climatol., 59, 1295–1319, https://doi.org/10.1175/JAMC-D-19-0212.1.

Pryor, S. C., F. Letson, T. Shepherd, and R. J. Barthelmie, 2023: Evaluation of WRF Simulation of Deep Convection in the U.S. Southern Great Plains. J. Appl. Meteor. Climatol., 62, 41–62, https://doi.org/10.1175/JAMC-D-22-0090.1.

Rajeevan, M., Kesarkar, A., Thampi, S. B., Rao, T. N., Radhakrishna, B., and Rajasekhar, M.: Sensitivity of WRF cloud microphysics to simulations of a severe thunderstorm event over Southeast India, Ann. Geophys., 28, 603–619, https://doi.org/10.5194/angeo-28-603-2010 2010

Romine, G. S., C. S. Schwartz, C. Snyder, J. L. Anderson, and M. L. Weisman, 2013: Model Bias in a Continuously Cycled Assimilation System and Its Influence on Convection-Permitting Forecasts. Mon. Wea. Rev., 141, 1263–1284, https://doi.org/10.1175/MWR-D-12-00112.1.

Samelson, R. M., L. W. O'Neill, D. B. Chelton, E. D. Skyllingstad, P. L. Barbour, and S. M. Durski, 2020: Surface Stress and Atmospheric Boundary Layer Response to Mesoscale SST Structure in Coupled Simulations of the Northern California Current System. Mon. Wea. Rev., 148, 259–287, https://doi.org/10.1175/MWR-D-19-0200.1.

Schwartz, C. S., and Coauthors, 2009: Next-day convection-allowing WRF model guidance: A second look at 2-km versus 4-km grid spacing. Mon. Wea. Rev., 137, 3351–3372.

Shepherd T.J., Letson F., Barthelmie R.J. and Pryor S.C. How well are hazards associated with derechos reproduced in regional climate simulations? *Natural Hazards and Earth System Sciences Discussions* *(*nhess-2021-373)

**Response to reviewer #1:**

Sobash, R., J. Kain, D. Bright, A. Dean, M. Coniglio, and S. Weiss, 2011: Probabilistic forecast guidance for severe thunderstorms based on the identification of extreme phenomena in convection-allowing model forecasts. Wea. Forecasting, 26, 714–728.

Wang, J., Xue, P., Pringle, W., Yang, Z., & Qian, Y. (2022). Impacts of lake surface temperature on the summer climate over the Great Lakes Region. Journal of Geophysical Research: Atmospheres, 127, e2021JD036231. https://doi.org/10.1029/2021JD036231

Weisman, M. L., Christopher, D., Wang, W., Manning, K. W., & Klemp, J. B. (2008). Experiences with 0–36-h explicit convective forecasts with the WRF-ARW model. Weather and Forecasting, 23, 407–437.

Xing Yu & Tae-Young Lee (2010) Role of convective parameterization in simulations of a convection band at grey-zone resolutions, Tellus A: Dynamic Meteorology and Oceanography, 62:5, 617-632, DOI: 10.1111/j.1600-0870.2010.00470.x